# Generic bounds on the approximation error for physics-informed (and) operator learning

**Tim De Ryck** [*]

**Siddhartha Mishra** [†]

## Abstract

We propose a very general framework for deriving rigorous bounds on the approximation error for physics-informed neural networks (PINNs) and operator learning architectures such as DeepONets and FNOs as well as for physics-informed operator learning. These bounds guarantee that PINNs and (physics-informed) DeepONets or FNOs will efficiently approximate the underlying solution or solution operator of generic partial differential equations (PDEs). Our framework utilizes existing neural network approximation results to obtain bounds on more involved learning architectures for PDEs. We illustrate the general framework by deriving the first rigorous bounds on the approximation error of physics-informed operator learning and by showing that PINNs (and physics-informed DeepONets and FNOs) mitigate the curse of dimensionality in approximating nonlinear parabolic PDEs.

## 1 Introduction

The efficient numerical approximation of partial differential equations (PDEs) is of paramount importance as PDEs mathematically describe an enormous range of interesting phenomena in the sciences and engineering. Machine learning techniques, particularly deep learning, are playing an increasingly important role in this context. For instance, given their universal approximation properties, deep neural networks serve as ansatz spaces for supervised learning of a variety of (parametric) PDEs [19, 70, 40, 53, 54] and references therein. In this setting, large amounts of training data might be required. However, this data is often acquired from expensive computer simulations or physical measurements [53], necessitating the design of learning frameworks that work with limited data. *Physics-informed neural networks (PINNs)*, proposed by [18, 42, 41] and popularized by [67, 68], are a prominent example of such a learning framework as the residual of the underlying PDE is minimized within the class of neural networks and in principle, little (or even no) training data is required. PINNs and their variants have proven to be a very powerful and computationally efficient framework for approximating solutions to PDEs, [69, 51, 55, 65, 76, 34, 35, 61, 59, 60, 2] and references therein.

Often in the context of PDEs, one needs to approximate the underlying solution operator that maps one infinite-dimensional function space into another [27, 39]. As neural networks can only map between finite dimensional spaces, a new field of *operator learning* is emerging wherein novel learning frameworks need to be designed in order to approximate operators. These include deep operator networks (DeepONets) [9, 49] and their variants as well as *neural operators* [39], which generalize neural networks to this setting. A variety of neural operators have been proposed, see [45, 46] but arguably, the most efficient form of neural operators is provided by the so-called *Fourier neural operators* (FNOs) [44]. Both DeepONets and FNOs have been very successfully deployed in

---

[*]Seminar for Applied Mathematics (SAM), D-MATH, ETH Zürich, Switzerland

[†]Seminar for Applied Mathematics (SAM), D-MATH and ETH AI center, ETH Zürich, Switzerland

36th Conference on Neural Information Processing Systems (NeurIPS 2022).

scientific computing [50, 56, 8, 48, 47, 66] and references therein. Finally, one can combine PINNs and operator learning to design physics-informed DeepONets/FNOs [74, 47, 73, 22].

From a theoretical perspective, one needs to provide a rigorous guarantee that the learning framework can approximate the underlying PDE solution (operator) to desired accuracy. More precisely, given an error tolerance $\varepsilon > 0$, we need to rigorously prove that the approximation error of the neural network (operator) can be made smaller than $\varepsilon$. For *efficient approximation*, one has to further ensure that the computational complexity (measured in terms of the size) of the learning architecture grows at most *polynomially* in $\varepsilon^{-1}$. In particular, *exponential* growth has to be ruled out. As neural networks, DeepONets and FNOs are all *universal approximators* [13, 9, 49, 43, 38] of the underlying functions or operators, it is possible to show that the approximation error can be made as small as desired. However, these results do not guarantee efficient approximation as the underlying network size could still grow exponentially with decreasing error, see [77] for neural networks in very high spatial dimensions, [43] for DeepONets and [38] for FNOs. Hence, the real theoretical challenge in this context lies in proving efficient approximation results for the different learning architectures in scientific computing. Such efficient approximation results have mostly been obtained for neural networks in the supervised learning setting e.g. [23, 36, 32, 4, 63] and references therein. In contrast, there is a relative scarcity of such efficient approximation results for PINNs and operator learning with notable exceptions being [25, 15, 16] (for PINNs), [43] (for DeepONets) and [38] (for FNOs). Moreover, the underlying proofs in these works are often on a case-by-case basis and the overall abstract structure is not clearly identified. Finally, no similar rigorous approximation results for physics informed operator learning are available till date.

This paucity of generic efficient approximation results for PINNs and operator learning for PDEs sets the stage for the current paper where our main contribution is to propose a very general *framework* (Section 3) for proving bounds on the approximation error for space-time neural networks, PINNs, DeepONets, FNOs and physics-informed DeepONets and FNOs for very general PDEs. Consequently, we obtain the first rigorous bounds for physics-informed operator learning in literature. Our framework is based on the observation that error estimates for different types of neural network architectures can all be obtained from one another. As error estimates for neural network approximations of PDE solutions at a fixed time are the easiest to obtain, and hence constitute the largest proportion of currently available estimates, we devote particular attention to demonstrating how these available estimates can be used to obtain novel bounds on the approximation error for space-time networks, PINNs and (physics-informed) operator learning. Our results provide a *roadmap* for deriving mathematical guarantees for deep learning methods in scientific computing by simplifying the proofs, as the needed work essentially reduces to verifying a small number of assumptions. We demonstrate how the generic error bounds from Section 3 can be applied in practice in Section 4, among others by giving short alternative proofs for known results and also proving a number of novel results. In particular, we show in Section 4.1 that PINNs can overcome the curse of dimensionality for nonlinear parabolic PDEs such as the Allen-Cahn equation i.e., that the network size does not grow exponentially with increasing spatial dimension. Moreover, dimension-independent convergence rates are also obtained for (physics-informed) DeepONets and FNOs, provided that the PDE solutions are sufficiently smooth. These are the first results of their kind. We note that many of the proofs and some examples are deferred to the supplementary material (**SM**).

## 2 Preliminaries

### 2.1 Setting

Given $T > 0$ and $D \subset \mathbb{R}^d$ compact, consider the function $u : [0,T] \times D \to \mathbb{R}^m$, for $m \geq 1$, that belongs to a function space $\mathcal{H}$ and solves the following (time-dependent) PDE,

$$\mathcal{L}_a(u)(t,x) = 0 \quad \text{and} \quad u(x,0) = u_0 \qquad \forall (t,x) \in [0,T] \times D, \tag{2.1}$$

where $u_0 \in \mathcal{Y} \subset L^2(D)$ is the initial condition and $\mathcal{L}_a \colon \mathcal{H} \to L^2([0,T] \times D)$ is a differential operator that can depend on a parameter (function) $a \in \mathcal{Z} \subset L^2(D)$. In our notation, we will often suppress the dependence of $\mathcal{L} := \mathcal{L}_a$ on $a$ for simplicity. Depending on the context, one might want to recover one of the following mathematical objects: for fixed $a$ and $u_0$, one might want to approximate $u(T, \cdot)$ or $u(\cdot, \cdot)$ with a neural network; a more challenging task would be to learn the *solution operator* $\mathcal{G} : \mathcal{X} \to L^2(\Omega) : v \mapsto u$, where $v \in \{u_0, a\}$, $\mathcal{X} \in \{\mathcal{Y}, \mathcal{Z}\}$ and $\Omega = D$ or

$\Omega = [0, T] \times D$. We will use this notation consistently throughout the paper, see **SM** A.1 for an overview.

## 2.2 Approximating PDEs with neural networks

**Neural networks** A (feedforward) neural network $u_\theta : \mathbb{R}^{d_0} \to \mathbb{R}^{d_L}$ is defined as a concatenation of affine maps $\mathcal{A}_l : \mathbb{R}^{d_{l-1}} \to \mathbb{R}^{d_l} : z \mapsto W_l z + b_l$ and an activation function $\sigma : \mathbb{R} \to \mathbb{R}$ that is applied component-wise, resulting in,

$$u_\theta(y) = \mathcal{A}_L \circ \sigma \circ \mathcal{A}_{L-1} \ldots \ldots \ldots \circ \sigma \circ \mathcal{A}_2 \circ \sigma \circ \mathcal{A}_1(y). \qquad (2.2)$$

The weights and biases of the affine maps $\theta = \{W_l, b_l\}_{1 \le l \le L}$ are the trainable parameters. We will quantify the size of a neural network by its $\mathrm{depth}(u_\theta) := L$ and its $\mathrm{width}(u_\theta) := \max_l d_l$. In order to obtain a neural network that approximates the solution $u$ of PDE (2.1) at time $t = T$, one chooses the parameters of $u_\theta : D \to \mathbb{R}$ such that a discretization (quadrature) of $\mathcal{J}(\theta) = \|u(T) - u_\theta\|_{L^2(D)}$ is minimized. The training data is acquired from either measurements or potentially expensive simulations.

**PINNs** Physics-informed neural networks (PINNs) are neural networks that are trained with a different, residual-based loss function. As the PDE solution $u$ satisfies $\mathcal{L}(u) = 0$, the goal of physics-informed learning is to find a neural network $u_\theta : [0, T] \times D \to \mathbb{R}$ for which the PDE residual is approximately zero, $\mathcal{L}(u_\theta) \approx 0$. To ensure uniqueness, one also needs to require that the initial condition is satisfied i.e., $u_\theta(0, x) \approx u_0(x)$, and similarly for boundary conditions. In practice one minimizes a quadrature approximation of $\mathcal{J}(\theta) = \|\mathcal{L}(u_\theta)\|^2_{L^2([0,T] \times D)} + \|u_\theta(0, \cdot) - u_0\|^2_{L^2(D)}$, where additional terms can be added to (approximately) impose boundary conditions and augment the loss function using data. A desirable property of PINNs is that only very little or even no training data is needed to construct the loss function.

**Operator learning** In order to approximate operators, one needs to allow the input and output of the learning architecture to be infinite-dimensional. A possible approach is to use *deep operator networks* (DeepONets), as proposed in [9, 49]. Given $m$, fixed sensor locations $\{x_j\}_{j=1}^m \subset D$ and the corresponding *sensor values* $\{v(x_j)\}_{j=1}^m$ as input, a DeepONet can be formulated in terms of two (deep) neural networks: a *branch net* $\beta : \mathbb{R}^m \to \mathbb{R}^p$ and a *trunk net* $\tau : D \to \mathbb{R}^{p+1}$. The branch and trunk nets are then combined to approximate the underlying nonlinear operator as the following *DeepONet* $\mathcal{G}_\theta : \mathcal{X} \to L^2(D)$, with $\mathcal{G}_\theta(v)(y) = \tau_0(y) + \sum_{k=1}^p \beta_k(v)\tau_k(y)$. A second approach is that of *neural operators*, which generalize hidden layers by including a non-local integral operator [45], of which particularly *Fourier neural operators* (FNOs) [44] are already well-established. The practical implementation (i.e. discretization) of an FNO maps from and to the space of trigonometric polynomials of degree at most $N \in \mathbb{N}$, denoted by $L^2_N$, and can be identified with a finite-dimensional mapping that is a composition of affine maps and nonlinear layers of the form $\mathfrak{L}_l(z)_j = \sigma(W_l v_j + b_{l,j} \mathcal{F}_N^{-1}(P_l(k) \cdot \mathcal{F}_N(z)(k)_j))$, where the $P_l(k)$ are coefficients that define a non-local convolution operator via the discrete Fourier transform $\mathcal{F}_N$, see [38].

**Physics-informed operator learning** Both DeepONets and FNOs are trained by choosing a suitable probability measure $\mu$ on $\mathcal{X}$ and minimizing a quadrature approximation of $\mathcal{J}(\theta) = \|\mathcal{G}_\theta(v) - \mathcal{G}(v)\|_{L^2_{\mu \times dx}(\mathcal{X} \times \Omega)}$. Generating training sets might require many calls to an expensive PDE solver, leading to an enormous computational cost. In order to reduce or even fully eliminate the need for training data, *physics-informed operator learning* has been proposed in [74] for DeepONets and in [47] for FNOs. Similar to PINNs, the training procedure aims to minimize a quadrature approximation of $\mathcal{J}(\theta) = \|\mathcal{L}(\mathcal{G}_\theta)\|_{L^2_{\mu \times dx}(\mathcal{X} \times \Omega)}$.

## 3 General results

We propose a framework to obtain bounds on the approximation error for the various neural network architectures introduced in Section 2.2. Figure 1 visualizes how different types of error estimates can be obtained from one another. Every box shows the name of the network architecture, the form of the relevant loss and the theorem which proves the corresponding estimate for the approximation error. Every arrow in the flowchart represents a proof technique that allows one to transfer an error estimate from one type of method to another (see caption of Figure 1 for an overview of those techniques).

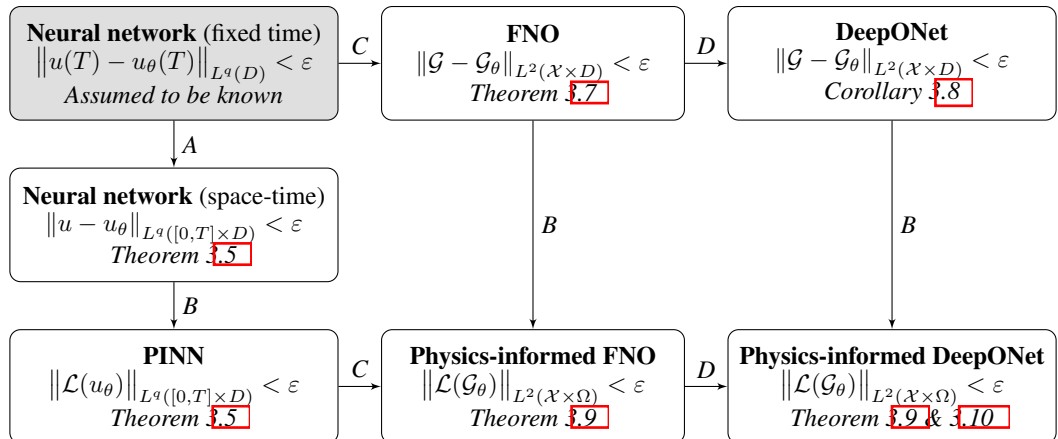

Figure 1: Flowchart of the structure of the results in this paper, with $q \in \{2, \infty\}$. The letters reflect the techniques used in the proofs: *A* uses Taylor approximations (Section 3.1), *B* is based on finite difference approximations (Section 3.1), *C* uses trigonometric polynomial interpolation (Section 3.2) and *D* uses the connection between FNOs and DeepONets (Section 3.2).

We give particular attention to the case where it is known that a neural network can efficiently approximate the solution to a time-dependent PDE at a fixed time. Such neural networks are usually obtained by emulating a classical numerical method. Examples include finite difference schemes, finite volume schemes, finite element methods, iterative methods and Monte Carlo methods, e.g. [36, 63, 10, 57]. More precisely, for $\varepsilon > 0$, we assume to have access to an operator $\mathcal{U}^\varepsilon : \mathcal{X} \times [0,T] \to \mathcal{H}$ that for any $t \in [0,T]$ maps any initial condition/parameter function $v \in \mathcal{X}$ to a neural network $\mathcal{U}^\varepsilon(v,t)$ that approximates the PDE solution $\mathcal{G}(v)(\cdot,t) = u(\cdot,t) \in L^q(D)$, $q \in \{2, \infty\}$, at time $t$, as specified below. Moreover, we will assume that we know how its size depends on the accuracy $\varepsilon$. Explicit examples of the operator $\mathcal{U}^\varepsilon$ will be given in Section 4 and **SM** C.

**Assumption 3.1.** *Let $q \in \{2, \infty\}$. For any $B, \varepsilon > 0$, $\ell \in \mathbb{N}$, $t \in [0,T]$ and any $v \in \mathcal{X}$ with $\|v\|_{C^\ell} \le B$ there exist a neural network $\mathcal{U}^\varepsilon(v,t) : D \to \mathbb{R}$ and a constant $C^B_{\varepsilon,\ell} > 0$ s.t.*

$$\left\| \mathcal{U}^\varepsilon(v,t) - \mathcal{G}(v)(\cdot,t) \right\|_{L^q(D)} \le \varepsilon \quad and \quad \max_{t \in [0,T]} \left\| \mathcal{U}^\varepsilon(v,t) \right\|_{W^{\ell,q}(D)} \le C^B_{\varepsilon,\ell}. \tag{3.1}$$

**Remark 3.2.** *For vanilla neural networks and PINNs one can set $\mathcal{X} := \{v\}$, $\mathcal{G}(v) := u$ and $v := u_0$ or $v := a$ in Assumption 3.1 above and Assumption 3.4 below.*

Under this assumption, we prove the existence of space-time neural networks and PINNs that efficiently approximate the PDE solution (Section 3.1), as well as FNOs and DeepONets (Section 3.2) and physics-informed FNOs and DeepONets (Section 3.3). Finally, we also prove a general result on the generalization error (Section 3.4).

## 3.1 Estimates for (physics-informed) neural networks

We will construct a space-time neural network $u_\theta$ for which both $\|u_\theta - u\|_{L^q([0,T] \times D)}$ and the PINN loss $\|\mathcal{L}(u_\theta)\|_{L^q([0,T] \times D)}$ are small. To accurately approximate the time derivatives of $u$ we emulate Taylor expansions, whereas for the spatial derivatives we employ finite difference (FD) operators in our proofs. Depending on whether forward, backward or central differences are used, a FD operator might not be defined on the whole domain $D$, e.g. for $f \in C([0,1])$ the (forward) operator $\Delta^+_h[f] := f(x+h) - f(x)$ is not well-defined for $x \in (1-h, 1]$. This can be solved by resorting to piecewise-defined FD operators, e.g. a forward operator on $[0, 0.5]$ and a backward operator on $(0.5, 1]$. In a general domain $\Omega$ one can find a well-defined piecewise FD operator if $\Omega$ satisfies the following assumption, which is satisfied by many domains (e.g. rectangular, smooth).

**Assumption 3.3.** *There exists a finite partition $\mathcal{P}$ of $\Omega$ such that for all $P \in \mathcal{P}$ there exists $\varepsilon_P > 0$ and $v_P \in B^1_\infty = \{x \in \mathbb{R}^{\dim(\Omega)} : \|x\|_\infty \le 1\}$ such that for all $x \in P$ it holds that $x + \varepsilon_P(v_P + B^1_\infty) \subset \Omega$.*

Additionally, we need to assume that the PINN error can be bounded in terms of the errors related to all relevant partial derivatives, denoted by $D^{(k,\boldsymbol{\alpha})} := D^k_t D^{\boldsymbol{\alpha}}_x := \partial^k_t \partial^{\alpha_1}_{x_1} \dots \partial^{\alpha_d}_{x_d}$, for $(k, \boldsymbol{\alpha}) \in \mathbb{N}^{d+1}_0$.

This assumption is valid for many classical solutions of PDEs. A few worked out examples can be found in **SM** D.5 (gravity pendulum) and **SM** D.6 (Darcy flow).

**Assumption 3.4.** *Let* $k, \ell \in \mathbb{N}$, $q \in \{2, \infty\}$, $C > 0$ *be independent from* $d$. *For all* $v \in \mathcal{X}$ *it holds,*

$$\left\|\mathcal{L}(\mathcal{G}_\theta(v))\right\|_{L^q([0,T]\times D)} \leq C \cdot \text{poly}(d) \cdot \sum_{\substack{(k', \boldsymbol{\alpha}) \in \mathbb{N}_0^{d+1} \\ k' \leq k, \|\boldsymbol{\alpha}\|_1 \leq \ell}} \left\|D^{(k', \boldsymbol{\alpha})}(\mathcal{G} - \mathcal{G}_\theta)\right\|_{L^q([0,T]\times D)}. \tag{3.2}$$

In this setting, we prove the following approximation result for space-time networks and PINNs.

**Theorem 3.5.** *Let* $s, r \in \mathbb{N}$, *let* $u \in C^{(s,r)}([0,T] \times D)$ *be the solution of the PDE* (2.1) *and let Assumption* 3.1 *be satisfied. There exists a constant* $C(s,r) > 0$ *such that for every* $M \in \mathbb{N}$ *and* $\varepsilon, h > 0$ *there exists a tanh neural network* $u_\theta : [0,T] \times D \to \mathbb{R}$ *for which it holds that,*

$$\|u_\theta - u\|_{L^q([0,T]\times D)} \leq C(\|u\|_{C^{(s,0)}} M^{-s} + \varepsilon). \tag{3.3}$$

*and if additionally Assumption* 3.3 *and Assumption* 3.4 *hold then,*

$$\begin{aligned}
\left\|\mathcal{L}(u_\theta)\right\|_{L^2([0,T]\times D)} &+ \|u_\theta - u\|_{L^2(\partial([0,T]\times D))} \\
&\leq C \cdot \text{poly}(d) \cdot \ln^k(M)(\|u\|_{C^{(s,\ell)}} M^{k-s} + M^{2k}(\varepsilon h^{-\ell} + C_{\varepsilon,\ell}^B h^{r-\ell})).
\end{aligned} \tag{3.4}$$

*Moreover,* $\text{depth}(u_\theta) \leq C \cdot \text{depth}(\mathcal{U}^\varepsilon)$ *and* $\text{width}(u_\theta) \leq CM \cdot \text{width}(\mathcal{U}^\varepsilon)$.

*Proof.* We only provide a sketch of the full proof (**SM** B.2). The main idea is to divide $[0,T]$ into $M$ uniform subintervals and construct a neural network that approximates a Taylor approximation in time of $u$ in each subinterval. In the obtained formula, we approximate the monomials and multiplications by neural networks (**SM** A.7) and approximate the derivatives of $u$ by finite differences and use (A.2) of **SM** A.2 to find an error estimate in $C^k([0,T], L^q(D))$-norm. We use again finite difference operators to prove that spatial derivatives of $u$ are accurately approximated as well. The neural network will also approximately satisfy the initial/boundary conditions as $\|u_\theta - u\|_{L^2(\partial([0,T]\times D))} \lesssim C\text{poly}(d)\|u_\theta - u\|_{H^1([0,T]\times D)}$, which follows from a Sobolev trace inequality. $\square$

We note that the bounds (3.3) and (3.4) together imply that there exists a neural network for which the total error as well as the PINN loss can be made as small as possible, providing a solid theoretical foundation to PINNs for approximating the PDE (2.1).

## 3.2 Estimates for operator learning

In this section, we use Assumption 3.1 to prove estimates for DeepONets and FNOs. First, we prove a generic error estimate for FNOs. Using the known connection between FNOs and DeepONets (**SM** Lemma B.6) this result can then easily be applied to DeepONets (Corollary 3.8). In order to prove these error estimates, we need to assume that the operator $\mathcal{U}^\varepsilon$ from Assumption 3.1 is stable with respect to its input function, as specified in Assumption 3.6 below. Moreover, we will take the $d$-dimensional torus as domain $D = \mathbb{T}^d = [0, 2\pi)^d$ and assume periodic boundary conditions for simplicity in what follows. This is not a restriction, as for every Lipschitz subset of $\mathbb{T}^d$ there exists a (linear and continuous) $\mathbb{T}^d$-periodic extension operator of which also the derivatives are $\mathbb{T}^d$-periodic [38, Lemma 41].

**Assumption 3.6.** *Assumption* 3.1 *is satisfied and let* $p \in \{2, \infty\}$. *For every* $\varepsilon > 0$ *there exists a constant* $C_{\text{stab}}^\varepsilon > 0$ *such that for all* $v, v' \in \mathcal{X}$ *it holds that,*

$$\left\|\mathcal{U}^\varepsilon(v,T) - \mathcal{U}^\varepsilon(v',T)\right\|_{L^2} \leq C_{\text{stab}}^\varepsilon \left\|v - v'\right\|_{L^p}. \tag{3.5}$$

In this setting, we prove a generic approximation result for FNOs.

**Theorem 3.7.** *Let* $r \in \mathbb{N}$, $T > 0$, *let* $\mathcal{G} : C^r(\mathbb{T}^d) \to C^r(\mathbb{T}^d)$ *be an operator that maps a function* $u_0$ *to the solution* $u(\cdot, T)$ *of the PDE* (2.1) *with initial condition* $u_0$, *let Assumption* 3.6 *be satisfied and let* $p^* \in \{2, \infty\} \setminus \{p\}$. *Then there exists a constant* $C > 0$ *such that for every* $\varepsilon > 0$, $N \in \mathbb{N}$ *there is an FNO* $\mathcal{G}_\theta : L_N^2(\mathbb{T}^d) \to L_N^2(\mathbb{T}^d)$ *of depth* $\mathcal{O}(\text{depth}(\mathcal{U}^\varepsilon))$ *and width* $\mathcal{O}(N^d\text{width}(\mathcal{U}^\varepsilon))$ *with accuracy,*

$$\|\mathcal{G} - \mathcal{G}_\theta\|_{L^2} \leq C(\varepsilon + C_{\text{stab}}^\varepsilon BN^{-r+d/p^*} + C_{\varepsilon,r}^{CB} N^{-r}). \tag{3.6}$$

*Proof.* We give a sketch of the proof, details can be found in **SM** B.3. Given function values of $v$ on a uniform grid with grid size $1/N$, we use trigonometric polynomial interpolation (**SM** A.6) to reconstruct $v$ and use this together with Assumption 3.1 to construct a neural network. The resulting approximation is then projected onto the space $L_N^2$, of trigonometric polynomials of degree at most $N \in \mathbb{N}$, again through trigonometric polynomial interpolation. $\square$

A recent result, [38, Theorem 36] (**SM** Lemma B.6), shows that any error bound for FNOs also implies an error bound for DeepONets, by choosing the trunk nets as neural network approximations of the Fourier basis. We apply this result with $\varepsilon \sim \mathrm{poly}(1/N)$ to Theorem 3.7 to obtain the following generic error bound for DeepONets.

**Corollary 3.8.** *Assume the setting of Theorem 3.7. Then for every $\varepsilon > 0$, $N \in \mathbb{N}$ and every corresponding FNO $\mathcal{G}_\theta$ from Theorem 3.7 there exists a DeepONet $\mathcal{G}_\theta^* : \mathcal{X} \to L^2(D)$ with* $\mathrm{width}(\boldsymbol{\beta}) = \mathcal{O}(N^d)$, $\mathrm{depth}(\boldsymbol{\beta}) = \mathcal{O}(\mathrm{depth}(\mathcal{G}_\theta))$, $\mathrm{width}(\boldsymbol{\tau}) = \mathcal{O}(N^{d+1})$ *and* $\mathrm{depth}(\boldsymbol{\tau}) \leq 3$ *that satisfies* (3.6).

### 3.3 Estimates for physics-informed operator learning

Using the techniques from previous sections, we now present the very first theoretical result for physics-informed operator learning. We demonstrate that if an error estimate for a DeepONet/FNO and the growth of its derivatives are known (see **SM** D.1 on how to obtain these), then one can prove an error estimate for the corresponding physics-informed DeepONet/FNO. For simplicity, the following result focuses only on operators mapping to $C^r(D)$ but the generalization to e.g. $C^r([0,T] \times D)$ is immediate by considering $D' := [0,T] \times D$.

**Theorem 3.9.** *Consider an operator $\mathcal{G} : \mathcal{X} \to C^r(D)$, $r \in \mathbb{N}$, that satisfies Assumption 3.3 and Assumption 3.4 with $\ell \in N$. Let $\lambda^* \in (0,\infty]$, let $\lambda, C(\lambda) > 0$ with $\lambda \leq \lambda^*$ and let $\sigma : \mathbb{N} \to \mathbb{R}$ be a function such that for all $p \in \mathbb{N}$ there is a DeepONet/FNO $\mathcal{G}_\theta$ such that*

$$\left\| \mathcal{G}(v) - \mathcal{G}_\theta(v) \right\|_{L^2(D)} \leq Cp^{-\lambda} \qquad \text{and} \qquad \left| \mathcal{G}_\theta(v) \right|_{C^r(D)} \leq Cp^{\sigma(r)} \quad \forall r \in \mathbb{N}, v \in \mathcal{X}. \quad (3.7)$$

*Then for all $\beta \in \mathbb{R}$ with $0 < \beta \leq \frac{(r-\ell)\lambda^* - \ell\sigma(r)}{r}$ there exists a constant $C^* > 0$ such that for all $v \in \mathcal{X}$ and $p \in \mathbb{N}$ it holds that*

$$\left\| \mathcal{L}(\mathcal{G}_\theta(v)) \right\|_{L^2(D)} \leq C^* p^{-\beta}. \quad (3.8)$$

*Proof.* For suitable $D^{\boldsymbol{\alpha}}$, use **SM** Lemma B.1 with $q = 2$, $f_1 = \mathcal{G}(v)$ and $f_2 = \mathcal{G}_\theta(v)$ together with (3.7) to find

$$\left\| D^{\boldsymbol{\alpha}}(\mathcal{G}(v) - \mathcal{G}_\theta(v)) \right\|_{L^2(D)} \leq C(r,\lambda)(p^{-\lambda}h^{-\ell} + p^{\sigma(r)}h^{r-\ell}). \quad (3.9)$$

Let $\beta \in \mathbb{R}$ with $0 < \beta \leq \frac{(r-\ell)\lambda^* - \ell\sigma(r)}{r}$. We carefully balance terms by setting $h = p^{-\frac{\sigma(r)+\beta}{r-\ell}}$ and $\lambda = \frac{\ell}{r-\ell}\sigma(r) + \frac{r}{r-\ell}\beta$ to find (3.8). Conclude using Assumption 3.4. $\square$

Finally, we use Theorem 3.5 to present an alternative error estimate for a physics-informed DeepONet in the case that Assumption 3.1 is satisfied. As this assumption is different from assuming access to an error bound for the corresponding DeepONet, it is interesting to use the techniques from the previous sections rather than directly apply Theorem 3.9. The proof of the following theorem can be found in **SM** B.4.

**Theorem 3.10.** *Let $s, r \in \mathbb{N}$, $T > 0$, let $\mathcal{G} : C^r(\mathbb{T}^d) \to C^{(s,r)}([0,T] \times \mathbb{T}^d)$ be an operator that maps a function $u_0$ to the solution $u$ of the PDE (2.1) with initial condition $u_0$, let Assumption 3.1 and Assumption 3.6 be satisfied and let $p^* \in \{2,\infty\} \setminus \{p\}$. There exists a constant $C > 0$ such that for every $Z, N, M \in \mathbb{N}$, $\varepsilon, \rho > 0$ there is an DeepONet $\mathcal{G}_\theta : C^r(\mathbb{T}^d) \to L^2([0,T] \times \mathbb{T}^d)$ with $Z^d$ sensors with accuracy,*

$$\left\| \mathcal{G}(v) - \mathcal{G}_\theta(v) \right\|_{L^2([0,T] \times \mathbb{T}^d)} \leq CM^\rho(\|u\|_{C^{(s,0)}}M^{-s} + M^{s-1}(\varepsilon + C_{\mathrm{stab}}^\varepsilon Z^{-r+d/p^*} + C_{\varepsilon,r}^{CB}N^{-r})) \quad (3.10)$$

*and if additionally Assumption 3.3 and Assumption 3.4 hold then,*

$$\left\| \mathcal{L}(\mathcal{G}_\theta(v)) \right\|_{L^2([0,T] \times \mathbb{T}^d)} \leq CM^{k+\rho}(\|u\|_{C^{(s,\ell)}}M^{-s} + M^{s-1}N^\ell(\varepsilon + C_{\mathrm{stab}}^\varepsilon Z^{-r+d/p^*} + C_{\varepsilon,r}^{CB}N^{-r})), \quad (3.11)$$

*for all $v$. Moreover, it holds that,* $\mathrm{depth}(\boldsymbol{\beta}) = \mathrm{depth}(\mathcal{U}^\varepsilon)$, $\mathrm{width}(\boldsymbol{\beta}) = \mathcal{O}(M(Z^d + N^d\mathrm{width}(\mathcal{U}^\varepsilon)))$, $\mathrm{depth}(\boldsymbol{\tau}) = 3$ *and* $\mathrm{width}(\boldsymbol{\tau}) = \mathcal{O}(MN^d(N + \ln(N)))$.

## 3.4 A posteriori bound on the generalization error

Although the main focus of this paper is on the approximation error for different neural network architectures, we now demonstrate that it is possible to provide similar bounds for other sources of error, such as the generalization error. We therefore prove a general *a posteriori* upper bound on the generalization error of the all the considered neural network architectures. Consider $f : \mathcal{D} \to \mathbb{R}$ (an operator or function) and the neural network architecture $f_\theta : \mathcal{D} \to \mathbb{R}, \theta \in \Theta$, which includes all architectures of Section 2.2: neural networks ($\mathcal{D} = \Omega$, $f = u$ and $f_\theta = u_\theta$), PINNs ($\mathcal{D} = \Omega$, $f = 0$ and $f_\theta = \mathcal{L}(u_\theta)$), operator learning ($\mathcal{D} = \Omega \times \mathcal{X}$, $f = \mathcal{G}$ and $f_\theta = \mathcal{G}_\theta$) and physics-informed operator learning ($\mathcal{D} = \Omega \times \mathcal{X}$, $f = 0$ and $f_\theta = \mathcal{L}(\mathcal{G}_\theta)$). Given a training set $\mathcal{S} = \{X_1, \ldots X_n\}$, where $\{X_i\}_{i=1}^n$ are iid random variables on $\mathcal{D}$ (according to a measure $\mu$), the training error $\mathcal{E}_T$ and generalization error $\mathcal{E}_G$ are,

$$\mathcal{E}_T(\theta, \mathcal{S})^2 = \frac{1}{n} \sum_{i=1}^n \big| f(z_i) - f_\theta(z_i) \big|^2, \qquad \mathcal{E}_G(\theta)^2 = \int_{\mathcal{D}} \big| f_\theta(z) - f(z) \big|^2 d\mu(z), \qquad (3.12)$$

where $\mu$ is a probability measure on $\mathcal{D}$. The following theorem provides a computable a posteriori error bound on the expectation of the generalization error for a general class of approximators. We refer to e.g. [6, 16] for bounds on $d_\Theta$, $c$ and $\mathfrak{L}$.

**Theorem 3.11.** *For $R > 0$ and $d_\Theta \in \mathbb{N}$, let $\Theta = [-R, R]^{d_\Theta}$ be the set of trainable parameters, and for every training set $\mathcal{S}$, let $\theta^*(\mathcal{S}) \in \Theta$ be an (approximate) minimizer of $\theta \mapsto \mathcal{E}_T(\theta, \mathcal{S})^2$, assume that $\theta \mapsto \mathcal{E}_G(\theta, \mathcal{S})^2$ and $\theta \mapsto \mathcal{E}_T(\theta)^2$ are bounded by $c > 0$ and Lipschitz continuous with Lipschitz constant $\mathfrak{L} > 0$. If $n \geq 2c^2 e^8 / (2R\mathfrak{L})^{d_\Theta/2}$ then it holds that*

$$\mathbb{E}\left[ \mathcal{E}_G(\theta^*(\mathcal{S}))^2 \right] \leq \mathbb{E}\left[ \mathcal{E}_T(\theta^*(\mathcal{S}), \mathcal{S})^2 \right] + \sqrt{\frac{2c^2(d_\Theta + 1)}{n} \ln\big(R\mathfrak{L}\sqrt{n}\big)}. \qquad (3.13)$$

*Proof.* The proof (**SM** B.5) combines standard techniques, based on covering numbers and Hoeffding's inequality, with an error composition from [16]. $\square$

For any type of neural network architecture of depth $L$, width $W$ and weights bounded by $R$, one finds that $d_\Theta \sim LW(W + d)$. For tanh neural networks and operator learning architectures, one has that $\ln(\mathfrak{L}) \sim L \ln(dRW)$, whereas for physics-informed neural networks and DeepONets one finds that $\ln(\mathfrak{L}) \sim (k + \ell)L \ln(dRW)$ with $k$ and $\ell$ as in Assumption 3.4 [43, 16]. Taking this into account, one also finds that the imposed lower bound on $n$ is not very restrictive. Moreover, the RHS of (3.13) depends at most polynomially on $L, W, R, d, k, \ell$ and $c$. For physics-informed architectures, however, upper bounds on $c$ often depend exponentially on $L$ [16, 14].

**Remark 3.12.** *As Theorem 3.11 is an a posteriori error estimate, one can use the network sizes of the trained networks for $L$, $W$ and $R$. The sizes stemming from the approximation error estimates of the previous sections can be disregarded for this result. Moreover, instead of considering the expected values of $\mathcal{E}_G$ and $\mathcal{E}_T$ in (3.13), one can also prove that such an inequality holds with a certain probability (see **SM** B.5).*

## 4 Applications

We demonstrate the power and generality of the framework proposed in Section 3 by applying the presented theory to the following case studies. First, we demonstrate how these generic bounds can be used to overcome the curse of dimensionality (CoD) for linear Kolmogorov PDEs and nonlinear parabolic PDEs (Section 4.1). These are the first available results that overcome the CoD for nonlinear parabolic PDEs for PINNs and (physics-informed) operator learning. Next, we apply the results of Section 3.3 to both linear and nonlinear operators and provide bounds on the approximation error for physics-informed operator learning.

### 4.1 Overcoming the curse of dimensionality

For high-dimensional PDEs, it is not possible to obtain efficient approximation results using standard neural network approximation theory [77, 15] as they will lead to convergence rates that suffer from the CoD, meaning that the neural network size scales exponentially in the input dimension.

In literature, one has shown for some PDEs that their solution at a fixed time can be approximated to accuracy $\varepsilon > 0$ with a network that has size $\mathcal{O}(\text{poly}(d)\varepsilon^{-\beta})$, with $\beta > 0$ independent of $d$, and therefore *overcomes the CoD*.

**Linear Kolmogorov PDEs**  We consider linear time-dependent PDEs of the following form.

**Setting 4.1.** *Let $s, r \in \mathbb{N}$, $u_0 \in C_0^2(\mathbb{R}^d)$ and let $u \in C^{(s,r)}([0,T] \times \mathbb{R}^d)$ be the solution of*

$$\mathcal{L}(u)(x,t) = \partial_t u(x,t) - \frac{1}{2}\text{Tr}(\sigma(x)\sigma(x)^T \Delta_x[u](x,t)) - \mu(x)^T \nabla_x[u](x,t) = 0, \quad u(0,x) = u_0(x) \tag{4.1}$$

*for all $(x,t) \in D \times [0,T]$, where $\sigma : \mathbb{R}^d \to \mathbb{R}^{d \times d}$ and $\mu : \mathbb{R}^d \to \mathbb{R}^d$ are affine functions and for which $\|u\|_{C^{(s,2)}}$ grows at most polynomially in $d$. For every $\varepsilon > 0$, there is a neural network $\widehat{u}_0$ of width $\mathcal{O}(\text{poly}(d)\varepsilon^{-\beta})$ such that $\|u_0 - \widehat{u}_0\|_{L^\infty(\mathbb{R}^d)} < \varepsilon$.*

Prototypical examples of such *linear Kolmogorov PDEs* include the heat equation and the Black-Scholes equation. In [23, 7, 36] the authors construct a neural network that approximates $u(T)$ and overcomes the CoD by emulating Monte-Carlo methods based on the Feynman-Kac formula. In [16] one has proven that PINNs overcome the CoD as well, in the sense that the network size grows as $\mathcal{O}(\text{poly}(d\rho_d)\varepsilon^{-\beta})$, with $\rho_d$ as defined in **SM** (C.10). For a subclass of Kolmogorov PDEs it is known that $\rho_d = \text{poly}(d)$, such that the CoD is fully overcome.

We demonstrate that the generic bounds of Section 3 (Theorem 3.5) can be used to provide a much shorter proof for this result. **SM** Lemma C.6 verifies that Assumption 3.1 is indeed satisfied. The full proof can be found in **SM** C.2.

**Theorem 4.2.** *Assume that Setting 4.1 holds. For every $\sigma, \varepsilon > 0$ and $d \in \mathbb{N}$, there is a tanh neural network $u_\theta$ of depth $\mathcal{O}(\text{depth}(\widehat{u}_0))$ and width $\mathcal{O}(\text{poly}(d\rho_d)\varepsilon^{-(2+\beta)\frac{r+\sigma}{r-2}\frac{s+1}{s-1} - \frac{1+\sigma}{s-1}})$ such that,*

$$\left\|\mathcal{L}(u_\theta)\right\|_{L^2([0,T]\times[0,1]^d)} + \|u_\theta - u\|_{L^2(\partial([0,T]\times[0,1]^d))} \leq \varepsilon. \tag{4.2}$$

**Nonlinear parabolic PDEs**  Next, we consider nonlinear parabolic PDEs as in Section 4.3, which typically arise in the context of nonlinear diffusion-reaction equations that describe the change in space and time of some quantities, such as in the well-known *Allen-Cahn equation* [1].

**Setting 4.3.** *Let $s, r \in \mathbb{N}$ and for $u_0 \in \mathcal{X} \subset C^r(\mathbb{T}^d)$ let $u \in C^{(s,r)}([0,T] \times \mathbb{T}^d)$ be the solution of*

$$\mathcal{L}(u)(x,t) = \partial_t u(t,x) - \Delta_x u(t,x) - F(u(t,x)) = 0, \qquad u(0,x) = u_0(x), \tag{4.3}$$

*for all $(t,x) \in [0,T] \times D$, with period boundary conditions, where $F : \mathbb{R} \to \mathbb{R}$ is a polynomial and for which $\|u\|_{C^{(s,2)}}$ grows at most polynomially in $d$. For every $\varepsilon > 0$, there is a neural network $\widehat{u}_0$ of width $\mathcal{O}(\text{poly}(d)\varepsilon^{-\beta})$ such that $\|u_0 - \widehat{u}_0\|_{L^\infty(\mathbb{T}^d)} < \varepsilon$. Let $\mu$, resp. $\mu^*$, be the normalized Lebesgue measure on $[0,T] \times \mathbb{T}^d$, resp. $\partial([0,T] \times \mathbb{T}^d)$.*

In [32] the authors have proven that ReLU neural networks overcome the CoD in the approximation of $u(T)$. We have reproven this result in **SM** Lemma C.14 for tanh neural networks to show that Assumption 3.1 is satisfied. Using Theorem 3.5 we can now prove that PINNs overcome the CoD for nonlinear parabolic PDEs. The proof is analogous to that of Theorem 4.2.

**Theorem 4.4.** *Assume Setting 4.3. For every $\sigma, \varepsilon > 0$ and $d \in \mathbb{N}$ there is a tanh neural network $u_\theta$ of depth $\mathcal{O}(\text{depth}(\widehat{u}_0) + \text{poly}(d)\ln(1/\varepsilon))$ and width $\mathcal{O}(\text{poly}(d)\varepsilon^{-(2+\beta)\frac{r+\sigma}{r-2}\frac{s+1}{s-1} - \frac{1+\sigma}{s-1}})$ such that,*

$$\left\|\mathcal{L}(u_\theta)\right\|_{L^2([0,T]\times\mathbb{T}^d,\mu)} + \|u - u_\theta\|_{L^2(\partial([0,T]\times\mathbb{T}^d,\mu^*))} \leq \varepsilon. \tag{4.4}$$

Similarly, one can use the results from Section 3.2 to obtain estimates for (physics-informed) DeepONets for nonlinear parabolic PDEs (4.3) such as the Allen-Cahn equation. In particular, a dimension-independent convergence rate can be obtained if the solution is smooth enough, which improves upon the result of [43], which incurred the CoD. For simplicity, we present results for $C^{(2,r)}$ functions, rather than $C^{(s,r)}$ functions, as we found that assuming more regularity did not necessarily further improve the convergence rate. The proof is given in **SM** B.4.

**Theorem 4.5.** *Assume Setting 4.3 and let $\mathcal{G} : \mathcal{X} \to C^r(\mathbb{T}^d) : u_0 \mapsto u(T)$ and $\mathcal{G}^* : \mathcal{X} \to C^{(2,r)}([0,T] \times \mathbb{T}^d) : u_0 \mapsto u$. For every $\sigma, \varepsilon > 0$, there exists a DeepONets $\mathcal{G}_\theta$ and $\mathcal{G}_\theta^*$ such that*

$$\|\mathcal{G} - \mathcal{G}_\theta\|_{L^2(\mathbb{T}^d \times \mathcal{X})} \leq \varepsilon, \qquad \left\|\mathcal{L}(\mathcal{G}_\theta^*)\right\|_{L^2([0,T]\times\mathbb{T}^d\times\mathcal{X})} \leq \varepsilon. \tag{4.5}$$

*Moreover, for $\mathcal{G}_\theta$ we have $\mathcal{O}(\varepsilon^{-\frac{d+\sigma}{r}})$ sensors and,*

$$
\begin{aligned}
\text{width}(\boldsymbol{\beta}) &= \mathcal{O}(\varepsilon^{-\frac{(d+\sigma)(2+\beta)}{r}}), & \text{depth}(\boldsymbol{\beta}) &= \mathcal{O}(\ln(1/\varepsilon)), \\
\text{width}(\boldsymbol{\tau}) &= \mathcal{O}(\varepsilon^{-\frac{d+1+\sigma}{r}}), & \text{depth}(\boldsymbol{\tau}) &= 3,
\end{aligned}
\tag{4.6}
$$

*whereas for $\mathcal{G}_\theta^*$ we have $\mathcal{O}(\varepsilon^{-\frac{(3+\sigma)d}{r-2}})$ sensors and,*

$$
\begin{aligned}
\text{width}(\boldsymbol{\beta}) &= \mathcal{O}(\varepsilon^{-1-\frac{(3+\sigma)(d+r(2+\beta))}{r-2}}), & \text{depth}(\boldsymbol{\beta}) &= \mathcal{O}(\ln(1/\varepsilon)), \\
\text{width}(\boldsymbol{\tau}) &= \mathcal{O}(\varepsilon^{-1-\frac{(3+\sigma)(d+1)}{r-2}}), & \text{depth}(\boldsymbol{\tau}) &= 3.
\end{aligned}
\tag{4.7}
$$

### 4.2 Error bounds for physics-informed operator learning

We demonstrate how Theorem 3.9 can be used to generalize available error estimates for DeepONets and FNOs, e.g. [43, 38] and **SM** D.1, to estimates for their physics-informed counterparts.

**Linear operators**    In the simplest case, the operator $\mathcal{G}$ of interest is linear. In [43, Theorem D.2], a general error bound for ReLU DeepONets for linear operators has been established, which still holds for tanh DeepONets. Using Theorem 3.9 it is then straightforward to prove convergence rates for physics-informed DeepONets for solution operators of linear PDEs (2.1).

Consider an operator $\mathcal{G} : \mathcal{X} \to L^2(\mathbb{T}^d) : v \mapsto u$ as in Section 2.1, where $v$ is the parameter/initial condition and $u$ the solution of the PDE (2.1). Following [43], we fix the measure $\mu$ on $L^2(\mathbb{T}^d)$ as a Gaussian random field, such that $v$ allows the Karhunen-Loève expansion $v = \sum_{k \in \mathbb{Z}^d} \alpha_k X_k \mathbf{e}_k$, where $|\alpha_k| \le \exp(-\ell|k|)$ with $\ell > 0$, the $X_k \sim \mathcal{N}(0,1)$ are iid Gaussian random variables and $\{\mathbf{e}_k\}_{k \in \mathbb{Z}^d}$ is the standard Fourier basis (**SM** A.5). In this setting, we can prove the following approximation result, the proof of which can be found in **SM** D.3. The result can be generalized to other data distributions $\mu$ for which a convergence result for DeepONets can be proven, as in [43].

**Theorem 4.6.** *Assume the setting above and that of Assumption 3.4, and assume that $\mathcal{G}(v) \in C^{\ell+1}(\mathbb{T}^d)$ for all $v \in \mathcal{X}$. For all $\beta > 0$ there exists a constant $C > 0$ such that for any $p \in \mathbb{N}$ there exists a DeepONet $\mathcal{G}_\theta$ with $p$ sensors and branch and trunk nets such that*

$$
\big\|\mathcal{L}(\mathcal{G}_\theta))\big\|_{L^2(L^2(\mathbb{T}^d),\mu)} \le Cp^{-\beta}.
\tag{4.8}
$$

*Moreover,* $\text{size}(\boldsymbol{\tau}) \le Cp^{\frac{d+1}{d}}$, $\text{depth}(\boldsymbol{\tau}) = 3$, $\text{size}(\boldsymbol{\beta}) \le p$ *and* $\text{depth}(\boldsymbol{\beta}) = 1$.

**Nonlinear operators**    For nonlinear PDEs a general result like Theorem 4.6 can not be obtained from the currently available tools. Instead one needs to use Theorem 3.9 for every PDE of interest on a case-by-case basis. In the **SM**, we demonstrate this for a nonlinear ODE (gravity pendulum with external force, **SM** D.5) and an elliptic PDE (Darcy flow, **SM** D.6).

## 5   Related work and discussion

This is the first paper to rigorously expose the connections between the different deep learning frameworks from Section 2.2 for generic PDEs. Until now, most available results focus on providing generic results for one specific method. In [31] and [24] one uses neural networks that approximate solutions to a generic ODE/PDE at a fixed time to construct space-time neural networks. A generalization to PINNs is not immediate as the proof involves the emulation of the forward Euler method. We have overcome this difficulty by constructing space-time neural networks using Taylor expansions instead (Theorem 3.5). To bound the approximation error of PINNs one can use the generic error bounds in Sobolev norms of e.g. [25, 26] for very general activation functions or the more concrete bounds [15] for tanh neural networks. In both approaches, the only assumption is that the solution of the PDE has sufficient Sobolev regularity. As a consequence, these results incur the curse of dimensionality and are not applicable to high-dimensional PDEs. The authors of [15] analyze PINNs based on three theoretical questions related to approximation, stability and generalization. Other theoretical analyses of PINNs include e.g. [71, 72, 30]. For DeepONets, convergence rates for advection-diffusion equations are presented in [17] and a clear workflow for obtaining generic error estimates as well as worked out examples can be found in [43]. Similar results are obtained for FNOs in [38]. A comprehensive comparison of DeepONets and FNOs is the topic of [50]. To the best of

the authors' knowledge, no theoretical results for physics-informed operator learning are currently available. Unrelated to the approximation error, we also report generic bounds on the expected value of the generalization error of all the aforementioned deep learning architectures, in the form of an a posteriori error estimate on the generalization error.

A second goal of the paper is to prove that deep learning-based frameworks can overcome the curse of dimensionality (CoD). PDEs for which the curse of dimensionality has been overcome include linear Kolmogorov PDEs e.g. [23, 36], nonlinear parabolic PDEs [32] and elliptic PDEs [4, 10, 57]. By assuming that the initial data lies in a Barron class, the authors of [52] proved for elliptic PDEs that the Deep Ritz Method [20] can overcome the CoD. Since the Barron class is a Banach algebra [10] it is possible that our results, which mostly only involve multiplications and additions of neural networks, can be extended to Barron functions. For PINNs, it is proven that they can overcome the CoD for linear Kolmogorov PDEs [16]. We give an alternative proof of this result, improve the convergence rate (Theorem 4.2) and additionally prove that PINNs can also overcome the CoD for nonlinear parabolic PDEs (Theorem 4.4). DeepONets and FNOs can overcome the CoD in many cases [43, 38] but we note that this does not yet include nonlinear parabolic PDEs such as the Allen-Cahn equation. In Theorem 4.5 we prove that dimension-independent convergence rates can be obtained if the solution is sufficiently regular. Similar results are expected to hold for e.g. elliptic PDEs by using the results from [4, 10, 57].

It is evident that the generic bounds presented here can only be obtained under suitable assumptions. These should always be checked to prevent misleading claims about mathematical guarantees for the considered deep learning methods. We briefly discuss how restrictive these are and whether they can be relaxed. Assuming the existence of a neural network that approximates the solution of PDE at a fixed time (Assumption 3.1) is of course essential, but such a result can usually be obtained by emulating an existing numerical method. Proving a bound on the Sobolev norm of that network is always possible as we only consider smooth networks. Assumption 3.3 holds for many domains, including rectangular and smooth ones. Assumption 3.4 and Assumption 3.6 also hold for a very broad class of PDEs, much like the assumption on the size of the neural network approximation in Setting 4.1 and 4.3 holds for most functions of interest. Therefore, the assumption that the PDE solution is $C^{(s,r)}$-regular seems to be the most restrictive. However, results like Theorem 3.5 could be extended to e.g. Sobolev regular functions by using the Bramble-Hilbert lemma instead of Taylor expansions. Another restriction is that we exclusively focused on neural networks with the tanh activation function. This was only for simplicity of exposition. All results still hold for other sigmoidal activation functions, as well as more general smooth activation functions, which might give rise to slightly different convergence rates. A last restriction is that the obtained rates are not optimal, but this is not the goal of our framework. In particular, for PINNs for low-dimensional PDEs it is beneficial to use e.g. [26, 15].

Optimizing the obtained convergence rates and comparing with optimal ones is one direction for future research. Previously mentioned possibilities include extending to more general activation functions and less regular functions. Another direction is to make the connection between our results and that of [10] where they prove that Barron spaces are Banach algebras and use this to obtain dimension-independent convergence rates for PDEs with initial data in a Barron class by emulating numerical methods.

Here, we have considered the approximation and generalization errors in the present analysis. It is clear that the bounds on the generalization error may not be sharp, as in traditional deep learning. Obtaining sharper bounds will be an interesting topic for further investigation. Finally, there is no explicit bound on the training (optimization) errors. Obtaining such bounds will be considered in the future.

## Acknowledgments and Disclosure of Funding

The research of SM was performed under a project that has received funding from the European Research Council (ERC) under the European Union's Horizon 2020 research and innovation programme (Grant Agreement No. 770880).

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
