# A Notation and preliminaries

We introduce notation and preliminary results regarding finite differences, Sobolev spaces, the Legendre basis, the Fourier basis, trigonometric polynomial interpolation and neural network approximation theory.

## A.1 Overview of used notation

Table 1: Glossary of used notation.

| Symbol | Description | Page |
|--------|-------------|------|
| $\sigma$ | tanh activation function | |
| $d$ | spatial dimension of domain | |
| $\mathbb{T}^d$ | periodic torus, identified with $[0, 2\pi)^d$ | |
| $D$ | general $d$-dimensional spatial domain | p. 2 |
| $\Omega$ | general domain, either $\Omega = D$ or $\Omega = [0, T] \times D$ | p. 2 |
| $\partial\Omega$ | boundary of $\Omega$ | |
| $\mathcal{H}$ | function space of PDE solution | p. 2 |
| $\mathcal{Y}$ | function space of parameters for $\mathcal{L}$, e.g. $\mathcal{L}_a$ with $a \in \mathcal{Y}$ | p. 2 |
| $\mathcal{Z}$ | function space of initial conditions for the PDE (2.1) | p. 2 |
| $\mathcal{X}$ | input function space of the operator $\mathcal{G}$ | p. 2 |
| $\mathcal{G}$ | operator of interest, $\mathcal{G} : \mathcal{X} \to L^2(\Omega)$ | p. 2 |
| $\mathcal{L}, \mathcal{L}_a$ | differential operator that describes the PDE (with parameter $a$) | p. 2 |
| $r, s$ | regularity of the PDE solution, $u \in C^r(D)$ or $u \in C^{(s,r)}([0, T] \times D)$ | p. 3 |
| $D^{(k,\boldsymbol{\alpha})}$ | $D^{(k,\boldsymbol{\alpha})} := D_t^k D_x^{\boldsymbol{\alpha}} := \partial_t^k \partial_{x_1}^{\alpha_1} \ldots \partial_{x_d}^{\alpha_d}$, for $(k, \boldsymbol{\alpha}) \in \mathbb{N}_0^{d+1}$ | p. 5 |
| $\ell$ | upper bound on $\|\boldsymbol{\alpha}\|_1$ | p. 5 |
| $q$ | see Assumption 3.1 | p. 4 |
| $p$ | see Assumption 3.6 | p. 5 |
| $\mathrm{poly}(d)$ | a polynomial in $d$ | p. 5 |
| $C_0^r$ | subset of $C^r$ functions with compact support | |
| $\Delta_h^{\boldsymbol{\alpha},r}$ | finite difference operator; if the variable is time: $\Delta_{h,t}^{\boldsymbol{\alpha},s}$ | p. 15 |
| $\mathcal{J}_N$ | grid point indices, $\mathcal{J}_N = \{0, \ldots, 2N\}^d$ | p. 16 |
| $\mathcal{K}_N$ | Fourier wavenumbers $\mathcal{K}_N = \{k \in \mathbb{Z}^d \mid |k|_\infty \leq N\}$ | |
| $L^2$ | Space of square-integrable functions | |
| $H^s$ | Sobolev space of smoothness $s$, with norm $\|\cdot\|_{H^s}$ | p. 16 |
| $L_N^2$ | $L_N^2 \subset L^2$ trigonometric polynomials of degree $\leq N$ | p. 17 |

## A.2 Finite differences

For $h > 0$, $\boldsymbol{\alpha} \in \mathbb{N}_0^d$, $r \in \mathbb{N}$ and $\ell := \|\boldsymbol{\alpha}\|_1$, we define a finite difference operator $\Delta_h^{\boldsymbol{\alpha},r}$ as,

$$\Delta_h^{\boldsymbol{\alpha},r}[f](t,x) = \sum_j c_j^{\boldsymbol{\alpha},r} f(t, x + h b_j^{\boldsymbol{\alpha},r}), \tag{A.1}$$

for $f \in C^{r+\ell}(\mathbb{R}^d)$, where the number of non-zero terms in the summation can be chosen to be finite and only dependent on $\ell$ and $r$ and where the choice of $b_j^{\boldsymbol{\alpha},r} \in \mathbb{R}^d$ allows to approximate $D_x^{\boldsymbol{\alpha}} f$ up to accuracy $\mathcal{O}(h^r)$. This means that for any $f \in C^{r+\ell}(\mathbb{R}^r)$ it holds for all $x$ that,

$$\left| h^{-\ell} \cdot \Delta_h^{\boldsymbol{\alpha},r}[f](t,x) - D_x^{\boldsymbol{\alpha}} f(t,x) \right| \leq c_{\ell,r} \left| f(t, \cdot) \right|_{C^{r+\ell}} h^r \quad \text{for } h > 0, \tag{A.2}$$

where $c_{\ell,r} > 0$ does not depend on $f$ and $h$. Similarly, we can define a finite difference operator $\Delta_{h,t}^{k,s}[f](t,x)$ to approximate $D_t^k f(t,x)$ to accuracy $\mathcal{O}(h^s)$.

## A.3 Sobolev spaces

Let $d \in \mathbb{N}$, $k \in \mathbb{N}_0$, $1 \leq p \leq \infty$ and let $\Omega \subseteq \mathbb{R}^d$ be open. For a function $f : \Omega \to \mathbb{R}$ and a (multi-)index $\boldsymbol{\alpha} \in \mathbb{N}_0^d$ we denote by

$$D^{\boldsymbol{\alpha}} f = \frac{\partial^{|\boldsymbol{\alpha}|} f}{\partial x_1^{\alpha_1} \cdots \partial x_d^{\alpha_d}} \tag{A.3}$$

the classical or distributional (i.e. weak) derivative of $f$. We denote by $L^p(\Omega)$ the usual Lebesgue space and for we define the Sobolev space $W^{k,p}(\Omega)$ as

$$W^{k,p}(\Omega) = \{ f \in L^p(\Omega) : D^{\boldsymbol{\alpha}} f \in L^p(\Omega) \text{ for all } \boldsymbol{\alpha} \in \mathbb{N}_0^d \text{ with } |\boldsymbol{\alpha}| \leq k \}. \tag{A.4}$$

For $p < \infty$, we define the following seminorms on $W^{k,p}(\Omega)$,

$$|f|_{W^{m,p}(\Omega)} = \left( \sum_{|\boldsymbol{\alpha}|=m} \|D^{\boldsymbol{\alpha}} f\|_{L^p(\Omega)}^p \right)^{1/p} \qquad \text{for } m = 0, \ldots, k, \tag{A.5}$$

and for $p = \infty$ we define

$$|f|_{W^{m,\infty}(\Omega)} = \max_{|\boldsymbol{\alpha}|=m} \|D^{\boldsymbol{\alpha}} f\|_{L^\infty(\Omega)} \qquad \text{for } m = 0, \ldots, k. \tag{A.6}$$

Based on these seminorms, we can define the following norm for $p < \infty$,

$$\|f\|_{W^{k,p}(\Omega)} = \left( \sum_{m=0}^{k} |f|_{W^{m,p}(\Omega)}^p \right)^{1/p}, \tag{A.7}$$

and for $p = \infty$ we define the norm

$$\|f\|_{W^{k,\infty}(\Omega)} = \max_{0 \leq m \leq k} |f|_{W^{m,\infty}(\Omega)}. \tag{A.8}$$

The space $W^{k,p}(\Omega)$ equipped with the norm $\|\cdot\|_{W^{k,p}(\Omega)}$ is a Banach space.

We denote by $C^k(\Omega)$ the space of functions that are $k$ times continuously differentiable and equip this space with the norm $\|f\|_{C^k(\Omega)} = \|f\|_{W^{k,\infty}(\Omega)}$.

**Lemma A.1** (Continuous Sobolev embedding). *Let $d, \ell \in \mathbb{N}$ and let $k \geq d/2 + \ell$. Then there exists a constant $C > 0$ such that for any $f \in H^k(\mathbb{T}^d)$ it holds that*

$$\|f\|_{C^\ell(\mathbb{T}^d)} \leq C \|f\|_{H^k(\mathbb{T}^d)}. \tag{A.9}$$

## A.4 Notation for Legendre basis

In a one-dimensional setting, we denote for $j \in \mathbb{N}_0$ the $j$-th Legendre polynomial by $L_j$. Following the notation of [64], it holds that $L_j(x) = \sum_{j=0}^{\ell} c_\ell^j x^\ell$ where, with $m(\ell) := (j - \ell)/2$,

$$c_\ell^j = \begin{cases} 0 & j - \ell \{0, \ldots, j\} \cup (2\mathbb{Z} + 1), \\ (-1)^m 2^{-j} \binom{j}{m} \binom{j+\ell}{j} \sqrt{2j+1} & j - \ell \{0, \ldots, j\} \cup 2\mathbb{Z}, \end{cases} \tag{A.10}$$

where each polynomial is normalized in $L^2([-1, 1], \lambda/2)$, where $\lambda$ is the Lebesgue measure. Similarly, the tensorized Legendre polynomials,

$$L_{\boldsymbol{\nu}}(x) = \prod_{j=1}^{d} L_{\nu_j}(x_j) \qquad \text{for all } \boldsymbol{\nu} \in \mathbb{N}_0^d, \tag{A.11}$$

constitute an orthonormal basis of $L^2([-1, 1]^d, \lambda/2^d)$. By considering the lexicographic order on $\mathbb{N}_0^d$, of which we denote the enumeration by $\kappa : \mathbb{N} \to \mathbb{N}_0^d$, one can defined an ordered basis $(L_j)_{j \in \mathbb{N}}$ by setting $L_j := L_{\kappa(j)}$.

From [64, eq. (2.19)] it also follows that,

$$\forall s \in \mathbb{N}_0, \boldsymbol{\nu} \in \mathbb{N}_0^d : \|L_{\boldsymbol{\nu}}\|_{C^s([-1,1]^d)} \leq \prod_{j=1}^{d} (1 + 2\nu_j)^{1/2+2s}. \tag{A.12}$$

## A.5 Notation for Standard Fourier basis

Using the notation from [43], we introduce the following "standard" real Fourier basis $\{\mathbf{e}_\kappa\}_{\kappa \in \mathbb{Z}^d}$ in $d$ dimensions. For $\kappa = (\kappa_1, \ldots, \kappa_d) \in \mathbb{Z}^d$, we let $\sigma(\kappa)$ be the sign of the first non-zero component of $\kappa$ and we define

$$\mathbf{e}_\kappa := C_\kappa \begin{cases} 1, & \sigma(\kappa) = 0, \\ \cos(\langle \kappa, x \rangle), & \sigma(\kappa) = 1, \\ \sin(\langle \kappa, x \rangle), & \sigma(\kappa) = -1, \end{cases} \tag{A.13}$$

where the factor $C_\kappa > 0$ ensures that $\mathbf{e}_\kappa$ is properly normalized, i.e. that $\|\mathbf{e}_\kappa\|_{L^2(\mathbb{T}^d)} = 1$. Next, let $\kappa : \mathbb{N} \to \mathbb{Z}^d$ be a fixed enumeration of $\mathbb{Z}^d$, with the property that $j \mapsto |\kappa(j)|_\infty$ is monotonically increasing, i.e. such that $j \leq j'$ implies that $|\kappa(j)|_\infty \leq |\kappa(j')|_\infty$. This will allow us to introduce an $\mathbb{N}$-indexed version of the Fourier basis,

$$\mathbf{e}_j(x) := \mathbf{e}_{\kappa(j)}(x), \quad \forall j \in \mathbb{N}. \tag{A.14}$$

Finally we note that

$$\|\mathbf{e}_\kappa\|_{C^s([0,2\pi]^d)} \leq \|\kappa\|_\infty^s. \tag{A.15}$$

## A.6 Trigonometric polynomial interpolation

For $N \in \mathbb{N}$, let $x_j = \frac{2\pi j}{2N+1}$ and let $y_j \in \mathbb{R}$ for all $j \in \mathcal{J}_N = \{0, \ldots, 2N+1\}^d$. We will construct an operator

$$\mathcal{Q}_N : \mathbb{R}^{|\mathcal{J}_N|} \to L^2(\mathbb{T}^d) : y \mapsto \mathcal{Q}_N(y), \tag{A.16}$$

where $\mathcal{Q}_N(y)$ is a trigonometric polynomial of degree at most $N$ such that $\mathcal{Q}_N(y)(x_j) = y_j$ for all $j \in \mathcal{J}_N$. We construct this polynomial using the discrete Fourier transform and its inverse. For $k \in \mathcal{K}_N = \{-N, \ldots, N\}^d$, we define the discrete Fourier transform as,

$$X_k(y) = \sum_{j \in \mathcal{J}_N} y_j \exp(-i\langle k, x_j \rangle), \tag{A.17}$$

and the trigonometric interpolation polynomial as,

$$\begin{aligned} \mathcal{Q}_N(y)(z) &= \frac{1}{|\mathcal{K}_N|} \sum_{k \in \mathcal{K}_N} X_k(y) \exp(i\langle k, z \rangle) \\ &= \frac{1}{|\mathcal{K}_N|} \sum_{k \in \mathcal{K}_N} \sum_{j \in \mathcal{J}_N} y_j \cos(\langle k, z - x_j \rangle) \\ &= \frac{1}{|\mathcal{K}_N|} \sum_{k \in \mathcal{K}_N} \sum_{j \in \mathcal{J}_N} y_j \Big( \cos(\langle k, x_j \rangle) \cos(\langle k, z \rangle) - \sin(\langle k, x_j \rangle) \sin(\langle k, z \rangle) \Big) \\ &= \frac{1}{|\mathcal{K}_N|} \sum_{k \in \mathcal{K}_N} \sum_{j \in \mathcal{J}_N} y_j a_{k,j} \mathbf{e}_k(z), \end{aligned} \tag{A.18}$$

where,

$$a_{k,j} = \begin{cases} 1, & \sigma(k) = 0, \\ \cos(\langle k, x_j \rangle), & \sigma(k) = 1, \\ \sin(\langle k, x_j \rangle), & \sigma(k) = -1, \end{cases} \tag{A.19}$$

with $\sigma$ as in **SM** A.5. We can also define an encoder $\mathcal{E}_N$ by,

$$\mathcal{E}_N : C(\mathbb{T}^d) \to \mathbb{R}^{|\mathcal{J}_N|} : f \mapsto (f(x_j))_{j \in \mathcal{J}_N}. \tag{A.20}$$

The composition $\mathcal{Q}_N \circ \mathcal{E}_N$ is called the pseudo-spectral projection onto the space of trigonometric polynomials of degree at most $N$ and has the following property [38].

**Lemma A.2.** *For $s, k \in \mathbb{N}_0$ with $s > d/2$ and $s \geq k$, and $f \in C^s(\mathbb{T}^d)$ it holds that*

$$\big\| f - (\mathcal{Q}_N \circ \mathcal{E}_N)(f) \big\|_{H^k(\mathbb{T}^d)} \leq C(s,d) N^{-(s-k)} \|f\|_{H^s(\mathbb{T}^d)}, \tag{A.21}$$

*for a constant $C(s,d) > 0$ that only depends on $s$ and $d$.*

### A.7 Neural network approximation theory

We recall some basic results on the approximation of functions by tanh neural networks in this section. All results are adaptations from results in [15]. The following two lemmas address the approximation of univariate monomials and the multiplication operator.

**Lemma A.3** (Approximation of univariate monomials, Lemma 3.2 in [15])**.** *Let $k \in \mathbb{N}_0, s \in 2\mathbb{N} - 1$, $M > 0$ and define $f_p : [-M, M] \to \mathbb{R} : x \mapsto x^p$ for all $p \in \mathbb{N}$. For every $\varepsilon > 0$, there exists a shallow tanh neural network $\psi_{s,\varepsilon} : [-M, M] \to \mathbb{R}^s$ of width $\frac{3(s+1)}{2}$ such that*

$$\max_{p \le s} \left\| f_p - (\psi_{s,\varepsilon})_p \right\|_{W^{k,\infty}} \le \varepsilon. \tag{A.22}$$

**Lemma A.4** (Shallow approximation of multiplication of $d$ numbers, Corollary 3.7 in [15])**.** *Let $d \in \mathbb{N}$, $k \in \mathbb{N}_0$ and $M > 0$. Then for every $\varepsilon > 0$, there exist a shallow tanh neural network $\widehat{\times}_d^\varepsilon : [-M, M]^d \to \mathbb{R}$ of width $3 \left\lceil \frac{d+1}{2} \right\rceil |P_{d,d}|$ (or 4 if $d = 2$) such that*

$$\left\| \widehat{\times}_d^\varepsilon(x) - \prod_{i=1}^d x_i \right\|_{W^{k,\infty}} \le \varepsilon. \tag{A.23}$$

## B  Additional material for Section 3

### B.1  Auxiliary results for Section 3

**Lemma B.1.** *Let $q \in [1, \infty]$, $r, \ell \in \mathbb{N}$ with $\ell \le r$ and $f_1, f_2 \in C^{(0,r)}([0, T] \times D)$. If Assumption 3.3 holds then there exists a constant $C(r) > 0$ such that for any $\boldsymbol{\alpha} \in \mathbb{N}_0^d$ with $\ell := \|\boldsymbol{\alpha}\|_1$ it holds that*

$$\left\| D_x^{\boldsymbol{\alpha}}(f_1 - f_2) \right\|_{L^q} \le C(\|f_1 - f_2\|_{L^q} h^{-\ell} + \max_{j=1,2} |f_j|_{C^{(0,r)}} h^{r-\ell}) \qquad \forall h > 0. \tag{B.1}$$

*Proof.* From the triangle inequality and (A.2) the existence of a constant $C(r) > 0$ follows such that,

$$\left\| D_x^{\boldsymbol{\alpha}}(f_1 - f_2) \right\|_{L^q} \le \max_{j=1,2} \left\| D^{\boldsymbol{\alpha}} f_j - h^{-\ell} \cdot \Delta_h^{\boldsymbol{\alpha},r}[f_j] \right\|_{L^q} + C(r) h^{-\ell} \|f_1 - f_2\|_{L^q}$$

$$\le c_{\ell,r} \max_{j=1,2} |f_j|_{C^{(0,r)}} h^{r-\ell} + C(r) h^{-\ell} \|f_1 - f_2\|_{L^q}. \qquad \square$$

**Lemma B.2.** *Using the notation of the proof of Theorem 3.5 (SM B.2), it holds that*

$$\left\| D^{(k,\boldsymbol{\alpha})}(\widetilde{u} - \widehat{u}) \right\|_{C^0} \le \delta. \tag{B.2}$$

*Proof.* Using the Faà di Bruno formula [12] and its consequences for estimating the norms of derivatives of compositions [15, Lemma A.7] one can prove for sufficiently regular functions $g_1, g_2, h_1, h_2$ and a suitable multi-index $\boldsymbol{\beta}$ estimates of the form,

$$\left\| D^{\boldsymbol{\beta}}(g_1 \circ h_1 - g_2 \circ h_2) \right\|_{C^0} \le C(\|g_1 - g_2\|_{C^{\|\boldsymbol{\beta}\|_1}} + \|h_1 - h_2\|_{C^{\|\boldsymbol{\beta}\|_1}}), \tag{B.3}$$

assuming that the compositions are well-defined and where the constant $C > 0$ may depend on $g_1, g_2, h_1, h_2$ and their derivatives. Using this theorem we can prove that

$$\left\| D^{(k,\boldsymbol{\alpha})}\widehat{u} - D^{(k,\boldsymbol{\alpha})} \sum_{m=1}^M \sum_{i=0}^{s-1} \frac{\Delta_{1/M,t}^{i,s-i}[\widehat{u}_m^\varepsilon](t_m, x)}{M^{-i} i!} \cdot \widehat{\varphi}_i^\delta(t - t_m) \cdot \Phi_m^M(t) \right\| < C\delta. \tag{B.4}$$

Because the size of the neural network $\widehat{\times}_\delta$ in the definition of $\widehat{u}$ does not depend on its accuracy $\delta$ (see Lemma A.4) we can rescale $\delta$ and therefore set $C = 1/2$ in the above inequality.

Next, we observe that,

$$D^{(k,\boldsymbol{\alpha})} \sum_{m=1}^M \sum_{i=0}^{s-1} \frac{\Delta_{1/M,t}^{i,s-i}[\widehat{u}_m^\varepsilon](t_m, x)}{M^{-i} i!} \cdot (\widehat{\varphi}_i^\delta - \varphi_i)(t - t_m) \cdot \Phi_m^M(t)$$

$$= \sum_{m=1}^M \sum_{i=0}^{s-1} \frac{\Delta_{1/M,t}^{i,s-i}[D_x^{\boldsymbol{\alpha}}\widehat{u}_m^\varepsilon](t_m, x)}{M^{-i} i!} \cdot \sum_{n=0}^k \binom{k}{n} \partial_t^n(\widehat{\varphi}_i^\delta - \varphi_i)(t - t_m) \cdot \partial_t^{k-n} \Phi_m^M(t) \tag{B.5}$$

Analogously to before, because the sizes of the neural networks $\widehat{\varphi}_i^\delta$ are independent of their accuracy $\delta$ we can rescale $\delta$ such that $\|(\text{B.5})\|_{C^0} \leq \delta/2$. The claim follows by the triangle inequality,

$$\left\| D^{(k,\boldsymbol{\alpha})}(\widetilde{u} - \widehat{u}) \right\|_{C^0} \leq \|(\text{B.4})\|_{C^0} + \|(\text{B.5})\|_{C^0} \leq \delta. \tag{B.6}$$

$\square$

**Lemma B.3.** *Let $\Delta_{h,t}^{k,s}$ be a finite difference operator cf. Section 3.3 and **SM** A.2, let $1 \leq j \leq d$, let $1 \leq q \leq \infty$, let $\ell \in \mathbb{N}_0$ and let $\boldsymbol{\alpha} \in \mathbb{N}_0^d$ with $\|\boldsymbol{\alpha}\|_1 = \ell$. Let $u, \widehat{u} \in C^{(s,\ell)}([-2h, 2h] \times D)$ such that for all $t \in [-2h, 2h]$,*

$$\left\| D_x^{\boldsymbol{\alpha}}(u(t, \cdot) - \widehat{u}(t, \cdot)) \right\|_{L^q(D)} \leq \varepsilon. \tag{B.7}$$

*Then there exists $c_s > 0$ holds that,*

$$\left\| D^{k,\boldsymbol{\alpha}} \left( \sum_{i=0}^{s-1} \frac{\Delta_{h,t}^{i,s-i}[\widehat{u}](0,x)}{h^i i!} t^i - u(t, \cdot) \right) \right\|_{L^q} \leq c_s \left( \varepsilon h^{-k} + |D_x^{\boldsymbol{\alpha}} u|_{C^{(s,0)}} h^{s-k} \right). \tag{B.8}$$

*Proof.* Let $t \in [-2h, 2h]$, $\boldsymbol{\alpha} \in \mathbb{N}_0^d$ with $\|\boldsymbol{\alpha}\|_1 = \ell$ and $x \in \mathbb{R}^d$ be arbitrary. We first observe that,

$$D^{k,\boldsymbol{\alpha}} \sum_{i=0}^{s-1} \frac{\Delta_{h,t}^{i,s-i}[u](0,x)}{h^i i!} t^i = \sum_{i=k}^{s-1} \frac{\Delta_{h,t}^{i,s-i}[D_x^{\boldsymbol{\alpha}} u](0,x)}{h^i (i-k)!} t^{i-k}. \tag{B.9}$$

Taylor's theorem then guarantees the existence of $\xi_{t,x} \in [-2h, 2h]$ such that

$$D^{k,\boldsymbol{\alpha}} \left( \sum_{i=0}^{s-1} \frac{\Delta_{h,t}^{i,s-i}[u](0,x)}{h^i i!} t^i - u(t, \cdot) \right)$$

$$= \sum_{i=0}^{s-1-k} \left[ \frac{\Delta_{h,t}^{i+k,s-i-k}[D_x^{\boldsymbol{\alpha}} u](0,x)}{h^{i+k} i!} t^i - \frac{D^{i+k,\boldsymbol{\alpha}} u(0,x)}{i!} t^i \right] + \frac{D^{s,\boldsymbol{\alpha}} u(\xi_{t,x}, x)}{(s-k)!} t^{s-k}. \tag{B.10}$$

Now observe that because of assumption (B.7) and the definition and properties (A.2) of the finite difference operator, there exists a constant $C_s > 0$ such that,

$$\left\| \Delta_{h,t}^{i+k,s-i-k}[D_x^{\boldsymbol{\alpha}} \widehat{u}](0,x) - \Delta_{h,t}^{i+k,s-i-k}[D_x^{\boldsymbol{\alpha}} u](0,x) \right\|_{L^q} \leq C_s \varepsilon,$$

$$\left| \frac{\Delta_{h,t}^{i+k,s-i-k}[D_x^{\boldsymbol{\alpha}} u](0,x)}{h^{i+k}} - D^{i+k,\boldsymbol{\alpha}} u(0,x) \right| \leq C_s |D_x^{\boldsymbol{\alpha}} u|_{C^{(s,0)}} h^{r-i-k}. \tag{B.11}$$

Combining all previous results provides us with the existence of a constant $c_s > 0$ such that,

$$\left\| D^{k,\boldsymbol{\alpha}} \left( \sum_{i=0}^{s-1} \frac{\Delta_{h,t}^{i,s-i}[\widehat{u}](0,x)}{h^i i!} t^i - u(t, \cdot) \right) \right\|_{L^q}$$

$$\leq \sum_{i=0}^{s-1-k} \left[ \frac{C_s \varepsilon}{h^{i+k} i!} h^i + \frac{C_s}{i!} |D_x^{\boldsymbol{\alpha}} u|_{C^{(s,0)}} h^{s-i-k} h^i \right] + \frac{1}{(s-k)!} |D_x^{\boldsymbol{\alpha}} u|_{C^{(s,0)}} h^{s-k}$$

$$\leq c_s \left( \varepsilon h^{-k} + |D_x^{\boldsymbol{\alpha}} u|_{C^{(s,0)}} h^{s-k} \right). \tag{B.12}$$

$\square$

**Definition B.4.** *Let $C > 0$, $N \in \mathbb{N}$, $0 < \varepsilon < 1$ and $\alpha = \ln\bigl(CN^k/\varepsilon\bigr)$. For every $1 \leq j \leq N$, we define the function $\Phi_j^N : [0,T] \to [0,1]$ by*

$$\Phi_1^N(t) = \frac{1}{2} - \frac{1}{2}\sigma\left(\alpha\left(t - \frac{T}{N}\right)\right),$$

$$\Phi_j^N(t) = \frac{1}{2}\sigma\left(\alpha\left(t - \frac{T(j-1)}{N}\right)\right) - \frac{1}{2}\sigma\left(\alpha\left(t - \frac{Tj}{N}\right)\right), \tag{B.13}$$

$$\Phi_N^N(t) = \frac{1}{2}\sigma\left(\alpha\left(t - \frac{T(N-1)}{N}\right)\right) + \frac{1}{2}.$$

The functions $\{\Phi_j^N\}_j$ approximate a partition of unity in the sense that for every $j$ it holds on $I_j^N$ that for some $\varepsilon > 0$,

$$1 - \sum_{v=-1}^{1} \Phi_{j+v}^N \lesssim \varepsilon \quad \text{and} \quad \sum_{\substack{|v| \geq 2, \\ j+v \in \{1,\ldots,N\}}} \Phi_{j+v}^N \lesssim \varepsilon. \tag{B.14}$$

This is made exact in [15, Section 4].

**Theorem B.5.** *Let $k \in \mathbb{N} \cup \{0\}$, $q \in \{2, \infty\}$, $\xi > 0$ and $s \in \mathbb{N}$. Let $\mu$ be a probability measure on $D$ and let $f \in C^s([0,T], L^q(\mu))$. Assume that for every $0 \leq \ell \leq k$ there is a constant $\mathcal{C}_\ell^* > 0$ for which it holds that for every $N \in \mathbb{N}$ there exist functions $\{p_j^N\}_{j=1}^N$ that satisfy for all $1 \leq j \leq N$,*

$$\left| f - p_j^N \right|_{C^\ell(J_j^N, L^q(\mu))} = \max_{t \in \left[\frac{(j-2)T}{N}, \frac{(j+1)T}{N}\right]} \left\| D_t^\ell(f(t, \cdot) - p_j^N(t, \cdot)) \right\|_{L^q(\mu)} \leq \mathcal{C}_\ell^* N^{-s+\ell} + \xi. \tag{B.15}$$

*Let $\mathcal{C}_k := \max\{\max_{0 \leq \ell \leq k} \mathcal{C}_\ell^*, \|f\|_{C^k([0,T], L^q(\mu))}, 1\}$. There exists a constant $C(k) > 0$ that only depends on $k$ such that for all $N \geq 3$ it holds that,*

$$\left\| f - \sum_{j=1}^{N} p_j^N \cdot \Phi_j^N \right\|_{C^k([0,T], L^q(\mu))} \leq C \ln^k(N) \left[\frac{\mathcal{C}_k}{N^{s-k}} + \xi N^k\right]. \tag{B.16}$$

*Proof.* We follow the proof of [15, Theorem 5.1]. All steps of the proofs are identical, with the only difference being that the $W^{k,\infty}([0,1]^d)$-norm of [15] is replaced by the $C^k([0,T], L^2(\mu))$-norm in this work. Following [15], one divides the domain $[0,T]$ into intervals $I_i^N = [t_{i-1}, t_i]$, with $t_i = iT/N$ and $N \in \mathbb{N}$ large enough. On each of these intervals, $f$ locally can be approximated (in Sobolev norm) by $p_j^N$, by virtue of the assumptions of the theorem. A global approximation can then be constructed by multiplying each $p_j^N$ with an approximation of the indicator function of the corresponding intervals and summing over all intervals.

We now highlight the main steps in the proof. *Step 2a* (as in [15]) results in the following estimate,

$$\left\| f - \sum_{j=1}^{N} f \cdot \Phi_j^N \right\|_{C^k(I_i^N, L^q(\mu))} \leq C\|f\|_{C^k(I_i^N, L^q(\mu))} \left(\varepsilon + N^{k+1} \ln^k\left(\frac{CN^k}{\varepsilon}\right)\varepsilon\right). \tag{B.17}$$

*Step 2b* results in the estimate,

$$\left\| \sum_{j=1}^{N} (f - p_j^N) \cdot \Phi_j^{N,d} \right\|_{C^k(I_i^N, L^q(\mu))} \leq C \ln^k\left(\frac{CN^k}{\varepsilon}\right)\left[\frac{\mathcal{C}_k}{N^{s-k}} + \xi N^k + \mathcal{C}_k N^{k+1}\varepsilon\right], \tag{B.18}$$

Putting everything together, we find that if $CN^k \geq \varepsilon e$,

$$\left\| f - \sum_{j=1}^{N} p_j^N \cdot \Phi_j^N \right\|_{C^k([0,T], L^q(\mu))}$$

$$\leq C \ln^k\left(\frac{CN^k}{\varepsilon}\right)\left[(\|f\|_{C^k(I_i^N, L^q(\mu))} + \mathcal{C}_k)N^{k+1}\varepsilon + \frac{\mathcal{C}_k}{N^{s-k}} + \xi N^k\right]. \tag{B.19}$$

In particular, if we set $N^{k+1}\varepsilon = N^{-s+k}$ and $N \geq 3$, then we find that

$$\left\| f - \sum_{j=1}^{N} p_j^N \cdot \Phi_j^N \right\|_{C^k([0,T],L^q(\mu))} \leq C \ln^k(N) \left[ \frac{\|f\|_{C^k(I_i^N, L^q(\mu))} + C_k}{N^{s-k}} + \xi N^k \right]. \quad \text{(B.20)}$$

$\square$

**Lemma B.6.** *Let $\mathcal{G}_\theta : \mathcal{X} \to \mathcal{H}$ be a tanh FNO with grid size $N \in \mathbb{N}$ and let $B > 0$. For every $\varepsilon > 0$, there exists a tanh DeepONet $\mathcal{G}_\theta^* : \mathcal{X} \to \mathcal{H}$ with $N^d$ sensors and $N^d$ branch and trunk nets such that*

$$\sup_{\|v\|_{L^\infty} \leq B} \sup_{x \in \mathbb{T}^d} \left| \mathcal{G}_\theta^*(v)(x) - \mathcal{G}_\theta(v)(x) \right| \leq \varepsilon. \quad \text{(B.21)}$$

*Furthermore,* $\operatorname{width}(\boldsymbol{\beta}) \sim N^d$, $\operatorname{depth}(\boldsymbol{\beta}) \sim \ln(N)$, $\operatorname{width}(\boldsymbol{\tau}) \sim N^d(N + \ln(N/\varepsilon))$ *and* $\operatorname{depth}(\boldsymbol{\tau}) = 3$.

*Proof.* This is a consequence of [38, Theorem 36] and Lemma D.1 with $\varepsilon \leftarrow N^d \varepsilon$. $\square$

## B.2 Proof of Theorem 3.5

*Proof.* **Step 1: construction.** To define the approximation, we divide $[0, T]$ into $M$ subintervals of the form $[t_{m-1}, t_m]$, where $t_m = mT/M$ with $1 \leq m \leq M$. One could approximate $u$ on every subinterval by an $s$-th order accurate Taylor approximation around $t_m$, provided that one has access to $D_t^i u(\cdot, t_m)$ for $0 \leq i \leq s - 1$. As those values are unknown, we resort to the finite difference approximation $D_t^i u(\cdot, t_m) \approx M^i \cdot \Delta_{1/M,t}^{i,s-i}[\mathcal{U}^\varepsilon(u_0, t_m)]$, which is a neural network. See **SM A.2** for an overview of the notation for finite difference operators. Moreover, we replace the univariate monomials $\varphi_i : [0, T] \to \mathbb{R} : t \mapsto t^i$ in the Taylor approximation by neural networks $\widehat{\varphi}_i^\delta : [0, T] \to \mathbb{R}$ with $\|\varphi_i - \widehat{\varphi}_i^\delta\|_{C^{k+1}} \lesssim \delta$. Lemma A.3 guarantees that the output of $(\widehat{\varphi}_i^\delta)_{i=1}^{s-1}$ can be obtained using a shallow network with width $2(s+1)$ (independent of $\delta$). The multiplication operator is replaced by a shallow neural network $\widehat{\times}_\delta : [-a, a]^2 \to \mathbb{R}$ (for suitable $a > 0$) for which $\| \times - \widehat{\times}_\delta\|_{C^{k+1}} \lesssim \delta$. By Lemma A.4 only four neurons are needed for this network. This results in the following approximation for $f \in C^0([0, T] \times D)$,

$$\widehat{N}_m^\delta[f](t, x) := \sum_{i=0}^{s-1} \widehat{\times}_\delta \left( \frac{\Delta_{1/M,t}^{i,s-i}[f](t_m, x)}{M^{-i}i!}, \widehat{\varphi}_i^\delta(t - t_m) \right) \quad \forall t \in [0, T], \ x \in D, \ 1 \leq m \leq M. \quad \text{(B.22)}$$

Next, we patch together these individual approximations by (approximately) multiplying them with a NN approximation of a partition of unity, denoted by $\Phi_1^M, \ldots, \Phi_M^M : [0, T] \to [0, 1]$, as introduced in Definition B.4 in **SM B**. Every $\Phi_m^M$ can be thought of as a NN approximation of the indicator function on $[t_{m-1}, t_m]$. For any $\varepsilon, \delta > 0$, we then define our final neural network approximation $\widehat{u} : [0, T] \times D \to \mathbb{R}$ as,

$$\widehat{u}(t, x) := \sum_{m=1}^{M} \widehat{\times}_\delta \left( \widehat{N}_m^\delta[\mathcal{U}^\varepsilon(u_0, t_m)](t, x), \Phi_m^M(t) \right) \quad \forall t \in [0, T], \ x \in D. \quad \text{(B.23)}$$

**Step 2: error estimate.** In order to facilitate the proof, we introduce the intermediate approximations $\widetilde{u} : [0, T] \times D \to \mathbb{R}$ and $N_m : C^0(D) \times [0, T] \times D \to \mathbb{R}$ by,

$$\widetilde{u}(t, x) := \sum_{m=1}^{M} N_m[\widehat{u}_m^\varepsilon](t, x) \cdot \Phi_m^M(t) := \sum_{m=1}^{M} \sum_{i=0}^{s-1} \frac{\Delta_{1/M,t}^{i,s-i}[\widehat{u}_m^\varepsilon](t_m, x)}{M^{-i}i!} \cdot \varphi_i(t - t_m) \cdot \Phi_m^M(t), \quad \text{(B.24)}$$

where $\widehat{u}_m^\varepsilon = \mathcal{U}^\varepsilon(u_0, t_m)$. Note that $\widehat{u}$ can be obtained from $\widetilde{u}$ by replacing the multiplication operator and the monomials by neural networks. Since these the size of these networks are independent of their accuracy $\delta$, we can assume without loss of generality that $\|D^{(k,\boldsymbol{\alpha})}(\widetilde{u} - \widehat{u})(t, \cdot)\|_{L^q} \leq \delta$ (see Lemma B.2) for any relevant $D^{(k,\boldsymbol{\alpha})}$ and $t$.

It remains to prove that $D^{(k,\boldsymbol{\alpha})}\widetilde{u} \approx D^{(k,\boldsymbol{\alpha})}u$. Combining the observation that $D^{(k,\boldsymbol{\alpha})}N_m[\widehat{u}_m^\varepsilon] = D_t^k N_m[D_x^{\boldsymbol{\alpha}}\widehat{u}_m^\varepsilon]$ with Lemma B.3 lets us conclude that for all $0 \le k \le s-1$ and $t \in [t_{m-2}, t_{m+2}]$,

$$\left\| D^{(k,\boldsymbol{\alpha})}(N_m[\widehat{u}_m^\varepsilon](t,\cdot) - u(t,\cdot)) \right\|_{L^q} \le C(r)M^k(\left\| D_x^{\boldsymbol{\alpha}}(\widehat{u}_m^\varepsilon - u)(\cdot, t_m) \right\|_{L^q} + |u|_{C^{(s,\ell)}}M^{-s}) \tag{B.25}$$

We use Theorem B.5 with $f \leftarrow u$, $p_j^N \leftarrow N_m[D_x^{\boldsymbol{\alpha}}\widehat{u}_m^\varepsilon]$, $\xi \leftarrow C(r)M^k\left\| D_x^{\boldsymbol{\alpha}}(\widehat{u}_m^\varepsilon - u)(\cdot, t_m) \right\|_{L^q}$, $\mathcal{C}_\ell^* \leftarrow C(s)|u|_{C^{(k,\ell)}}$, $N \leftarrow M$ to find that,

$$\left\| D^{(k,\boldsymbol{\alpha})}(\widehat{u} - u) \right\|_{L^q} \le C\ln^k(M)(\|u\|_{C^{(s,\ell)}}M^{k-s} + M^{2k}\left\| D_x^{\boldsymbol{\alpha}}(\widehat{u}_m^\varepsilon - u)(\cdot, t_m) \right\|_{L^q}), \tag{B.26}$$

where $C(r,s) > 0$ only depend on $r$ and $s$. Finally, using Lemma B.1 to bound $\left\| D_x^{\boldsymbol{\alpha}}(\widehat{u}_m^\varepsilon - u)(\cdot, t_m) \right\|_{L^q}$ and combining this with Assumption 3.1 proves (3.4).

**Step 3: size estimate.** The following holds,

$$\mathrm{depth}(\widehat{u}) \le C\mathrm{depth}(\mathcal{U}^\varepsilon), \mathrm{width}(\widehat{u}) \le CM\mathrm{width}(\mathcal{U}^\varepsilon). \tag{B.27}$$

$\square$

## B.3 Proof of Theorem 3.7

*Proof.* **Step 1: construction.** Let $N \in \mathbb{N}$, let $\mathcal{E}_N : C^0(T^d) \to \mathbb{R}^{|\mathcal{J}_N|}$ be an encoder and $\mathcal{Q}_N : \mathbb{R}^{|\mathcal{J}_N|} \to L_N^2$ be a trigonometric polynomial interpolation operator, cf. **SM** A.6. If we let $\widehat{\mathcal{G}} = \mathcal{U}^\varepsilon \circ \mathcal{Q}_N \circ \mathcal{E}_N$ then we can define an FNO $\mathcal{G}_\theta : L_N^2(\mathbb{T}^d) \to L_N^2(\mathbb{T}^d)$ as $\mathcal{G}_\theta(u_0)(x) = (\mathcal{Q}_N \circ \mathcal{E}_N \circ \widehat{\mathcal{G}})(u_0)(x)$.

**Step 2: error estimate.** We decompose the $L^2$-error of the FNO using the triangle inequality and the inequality $\|\mathcal{U}^\varepsilon - \widehat{\mathcal{G}}\|_{L^2} \le C_{\mathrm{stab}}^\varepsilon \|u_0 - \mathcal{Q}_N \circ \mathcal{E}_N \circ u_0\|_{L^p}$, which follows from Assumption 3.6,

$$\|\mathcal{G} - \mathcal{G}_\theta\|_{L^2} \le \|\mathcal{G} - \mathcal{U}^\varepsilon\|_{L^2} + C_{\mathrm{stab}}^\varepsilon \|u_0 - \mathcal{Q}_N \circ \mathcal{E}_N \circ u_0\|_{L^p} + \|\widehat{\mathcal{G}} - \mathcal{G}_\theta\|_{L^2}. \tag{B.28}$$

First, we find using a Sobolev embedding result (Lemma A.1) and Lemma A.2 that,

$$\left\| u_0 - (\mathcal{Q}_N \circ \mathcal{E}_N)(u_0) \right\|_{L^p} \le \left\| u_0 - (\mathcal{Q}_N \circ \mathcal{E}_N)(u_0) \right\|_{H^{d/p^*}} \le C(d,r)N^{-r+d/p^*}\|u_0\|_{H^r}, \tag{B.29}$$

where $p^*$ is such that $1/p + 1/p^* = 1/2$. Next, we observe that for any $u_0 \in \mathcal{X}$ with $\|u_0\|_{C^r} \le B$ that $\left\|(\mathcal{Q}_N \circ \mathcal{E}_N)(u_0)\right\|_{H^r(\mathbb{T}^d)} \le CB =: \overline{B}$. Hence, by applying Lemma A.2 to the second and last term of (B.28) we find that,

$$\|\mathcal{G} - \mathcal{G}_\theta\|_{L^2} \le C(\varepsilon + C_{\mathrm{stab}}^\varepsilon BN^{-r+d/p^*} + C_{\varepsilon,r}^{\overline{B}}N^{-r}). \tag{B.30}$$

**Step 3: size estimate.** As for any FNO, the width is equal to $N^d\mathrm{width}(\mathcal{U}^\varepsilon)$. The depth in this case is equal to $\mathrm{depth}(\mathcal{U}^\varepsilon)$.

$\square$

## B.4 Proof of Theorem 3.10

*Proof.* **Step 1: construction.**

Let $\varepsilon > 0$ and $n, N \in \mathbb{N}$. We first introduce some notation. Let $\mathcal{J}_N = \{0, \ldots, 2N+1\}^d$, $\mathcal{K}_N = \{-N, \ldots, N\}^d$, let $\{\mathbf{e}_j\}_{j\in\mathbb{N}}$ be an ordered Fourier basis, as described in **SM** A.5, and let $\{\widehat{\mathbf{e}}_j\}_{j\in\mathbb{N}}$ be a neural network approximation of the same basis such that

$$\max_{k\in\mathcal{K}_N} \|\mathbf{e}_k - \widehat{\mathbf{e}}_k\|_{C^r} \le \eta, \tag{B.31}$$

cf. Lemma D.1. Using notation from **SM** A.6, let $\mathcal{Q}_N : \mathbb{R}^{|\mathcal{J}_N|} \to C(\mathbb{T}^d)$ be the trigonometric polynomial interpolation operator as in (A.18) and let $\mathcal{E}_N : C(\mathbb{T}^d) \to \mathbb{R}^{|\mathcal{J}_N|}$ be the encoder as in (A.20). We define

$$\widehat{\mathcal{Q}}_N : \mathbb{R}^{|\mathcal{J}_N|} \to C(\mathbb{T}^d) : y \mapsto \frac{1}{|\mathcal{K}_N|} \sum_{k\in\mathcal{K}_N} \sum_{j\in\mathcal{J}_N} y_j a_{k,j} \widehat{\mathbf{e}}_k, \tag{B.32}$$

with coefficients $a_{k,j}$ as in (A.19), as a neural network approximation of $\mathcal{Q}_N$.

Inspired by the proof of Theorem 3.5 (and using its notation as well), we define $\widehat{\mathcal{G}} : C(\mathbb{T}^d) \to L^2(\mu)$ by

$$\widehat{\mathcal{G}}(u_0)(t,x) = \sum_{m=1}^{M} \sum_{i=0}^{s-1} \frac{\Delta_{1/M}^{i,s-i}[\mathcal{U}^\varepsilon(\mathcal{Q}_Z \circ \mathcal{E}_Z \circ u_0, t_m)](t_m, x)}{M^{-i}i!} \cdot \varphi_i^\delta(t - t_m)\Phi_m^M(t), \qquad \text{(B.33)}$$

Then it holds that

$$(\mathcal{Q}_N \circ \mathcal{E}_N \circ \widehat{\mathcal{G}})(u_0)(t,x) = \sum_{k \in \mathcal{K}_N} \sum_{j \in \mathcal{J}_N} \sum_{m=1}^{M} \sum_{i=0}^{s-1} \frac{a_{k,j}}{|\mathcal{K}_N|} \frac{\Delta_{1/M}^{i,s-i}[\mathcal{U}^\varepsilon(\mathcal{Q}_Z \circ \mathcal{E}_Z \circ u_0, t_m)](t_m, x_j)}{M^{-i}i!} \cdot \Psi_{i,m,k}(t,x)$$

$$\Psi_{i,m,k}(t,x) = \varphi_i^\delta(t - t_m)\Phi_m^M(t)\mathbf{e}_k(x).$$

$$\text{(B.34)}$$

Now for every $i, m, k$ let $\Psi_{i,m,k} : [0,T] \times \mathbb{T}^d \to \mathbb{R}$ be defined as,

$$\widehat{\Psi}_{i,m,k}(t,x) = \widehat{\times}_\delta \left( \varphi_i^\delta(t - t_m), \Phi_m^M(t), \widehat{\mathbf{e}}_k(x) \right), \qquad \text{(B.35)}$$

where $\widehat{\times}_\delta$ is a neural network approximation of the multiplication operator. We can then construct a DeepONet as

$$\mathcal{G}_\theta(u_0)(t,x) = \sum_{j=1}^{p} \beta_j(u_0)\tau_j(t,x)$$

$$= \sum_{i=0}^{s-1} \sum_{m=1}^{M} \sum_{k \in \mathcal{K}_N} \left[ \sum_{j \in \mathcal{J}_N} \frac{a_{k,j}}{|\mathcal{K}_N|} \frac{\Delta_{1/M}^{i,s-i}[\mathcal{U}^\varepsilon(\mathcal{Q}_Z \circ \mathcal{E}_Z \circ u_0, t_m)](t_m, x_j)}{M^{-i}i!} \right] \cdot \widehat{\Psi}_{i,m,k}(t,x).$$

$$\text{(B.36)}$$

We see that we need to set $p = sM(2N+1)^d$ and that the trunk nets are given by $\tau_j \sim \widehat{\Psi}_{i,m,k}$, up to a different indexing.

**Step 2: error estimate.** First we use Assumption 3.4 to see that

$$\left\| \mathcal{L}(\mathcal{G} - \mathcal{G}_\theta) \right\|_{L^2} \le C \sum_{k,\alpha} \left\| D^{(k,\alpha)}(\mathcal{G} - \mathcal{G}_\theta) \right\|_{L^2}. \qquad \text{(B.37)}$$

Next, we observe that using Assumption 3.1, Assumption 3.6 and (B.29) it holds that for all $t$,

$$\left\| (\mathcal{U}^\varepsilon(\mathcal{Q}_Z \circ \mathcal{E}_Z \circ u_0) - \mathcal{G}(u_0))(\cdot, t) \right\|_{L^2} \le \varepsilon + C_{\text{stab}}^\varepsilon CBZ^{-r+d/p^*}. \qquad \text{(B.38)}$$

One can then use Theorem 3.5, but by replacing $\varepsilon$ by (B.38) in the error bound (3.4), to find that

$$\left\| D^{(k,\alpha)}(\mathcal{G} - \widehat{\mathcal{G}}) \right\|_{L^2} \le C \ln^k(M)(\|u\|_{C^{(s,\ell)}} M^{k-s} + M^{2k}((\varepsilon + C_{\text{stab}}^\varepsilon Z^{-r+d/p^*})h^{-\ell} + C_{\varepsilon,\ell}^{CB} h^{r-\ell}))$$

$$\text{(B.39)}$$

Then, using the observation that $D^{(k,\alpha)}(\text{Id} - \mathcal{Q}_N \circ \mathcal{E}_N)\widehat{\mathcal{G}} = D_x^\alpha(\text{Id} - \mathcal{Q}_N \circ \mathcal{E}_N)D_t^k\widehat{\mathcal{G}}$ we find that

$$\left\| D^{(k,\alpha)}(\text{Id} - \mathcal{Q}_N \circ \mathcal{E}_N)\widehat{\mathcal{G}}(u_0) \right\|_{L^2} \le CN^{-(r-\ell)} \left\| D_t^k\widehat{\mathcal{G}}(u_0) \right\|_{H^r}, \qquad \text{(B.40)}$$

which can be combined with the estimate

$$\left\| D_t^k\widehat{\mathcal{G}}(u_0) \right\|_{H^r} \le M^{s-1} \cdot M^k \ln^k(M) \left\| \mathcal{U}^\varepsilon(\mathcal{Q}_Z \circ \mathcal{E}_Z \circ u_0) \right\|_{H^r} \le M^{s+k-1} \ln^k(M) C_{\varepsilon,r}^{\overline{B}}, \quad \text{(B.41)}$$

where we used that for $u_0 \in \mathcal{X}$ with $\|u_0\|_{C^r} \le B$ it holds $\left\| (\mathcal{Q}_N \circ \mathcal{E}_N)(u_0) \right\|_{H^r(\mathbb{T}^d)} \le CB =: \overline{B}$.
Next, we make the rough estimate that,

$$\left\| D^{(k,\alpha)}(\widehat{\mathcal{Q}}_N - \mathcal{Q}_N) \circ \mathcal{E}_N)\widehat{\mathcal{G}}(u_0) \right\|_{L^2} \le CN^d M^{s+k-1} \ln^k(M) \max_k \|\mathbf{e}_k - \widehat{\mathbf{e}}_k\|_{C^r}. \qquad \text{(B.42)}$$

Finally, using Lemma B.2 we find that

$$\left\| D^{(k,\alpha)}(\widehat{\mathcal{Q}}_N \circ \mathcal{E}_N \circ \widehat{\mathcal{G}} - \mathcal{G}_\theta)(u_0) \right\|_{L^2} \le \delta. \qquad \text{(B.43)}$$

By setting $\eta = N^{\ell-r-d}$, $h = 1/N$ and using that $M^{2k} \leq M^{k+s-1}$ and $C_{\varepsilon,\ell}^{\overline{B}} \leq C_{\varepsilon,r}^{\overline{B}}$ we find,

$$\left\|\mathcal{L}(\mathcal{G} - \mathcal{G}_\theta)\right\|_{L^2} \leq C \ln^k(M)(\|u\|_{C^{(s,\ell)}} M^{k-s} + M^{k+s-1}((\varepsilon + C_{\text{stab}}^\varepsilon Z^{-r+d/p^*})N^\ell + C_{\varepsilon,r}^{\overline{B}} N^{\ell-r})). \tag{B.44}$$

We conclude by using that $\ln^k(M) \leq CM^\rho$ for any $\rho > 0$.

**Step 3: size estimate.** It follows immediately that $\text{depth}(\boldsymbol{\beta}) = \text{depth}(\mathcal{U}^\varepsilon)$, $\text{width}(\boldsymbol{\beta}) = \mathcal{O}(M(Z^d + N^d\text{width}(\mathcal{U}^\varepsilon)))$, $\text{depth}(\boldsymbol{\tau}) = 3$ and $\text{width}(\boldsymbol{\tau}) = \mathcal{O}(MN^d(N + \ln(N)))$.

$\square$

### B.5 Proof of Theorem 3.11

*Proof.* Define the random variable $Y = \mathcal{E}_G(\theta^*(\mathcal{S}))^2 - \mathcal{E}_T(\theta^*(\mathcal{S}), \mathcal{S})^2$. Then if follows from equation (4.8) in the proof of [16, Theorem 5] that

$$\mathbb{P}(Y > \varepsilon^2) \leq \left(\frac{2R\mathfrak{L}}{\varepsilon^2}\right)^{d_\Theta} \exp\left(\frac{-2\varepsilon^4 n}{c^2}\right), \tag{B.45}$$

since $\mathbb{P}(Y > \varepsilon^2) = 1 - \mathbb{P}(\mathcal{A})$, where $\mathcal{A}$ is as defined in the proof of [16, Theorem 5]. It follows that

$$\mathbb{E}[Y] = \mathbb{E}[Y\mathbb{1}_{Y \leq \varepsilon^2}] + \mathbb{E}[Y\mathbb{1}_{Y > \varepsilon^2}] \leq \varepsilon^2 + c\mathbb{P}\left(Y > \varepsilon^2\right). \tag{B.46}$$

Setting $\varepsilon^2 = c\mathbb{P}\left(Y > \varepsilon^2\right)$ leads to

$$\mathbb{E}[Y] \leq 2\varepsilon^2 = \sqrt{\frac{2c^2}{n}\ln\left(\frac{c}{\varepsilon^2}\left(\frac{2R\mathfrak{L}}{\varepsilon^2}\right)^{d_\Theta}\right)}. \tag{B.47}$$

For $\varepsilon < 1$, and using that $\ln(x) \leq \sqrt{x}$ for all $x > 0$, this equality implies that

$$\varepsilon^{d_\Theta+1} \leq \frac{2c^3(2R\mathfrak{L})^{d_\Theta/2}}{n}. \tag{B.48}$$

Hence, we find that if $n \geq 2c^2 e^8/(2R\mathfrak{L})^{d_\Theta/2}$ then $\varepsilon^{d_\Theta+1} \leq ce^{-8}(2R\mathfrak{L})^{d_\Theta}$ which implies that

$$\left[\ln\left(\frac{c}{\varepsilon^2}\left(\frac{2R\mathfrak{L}}{\varepsilon^2}\right)^{d_\Theta}\right)\right]^{-1/2} \leq \frac{1}{2\sqrt{2}}. \tag{B.49}$$

Using once more that $\varepsilon^2 = c\mathbb{P}\left(Y > \varepsilon^2\right)$ and (B.49) gives us,

$$\mathbb{E}[Y] \leq \sqrt{\frac{2c^2}{n}\ln\left(c(2R\mathfrak{L})^{d_\Theta}\left(\frac{\sqrt{2n}}{c}\left[\ln\left(\frac{c}{\varepsilon^2}\left(\frac{2R\mathfrak{L}}{\varepsilon^2}\right)^{d_\Theta}\right)\right]^{-1/2}\right)^{d_\Theta+1}\right)} \tag{B.50}$$

$$\leq \sqrt{\frac{2c^2}{n}\ln((a\mathfrak{L}\sqrt{n})^{d_\Theta+1})} = \sqrt{\frac{2c^2(d_\Theta+1)}{n}\ln(a\mathfrak{L}\sqrt{n})}.$$

$\square$

## C Additional material for Section 4.1

### C.1 Auxiliary results

**Lemma C.1.** *Let $\varepsilon > 0$, let $(\Omega, \mathcal{F}, \mathcal{P})$ be a probability space, and let $X : \Omega \to \mathbb{R}$ be a random variable that satisfies $\mathbb{E}\left[|X|\right] \leq \varepsilon$. Then it holds that $\mathbb{P}(|X| \leq \varepsilon) > 0$.*

*Proof.* This result is [23, Proposition 3.3]. $\square$

**Lemma C.2.** *Let $\gamma \in \{0, 1\}$, $\beta \in [1, \infty)$, $\alpha_0, \alpha_1, x_0, x_1, x_2, \ldots \in [0, \infty)$ satisfy for all $k \in \mathbb{N}_0$ that*

$$x_k \leq \mathbb{1}_{\mathbb{N}}(k)(\alpha_0 + \alpha_1 k)\beta^k + \sum_{l=0}^{k-1} (k - l)^\gamma \beta^{(k-l)} \left[ x_l + x_{\max\{l-1, 0\}} \right]. \tag{C.1}$$

*Then it holds for all $k \in \mathbb{N}_0$ that*

$$x_k \leq \frac{(\alpha_0 + \alpha_1)\beta^k \mathbb{1}_{\mathbb{N}}(k)}{(4 + \gamma)^{1/2}(1 + 2^{(1+\gamma)/2})^{-k}} = \begin{cases} \mathbb{1}_{\mathbb{N}}(k)(\alpha_0 + \alpha_1)2^{-1}(1 + 2^{1/2})^k \beta^k & : \gamma = 0 \\ \mathbb{1}_{\mathbb{N}}(k)(\alpha_0 + \alpha_1)5^{-1/2}(3\beta)^k & : \gamma = 1. \end{cases} \tag{C.2}$$

*Proof.* This result is [33, Corollary 4.3]. □

**Lemma C.3.** *Let $\alpha \in [1, \infty)$, $x_0, x_1, \ldots \in [0, \infty)$ satisfy for all $k \in \mathbb{N}_0$ that $x_k \leq \alpha x_{k-1}^k$. Then it holds for all $k \in \mathbb{N}_0$ that*

$$x_k \leq \alpha^{(k+1)!} x_0^{k!} \tag{C.3}$$

*Proof.* We provide a proof by induction. First of all, it is clear that $x_0 \leq \alpha x_0$. For the induction step, assume that $x_{k-1} \leq \alpha^{k!} x_0^{(k-1)!}$ for an arbitrary $k \in \mathbb{N}_0$. We calculate that

$$x_k \leq \alpha \left( \alpha^{k!} x_0^{(k-1)!} \right)^k \leq \alpha^{(k+1)!} x_0^{k!}. \tag{C.4}$$

This proves the statement. □

**Lemma C.4.** *Let $\ell \in \mathbb{N}$, $f \in C^\ell(\mathbb{R}, \mathbb{R})$, $h \in C^\ell(\mathbb{T}^d, \mathbb{R})$ and let $B_\ell$ denote the $\ell$-th Bell number. Then it holds that*

$$|f \circ h|_{C^\ell(\mathbb{R})} \leq \|f\|_{C^\ell(\mathbb{R})} \left( B_\ell \|h\|_{C^{\ell-1}(\mathbb{T}^d)}^\ell + |h|_{C^\ell(\mathbb{T}^d)} \right). \tag{C.5}$$

*Proof.* Let $\Pi$ be the set of all partitions of the set $\{1, \ldots, \ell\}$, let $\alpha \in \mathbb{N}_0^d$ such that $\|\alpha\|_1 = \ell$ and let $\iota : \mathbb{N}^\ell \to \mathbb{N}^d$ be a map such that $D^\alpha = \frac{\partial^\ell}{\prod_{j=1}^\ell x_{\iota(j)}}$. Then the Faà di Bruno formula can be reformulated as [12],

$$\begin{aligned} D^\alpha f(h(x)) &= \sum_{\pi \in \Pi} f^{(|\pi|)}(h(x)) \cdot \prod_{B \in \pi} \frac{\partial^{|B|} h(x)}{\prod_{j \in B} \partial x_{\iota(j)}} \\ &= \sum_{\substack{\pi \in \Pi, \\ |\pi| \geq 2}} f^{(|\pi|)}(h(x)) \cdot \prod_{B \in \pi} \frac{\partial^{|B|} h(x)}{\prod_{j \in B} \partial x_{\iota(j)}} + f'(h(x)) D^\alpha h(x). \end{aligned} \tag{C.6}$$

Combining this formula with the definition of the Bell number as $B_\ell = |\Pi|$, we find the following upper bound,

$$\begin{aligned} |f \circ h|_{C^\ell(\mathbb{R})} &\leq \sum_{\pi \in \Pi} \|f\|_{C^\ell(\mathbb{R})} \|h\|_{C^{\ell-1}(\mathbb{R})}^\ell + \|f\|_{C^1(\mathbb{R})} |h|_{C^\ell(\mathbb{R})} \\ &\leq \|f\|_{C^\ell(\mathbb{R})} \left( B_\ell \|h\|_{C^{\ell-1}(\mathbb{R})}^\ell + |h|_{C^\ell(\mathbb{R})} \right). \end{aligned} \tag{C.7}$$

□

## C.2 Proof of Theorem 4.2

**Definition C.5.** *Let $(\Omega, \mathcal{F}, \mu)$ be a measure space and let $q > 0$. For every $\mathcal{F}/\mathcal{B}(\mathbb{R}^d)$-measurable function $f : \Omega \to \mathbb{R}^d$, we define*

$$\|f\|_{\mathcal{L}^q(\mu, \|\cdot\|_{\mathbb{R}^d})} := \left[ \int_\Omega \|f(\omega)\|_{\mathbb{R}^d}^q \mu(d\omega) \right]^{1/q}. \tag{C.8}$$

Let $(\Omega, \mathcal{F}, P, (\mathbb{F}_t)_{t\in[0,T]})$ be a stochastic basis, $D \subseteq \mathbb{R}^d$ a compact set and, for every $x \in D$, let $X^x : \Omega \times [0,T] \to \mathbb{R}^d$ be the solution, in the Itô sense, of the following stochastic differential equation,

$$dX_t^x = \mu(X_t^x)dt + \sigma(X_t^x)dB_t, \quad X_0^x = x, \quad x \in D, t \in [0,T], \tag{C.9}$$

where $B_t$ is a standard $d$-dimensional Brownian motion on $(\Omega, \mathcal{F}, P, (\mathbb{F}_t)_{t\in[0,T]})$. The existence of $X^x$ is guaranteed by [3, Theorem 4.5.1].

As in [16, Theorem 3.3] we define $\rho_d$ as

$$\rho_d := \max_{x \in D} \sup_{\substack{s,t\in[0,T], \\ s<t}} \frac{\|X_s^x - X_t^x\|_{\mathcal{L}^q(P, \|\cdot\|_{\mathbb{R}^d})}}{|s-t|^{\frac{1}{p}}} < \infty, \tag{C.10}$$

where $X^x$ is the solution, in the Itô sense, of the SDE (C.9), $q > 2$ is independent of $d$ and $\|\cdot\|_{\mathcal{L}^q(P, \|\cdot\|_{\mathbb{R}^d})}$ is as in Definition C.5.

**Lemma C.6.** *In Setting 4.1, Assumption 3.1 and Assumption 3.6 are satisfied with*

$$\left\|u(\cdot, t) - \mathcal{U}^\varepsilon(\varphi, t)\right\|_{L^2(\mu)} \le \varepsilon, \qquad C_{\varepsilon,\ell}^B = CB \cdot \mathrm{poly}(d\rho_d), \qquad C_{\mathrm{stab}}^\varepsilon = 1, \qquad p = \infty, \tag{C.11}$$

*where $t \in [0,T]$ and $\varphi \in C_0^2(\mathbb{R}^d)$. Moreover, there exists $C^* > 0$ (independent of $d$) for which it holds that $\mathrm{depth}(\mathcal{U}^\varepsilon) \le C^* \mathrm{depth}(\widehat{\varphi}^\varepsilon)$ and $\{\mathrm{width, size}\}(\mathcal{U}^\varepsilon) \le C^* \varepsilon^{-2}\{\mathrm{width, size}\}(\widehat{\varphi}^\varepsilon)$.*

*Proof.* It follows from the Feynman-Kac formula that $u(t,x) = \mathbb{E}\left[\varphi(X_t^x)\right]$ [62]. Replacing $\varphi$ by a neural network $\widehat{\varphi}^\varepsilon$ with $\|\varphi - \widehat{\varphi}^\varepsilon\|_{C^0} \le \varepsilon$ gives us for any probability measure $\mu$ that,

$$\left\|\mathbb{E}\left[\varphi(X_t^x)\right] - \mathbb{E}\left[\widehat{\varphi}^\varepsilon(X_t^x)\right]\right\|_{L^2(\mu)} \le \|\varphi - \widehat{\varphi}^\varepsilon\|_{C^0}. \tag{C.12}$$

Using [16, Lemma A.2] (which is based on [23]) we find,

$$\mathbb{E}\left[(I)\right] := \mathbb{E}\left[\left(\int_D \left|\mathbb{E}\left[\widehat{\varphi}^\varepsilon(X_t^x)\right] - \frac{1}{m}\sum_{i=1}^m \widehat{\varphi}^\varepsilon(X_t^x(\omega_m))\right|^2 \mu(dx)\right)^{1/2}\right] \le \frac{2\|\widehat{\varphi}^\varepsilon\|_{C^0}}{\sqrt{m}}. \tag{C.13}$$

From [16, Lemma A.5], for all $x \in \mathbb{R}^d$, $t \in [0,T]$ and $\omega \in \Omega$ it holds that

$$X_t^x(\omega) = \sum_{i=1}^d \left(X_t^{e_i}(\omega) - X_t^0(\omega)\right)x_i + X_t^0(\omega). \tag{C.14}$$

Using this equality, together with Hölder's inequality and the boundedness of $\|X_t^x\|_{L^p}$ [16, Lemma A.5] we find that,

$$\mathbb{E}\left[(II_{\boldsymbol{\alpha}})\right] := \mathbb{E}\left[\left(\int_D \left|\mathbb{E}\left[D_x^{\boldsymbol{\alpha}}\widehat{\varphi}^\varepsilon(X_t^x)\right] - \frac{1}{m}\sum_{i=1}^m D_x^{\boldsymbol{\alpha}}\widehat{\varphi}^\varepsilon(X_t^x(\omega_m))\right|^2 \mu(dx)\right)^{1/2}\right]$$

$$\le C \cdot \mathrm{poly}(d\rho_d) \cdot \mathbb{E}\left[\left(\int_D \left|\mathbb{E}\left[\widehat{\varphi}^\varepsilon(X_t^x)\right] - \frac{1}{m}\sum_{i=1}^m \widehat{\varphi}^\varepsilon(X_t^x(\omega_m))\right|^4 \mu(dx)\right)^{1/4}\right]$$

$$\le CB \cdot \mathrm{poly}(d\rho_d) \tag{C.15}$$

Combining the previous results gives us,

$$\mathbb{E}\left[\sqrt{m} \cdot (I) + \sum_{\|\boldsymbol{\alpha}\|_1 \le \ell} (II_{\boldsymbol{\alpha}})\right] \le CB \cdot \mathrm{poly}(d\rho_d). \tag{C.16}$$

If we combine this with Lemma C.1 then we find the existence of $(\omega_i^*)_{i=1}^m$ such that for

$$\mathcal{U}^\varepsilon(\varphi,t)(x) = \frac{1}{m}\sum_{i=1}^m \widehat{\varphi}^\varepsilon\left(\sum_{i=1}^d \left(X_t^{e_i}(\omega_i^*) - X_t^0(\omega_i^*)\right)x_i + X_t^0(\omega_i^*)\right) \tag{C.17}$$

it holds that

$$\left\|\mathbb{E}\left[\widehat{\varphi}^\varepsilon(X_t^x)\right] - \mathcal{U}^\varepsilon(\varphi,t)\right\|_{L^2(D)} \le \frac{2C}{\sqrt{m}}. \tag{C.18}$$

and by setting $m = \varepsilon^{-2}$ and (C.12) we find that,

$$\left\|u(\cdot,t) - \mathcal{U}^\varepsilon(\varphi,t)\right\|_{L^2(D)} \le \varepsilon, \qquad C_{\varepsilon,\ell}^B = CB \cdot \mathrm{poly}(\rho_d), \qquad C_{\mathrm{stab}}^\varepsilon = 1, \qquad p = \infty. \tag{C.19}$$

Moreover, it holds that

$$\mathrm{depth}(\mathcal{U}^\varepsilon) \le C^*\mathrm{depth}(\widehat{\varphi}^\varepsilon), \qquad \{\mathrm{width},\mathrm{size}\}(\mathcal{U}^\varepsilon) \le C^*\varepsilon^{-2}\{\mathrm{width},\mathrm{size}\}(\widehat{\varphi}^\varepsilon) \tag{C.20}$$

where we write $C^* = C\mathrm{poly}(d\rho_d)$. $\qquad\square$

We can now present the proof of the actual theorem.

*Proof of Theorem 4.2.* We use Theorem 3.5 with $k = 1$ and $\ell = 2$ and combine the result with Lemma C.6. We find that for every $M \in \mathbb{N}$ and $\delta, h > 0$ it holds that,

$$\begin{aligned}
&\left\|\mathcal{L}(\widehat{u} - u)\right\|_{L^q([0,T]\times D)} + \|\widehat{u} - u\|_{L^2(\partial([0,T]\times D))} \\
&\qquad \le CB \cdot \mathrm{poly}(d\rho_d) \cdot \ln(M)(\|u\|_{C^{(1,2)}}M^{1-s} + M^2(\delta h^{-2} + h^{r-2})).
\end{aligned} \tag{C.21}$$

Set $\delta = h^r$, $M^{-1-s} = \delta^{1-2/r}$ we find that

$$\left\|\mathcal{L}(\widehat{u} - u)\right\|_{L^q([0,T]\times D)} + \|\widehat{u} - u\|_{L^2(\partial([0,T]\times D))} \le CB \cdot \mathrm{poly}(d\rho_d)\ln\left(1/\delta\right)\delta^{\frac{r-2}{r}\frac{s-1}{s+1}} \tag{C.22}$$

Using that $\ln(M) \le CM^\sigma$ for arbitrarily small $\sigma > 0$, we find that we should set

$$\delta = \varepsilon^{\frac{r+\sigma}{r-2}\frac{s+1}{s-1}}, \qquad M = \varepsilon^{\frac{-1-\sigma}{s-1}} \tag{C.23}$$

$\square$

## C.3 Nonlinear parabolic equations

Some examples of nonlinear parabolic PDEs of the type (4.3) are:

- The *Kolmogorov–Petrovsky–Piskunov (KPP) equation* [37] is a celebrated model that is often used to model wave propagation and population genetics. The model is particularly useful for systems that exhibit phase transitions. One obtains the KPP equation if one chooses a sufficiently smooth nonlinearity $F$ that satisfies the requirements $F(0) = F(1) = 0$, $F'(0) = r > 0$, $F(u) > 0$ and $F'(u) < r$ for all $0 < u < 1$. Well-known examples include the *Fisher equation* [21] with $F(u) = ru(1 - u)$ and the *Allen-Cahn equation* [1] with $F(u) = ru(1 - u^2)$.

- *Branching diffusion processes* give a probabilistic representation of the KPP equation for the case where $F(u) = \beta(\sum_{k=0}^\infty a_k u^k - u)$ with $a_k \ge 0$ and $\sum_k a_k = 1$. In this setting, the PDE (4.3) describes a $d$-dimensional branching Brownian motion, where every particle in the system dies in an an exponential time of parameter $\beta$ and created $k$ i.i.d. descendants with probability $a_k$ [29, 58].

- Finally, the PDE (4.3) arises in the context of credit valuation adjustment when pricing derivative contracts to compute the counterparty risk valuation, e.g. [28]. The dimension $d$ corresponds to the number of underlying assets and can be very high.

## C.4 Multilevel Picard approximations

In what follows, we will provide a definition of a particular kind of MLP approximation (cf. [33]) and a theorem that quantifies the accuracy of the approximation. First, we rigorously introduce the setting of the nonlinear parabolic PDE (4.3) that is under consideration, cf. [33, Setting 3.2 with $p \leftarrow 0$]. We choose the $d$-dimensional torus $\mathbb{T}^d = [0, 2\pi)^d$ as domain and impose periodic boundary conditions. This setting allows us to use the results of [33], which are set in $\mathbb{R}^d$, and yet still consider a bounded domain so that the error can be quantified using an uniform probability measure.

**Setting C.7.** *Let $d, m \in \mathbb{N}$, $T, L, \mathcal{L} \in [0, \infty)$, let $(\mathbb{T}^d, \mathcal{B}(\mathbb{T}^d), \mu)$ be a probability space where $\mu$ is the rescaled Lebesgue measure, let $g \in C(\mathbb{T}^d, \mathbb{R}) \cap L^2(\mu)$, let $F \in C(\mathbb{R}, \mathbb{R})$, assume for all $x \in \mathbb{T}^d$, $y, z \in \mathbb{R}$ that*

$$|F(y) - F(z)| \leq L|y - z|, \qquad \max\{|F(y)|, |g(x)|\} \leq \mathcal{L}. \tag{C.24}$$

*Let $u_d \in C^{1,2}([0, T] \times \mathbb{R}^d, \mathbb{R}) \cap L^2(\mu)$ satisfy for all $t \in [0, T]$, $x \in \mathbb{R}^d$ that*

$$(\partial_t u_d)(t, x) = (\Delta_x u_d)(t, x) + F(u_d(t, x)), \qquad u_d(0, x) = g(x). \tag{C.25}$$

*Assume that for every $\varepsilon > 0$ there exists a neural network $\widehat{F}_\varepsilon$, a neural network $\widehat{g}_\varepsilon$ and a neural network $\mathcal{I}_\varepsilon$ with depth $\mathrm{depth}(\mathcal{I}_\varepsilon) = \mathrm{depth}(\widehat{F}_\varepsilon)$ such that*

$$\left\|\widehat{F}_\varepsilon - F\right\|_{C^0(\mathbb{R})} \leq \varepsilon, \quad \|\widehat{g}_\varepsilon - g\|_{L^2(\mu)} \leq \varepsilon, \quad \|\mathcal{I}_\varepsilon - \mathrm{Id}\|_{C^0([-1-\mathcal{L}, 1+\mathcal{L}])} \leq \varepsilon. \tag{C.26}$$

Note that for some of the equations introduced in Section C.3 the nonlinearity $F$ might not be globally Lipschitz and hence does not satisfy (C.24). However, it is easy to argue or rescale $g$ [43, 5] such that $u_d$ is globally bounded by some constant $C$. For instance, for the Allen-Cahn equation it holds that if $\|g\|_{L^\infty} \leq 1$ then $\|u_d(t, \cdot)\|_{L^\infty} \leq 1$ for any $t \in [0, T]$ [75]. One can then define a 'smooth', globally Lipschitz, bounded function $\widetilde{F} : \mathbb{R} \to \mathbb{R}$ such that $\widetilde{F}(v) = F(v)$ for $|v| \leq C$ and such that $\widetilde{F}(v) = 0$ for $|v| > 2C$. This will then also ensure the existence of a neural network $\widehat{F}$ that is close to $\widetilde{F}$ in $C^0(\mathbb{R})$-norm.

In this setting, multilevel Picard approximations can be introduced. We follow the definition of [33].

**Definition C.8** (MLP approximation). *Assume Setting C.7. Let $\Theta = \bigcup_{n \in \mathbb{N}} \mathbb{Z}^n$, let $(\Omega, \mathcal{F}, \mathbb{P})$ be a probability space, let $Y^\theta : \Omega \to [0, 1]$, $\theta \in \Theta$, be i.i.d. random variables, assume for all $\theta \in \Theta$, $r \in (0, 1)$ that $\mathbb{P}(Y^\theta \leq r) = r$, let $\mathcal{U}^\theta : [0, T] \times \Omega \to [0, T]$, $\theta \in \Theta$, satisfy for all $t \in [0, T]$, $\theta \in \Theta$ that $\mathcal{U}_t^\theta = t + (T - t)Y^\theta$, let $W^\theta : [0, T] \times \Omega \to \mathbb{R}^d$, $\theta \in \Theta$, be independent standard Brownian motions, assume that $(\mathcal{U}^\theta)_{\theta \in \Theta}$ and $(W^\theta)_{\theta \in \Theta}$ are independent, and let $U_n^\theta : [0, T] \times \mathbb{T}^d \times \Omega \to \mathbb{R}$, $n \in \mathbb{Z}$, $\theta \in \Theta$, satisfy for all $n \in \mathbb{N}_0$, $\theta \in \Theta$, $t \in [0, T]$, $x \in \mathbb{T}^d$ that*

$$U_n^\theta(t, x) = \frac{\mathbb{1}_\mathbb{N}(n)}{m^n} \left[\sum_{k=1}^{m^n} g(x + W_{T-t}^{(\theta,0,-k)})\right]$$

$$+ \sum_{i=0}^{n-1} \frac{(T-t)}{m^{n-i}} \left[\sum_{k=1}^{m^{n-i}} (F(U_i^{(\theta,i,k)}) - \mathbb{1}_\mathbb{N}(i)F(U_{i-1}^{(\theta,-i,k)}))(\mathcal{U}_t^{(\theta,i,k)}, x + W_{\mathcal{U}_t^{(\theta,i,k)}-t}^{(\theta,i,k)})\right]. \tag{C.27}$$

**Example C.9.** *In order to improve the intuition of the reader regarding Definition C.8, we provide explicit formulas for the multilevel Picard approximation (C.27) for $n = 0$ and $n = 1$,*

$$U_0^\theta(t, x) = 0 \quad and \quad U_1^\theta(t, x) = \frac{1}{m}\left[\sum_{k=1}^m g(x + W_{T-t}^{(\theta,0,-k)})\right] + (T - t)F(0). \tag{C.28}$$

Finally, we provide a result on the accuracy of MLP approximations at single space-time points.

**Theorem C.10.** *It holds for all $n \in \mathbb{N}_0$, $t \in [0, T]$, $x \in \mathbb{T}^d$ that*

$$\left(\mathbb{E}\left[\left|U_n^0(t, x) - u(t, x)\right|^2\right]\right)^{1/2} \leq \frac{\mathcal{L}(T+1)\exp(LT)(1 + 2LT)^n}{m^{n/2}\exp(-m/2)}. \tag{C.29}$$

*Proof.* This result is [33, Corollary 3.15] with $p \leftarrow 0$, $\mathfrak{p} \leftarrow 2$ and $\mathcal{L} \leftarrow \mathcal{L}/2$. □

## C.5 Neural network approximation of nonlinear parabolic equations

In this section, we will prove that the solution of the nonlinear parabolic PDE as in Setting C.7 can be approximated with a neural network without the curse of dimensionality. At this point, we do not specify the activation function, with the only restriction being that the considered neural networks should be expressive enough to satisfy (C.26). By emulating an MLP approximation and using that $F$, $g$ and the identity function can be approximated using neural networks, the following theorem can be proven.

**Theorem C.11.** *Assume Setting C.7. For every $\varepsilon, \sigma > 0$ and $t \in [0, T]$ there exists a neural network $\widehat{u}_\varepsilon : \mathbb{T}^d \to \mathbb{R}$ such that*

$$\left\| \widehat{u}_\varepsilon(\cdot) - u(t, \cdot) \right\|_{L^2(\mu)} \leq \varepsilon. \tag{C.30}$$

*In addition, $\widehat{u}$ satisfies that*

$$\mathrm{depth}(\widehat{u}_\varepsilon) \leq \mathrm{depth}(\widehat{g}_\delta) + \log_{C_2}(3C_1 \exp(m/2)/\varepsilon)\mathrm{depth}(\widehat{F}_\delta),$$

$$\mathrm{width}(\widehat{u}_\varepsilon), \mathrm{size}(\widehat{u}_\varepsilon) \leq (\mathrm{size}(\widehat{g}_\delta) + \mathrm{size}(\widehat{F}_\delta) + \mathrm{size}(\mathcal{I}_\delta)) \left( \frac{4C_1 \exp(m/2)}{\varepsilon} \right)^{2+3\sigma}, \tag{C.31}$$

*where*

$$C_1 = (T+1)(1 + \mathcal{L}\exp(LT)), \qquad C_2 = 5 + 3LT,$$

$$\delta = \frac{\varepsilon^2}{9C_1^2 \exp(m/2)}, \qquad m = C_2^{2(1+1/\sigma)}. \tag{C.32}$$

*Proof.* **Step 1: construction of the neural network.** Let $\varepsilon, \delta > 0$ be arbitrary and let $\widehat{F} = \widehat{F}_\delta$, $\widehat{g} = \widehat{g}_\delta$ and $\mathcal{I} = \mathcal{I}_\delta$ as in Setting C.7. We then define for all $n \in \mathbb{N}$ and $\theta \in \Theta$,

$$\widehat{U}_n^\theta(t, x) = \frac{\mathbb{1}_{\mathbb{N}}(n)}{m^n} \left[ \sum_{k=1}^{m^n} (\mathcal{I}^{n-1} \circ \widehat{g})(x + W_{T-t}^{(\theta, 0, -k)}) \right]$$

$$+ \sum_{i=0}^{n-1} \frac{(T-t)}{m^{n-i}} \left[ \sum_{k=1}^{m^{n-i}} ((\mathcal{I}^{n-i-1} \circ \widehat{F})(\widehat{U}_i^{(\theta, i, k)}) - \mathbb{1}_{\mathbb{N}}(i)(\mathcal{I}^{n-i} \circ \widehat{F})(\widehat{U}_{i-1}^{(\theta, -i, k)}))(\mathcal{U}_t^{(\theta, i, k)}, x + W_{\mathcal{U}_t^{(\theta, i, k)} - t}^{(\theta, i, k)}) \right], \tag{C.33}$$

with notation and random variables cf. Definition C.8. Note that for every $t \in [0, T]$, $n \in \mathbb{N}$, $\theta \in \Theta$, every realization of the random variable $\widehat{U}_n^\theta(t, \cdot)$ is a neural network that maps from $\mathbb{T}^d$ to $\mathbb{R}$.

Let $n \in \mathbb{N}_0$, $m \in \mathbb{N}$ and $t \in [0, T]$ be arbitrary. Integrating the square of the error bound of Theorem C.10 and Fubini's theorem tell us that

$$\mathbb{E}\left[ \int_{\mathbb{T}^d} \left| U_n^0(t, x) - u(t, x) \right|^2 d\mu(x) \right] = \int_{\mathbb{T}^d} \mathbb{E}\left[ \left| U_n^0(t, x) - u(t, x) \right|^2 \right] d\mu(x)$$

$$\leq \frac{4\mathcal{L}^2(T+1)^2 \exp(2LT)(1 + 2LT)^{2n}}{m^n \exp(-m)}. \tag{C.34}$$

From Lemma C.1 it then follows that

$$\mathbb{P}\left( \int_{\mathbb{T}^d} \left| U_n^0(t, x) - u(t, x) \right|^2 d\mu(x) \leq \frac{4\mathcal{L}^2(T+1)^2 \exp(2LT)(1 + 2LT)^{2n}}{m^n \exp(-m)} \right) > 0. \tag{C.35}$$

As a result, there exists $\overline{\omega} = \overline{\omega}(t, n, m) \in \Omega$ and a realization $U_n^0(\overline{\omega})$ such that

$$\left\| U_n^0(\overline{\omega})(t, \cdot) - u(t, \cdot) \right\|_{L^2(\mu)} \leq \frac{\mathcal{L}(T+1)\exp(LT)(1 + 2LT)^n}{m^{n/2} \exp(-m/2)}. \tag{C.36}$$

We define

$$\omega : [0, T] \times \mathbb{N}^2 \to \Omega : (t, n, m) \mapsto \overline{\omega}(t, n, m) \tag{C.37}$$

and set for every $1 \leq k \leq n$,

$$\widehat{U}_{k,\omega}^\theta(t, x) = \widehat{U}_k^\theta(\omega(t, n, m))(t, x) \quad \text{and} \quad U_{k,\omega}^\theta(t, x) = U_k^\theta(\omega(t, n, m))(t, x) \tag{C.38}$$

for all $k \in \mathbb{N}_0$ and all $\theta \in \Theta$. We then define our approximation as $\widehat{U}_{n,\omega}^0(t, \cdot)$.

**Step 2: error estimate.** We will quantify how well $\widehat{U}_{n,\omega}^0$ approximates $U_{n,\omega}^0$. Using the calculation that for $f_1, f_2 \in C^1(\mathbb{R})$ and $h_1, h_2 \in L^2(\mu)$ it holds that

$$
\begin{aligned}
\|f_1 \circ h_1 - f_2 \circ h_2\|_{L^2(\mu)} &\leq \|f_1 \circ h_1 - f_2 \circ h_1\|_{L^2(\mu)} + \|f_2 \circ h_1 - f_2 \circ h_2\|_{L^2(\mu)} \\
&\leq \|f_1 - f_2\|_{C^0(\mathbb{R})} + |f_2|_{Lip(\mathbb{R})}\|h_1 - h_2\|_{L^2(\mu)},
\end{aligned}
\tag{C.39}
$$

and the fact that $2n \leq 2^n$ for $n \in \mathbb{N}$ we find that it holds for every $\theta \in \Theta$ that,

$$
\left\|\widehat{U}_{n,\omega}^\theta(t, \cdot) - U_{n,\omega}^\theta(t, \cdot)\right\|_{L^2(\mu)}
$$

$$
\leq \mathbb{1}_{\mathbb{N}}(n)\left(\|\widehat{g} - g\|_{L^2(\mu)} + (n-1)\|\mathcal{I} - \mathrm{Id}\|_{C^0}\right) + T\sum_{i=0}^{n-1}\left\|\mathcal{I}^{n-i-1} \circ \widehat{F} \circ \widehat{U}_{i,\omega}^{(\theta,i,k)} - F \circ U_{i,\omega}^{(\theta,i,k)}\right\|_{L^2(\mu)}
$$

$$
+ T\sum_{i=0}^{n-1}\mathbb{1}_{\mathbb{N}}(i)\left\|\mathcal{I}^{n-i} \circ \widehat{F} \circ \widehat{U}_{i-1,\omega}^{(\theta,-i,k)} - F \circ U_{i-1,\omega}^{(\theta,-i,k)}\right\|_{L^2(\mu)}
$$

$$
\leq \mathbb{1}_{\mathbb{N}}(n)\left[\|\widehat{g} - g\|_{L^2(\mu)} + 2Tn\left\|\widehat{F} - F\right\|_{C^0(\mathbb{R})} + \left((n-1) + \frac{(n-1)n}{2} + \frac{(n+1)n}{2}\right)\|\mathcal{I} - \mathrm{Id}\|_{C^0}\right],
$$

$$
+ LT\sum_{i=0}^{n-1}\left(\left\|\widehat{U}_i^{(\theta,i,k)} - U_i^{(\theta,i,k)}\right\|_{L^2(\mu)} + \mathbb{1}_{\mathbb{N}}(i)\left\|\widehat{U}_{i-1}^{(\theta,-i,k)} - U_{i-1}^{(\theta,-i,k)}\right\|_{L^2(\mu)}\right)
$$

$$
\leq \mathbb{1}_{\mathbb{N}}(n)2^n\left[\|\widehat{g} - g\|_{L^2(\mu)} + T\left\|\widehat{F} - F\right\|_{C^0(\mathbb{R})} + n\|\mathcal{I} - \mathrm{Id}\|_{C^0}\right]
$$

$$
+ \sum_{i=0}^{n-1}(\max\{1, LT\})^{n-i}\left(\left\|\widehat{U}_i^{(\theta,i,k)} - U_i^{(\theta,i,k)}\right\|_{L^2(\mu)} + \mathbb{1}_{\mathbb{N}}(i)\left\|\widehat{U}_{i-1}^{(\theta,-i,k)} - U_{i-1}^{(\theta,-i,k)}\right\|_{L^2(\mu)}\right).
\tag{C.40}
$$

Now let us set for every $k \in \mathbb{N}_0$,

$$
x_k = \sup_{\theta \in \Theta}\left\|\widehat{U}_{k,\omega}^\theta(t, \cdot) - U_{k,\omega}^\theta(t, \cdot)\right\|_{L^2(\mu)},
\tag{C.41}
$$

and in addition we define $\alpha_0 = \|\widehat{g} - g\|_{L^2(\mu)} + T\left\|\widehat{F} - F\right\|_{C^0(\mathbb{R})}$, $\alpha_1 = \|\mathcal{I} - \mathrm{Id}\|_{C^0}$ and $\beta = 2 + LT$. Taking the supremum over all $\theta \in \Theta$ in (C.40) gives us for all $k \in \mathbb{N}_0$ that,

$$
x_k \leq \mathbb{1}_{\mathbb{N}}(k)(\alpha_0 + \alpha_1 k)\beta^k + \sum_{i=0}^{k-1}\beta^{k-i}(x_i + x_{\max\{i-1,0\}}).
\tag{C.42}
$$

Therefore, we can use Lemma C.2 with $\gamma \leftarrow 0$ then gives us that for all $k \in \mathbb{N}_0$ it holds that,

$$
\sup_{\theta \in \Theta}\left\|\widehat{U}_{k,\omega}^\theta(t, \cdot) - U_{k,\omega}^\theta(t, \cdot)\right\|_{L^2(\mu)}
$$

$$
\leq \mathbb{1}_{\mathbb{N}}(k)\frac{(1+\sqrt{2})^k}{2}\left(\|\widehat{g} - g\|_{L^2(\mu)} + T\left\|\widehat{F} - F\right\|_{C^0(\mathbb{R})} + \|\mathcal{I} - \mathrm{Id}\|_{C^0}\right)(2 + LT)^k.
\tag{C.43}
$$

Next we define

$$
C_1 = (T+1)(1 + \mathcal{L}\exp(LT)), \qquad C_2 = 5 + 3LT.
\tag{C.44}
$$

Combining (C.36) with (C.43) then gives us that,

$$
\left\|\widehat{U}_{n,\omega}^0(t, \cdot) - u(t, \cdot)\right\|_{L^2(\mu)}
$$

$$
\leq \left\|\widehat{U}_{n,\omega}^0(t, \cdot) - U_{n,\omega}^0(t, \cdot)\right\|_{L^2(\mu)} + \left\|U_{n,\omega}^0(t, \cdot) - u(t, \cdot)\right\|_{L^2(\mu)}
\tag{C.45}
$$

$$
\leq C_1 C_2^n\left(\|\widehat{g} - g\|_{L^2(\mu)} + \left\|\widehat{F} - F\right\|_{C^0(\mathbb{R})} + \|\mathcal{I} - \mathrm{Id}\|_{C^0} + m^{-n/2}\exp(m/2)\right).
$$

For an arbitrary $\sigma > 0$, we choose

$$m = C_2^{2(1+1/\sigma)}, \qquad n = \sigma \log_{C_2}\left(4C_1 \exp\left(m/2\right)/\varepsilon\right) \tag{C.46}$$

and if we choose $\widehat{g} = \widehat{g}_\delta$ and $\widehat{F} = \widehat{F}_\delta$ such that,

$$\|\widehat{g} - g\|_{L^2(\mu)} \leq \delta = \frac{\varepsilon}{4C_1 C_2^n} = \frac{\varepsilon^{1+\sigma}}{(4C_1)^{1+\sigma} \exp\left(\sigma m/2\right)}, \tag{C.47}$$

then we obtain that

$$\left\|\widehat{U}_{n,\omega}^0(t,\cdot) - u(t,\cdot)\right\|_{L^2(\mu)} \leq \varepsilon. \tag{C.48}$$

**Step 3: size estimate.** We now provide estimates on the size of the network constructed in Step 1. First of all, it is straightforward to see that the depth of the network can be bounded by

$$\mathcal{L}_\varepsilon(\widehat{U}_{n,\omega}^0) \leq \mathcal{L}_\delta(\widehat{g}) + (n-1)\mathcal{L}_\delta(\widehat{F}) \leq \mathcal{L}_\delta(\widehat{g}) + \log_{C_2}\left(3C_1 \exp\left(m/2\right)/\varepsilon\right)\mathcal{L}_\delta(\widehat{F}). \tag{C.49}$$

Next we prove an estimate on the number of needed neurons. For notation, we write $\mathcal{M}_n = \mathcal{M}_\varepsilon(\widehat{U}_{n,\omega}^0)$. We find that for all $0 \leq k \leq n$,

$$\mathcal{M}_k \leq \mathbb{1}_\mathbb{N}(k)m^k(\mathcal{M}_\delta(\widehat{g}) + (k-1)\mathcal{M}_\delta(\mathcal{I}))$$

$$+ \sum_{i=0}^{k-1} m^{k-i}(2\mathcal{M}_\delta(\widehat{F}) + (2k-2i-1)\mathcal{M}_\delta(\mathcal{I}) + \mathcal{M}_i + \mathcal{M}_{\max\{i-1,0\}}) \tag{C.50}$$

$$\leq \mathbb{1}_\mathbb{N}(k)(\mathcal{M}_\delta(\widehat{g}) + \mathcal{M}_\delta(\widehat{F}) + k\mathcal{M}_\delta(\mathcal{I}))(2m)^k + \sum_{i=0}^{k-1} m^{k-i}(\mathcal{M}_i + \mathcal{M}_{\max\{i-1,0\}}).$$

Applying Lemma C.2 to (C.50) (i.e. $\alpha_0 \leftarrow \mathcal{M}_\delta(\widehat{g}) + \mathcal{M}_\delta(\widehat{F})$, $\alpha_1 \leftarrow \mathcal{M}_\delta(\mathcal{I})$ and $\beta \leftarrow 2m$) then gives us that

$$\mathcal{M}_n \leq \frac{1}{2}(\mathcal{M}_\delta(\widehat{g}) + \mathcal{M}_\delta(\widehat{F}) + \mathcal{M}_\delta(\mathcal{I}))(1 + \sqrt{2})^n(2m)^n. \tag{C.51}$$

Observing that $2 + 2\sqrt{2} \leq C_2$ and recalling that $m = C_2^{2(1+1/\sigma)}$ we find that

$$\mathcal{M}_n \leq \frac{1}{2}(\mathcal{M}_\delta(\widehat{g}) + \mathcal{M}_\delta(\widehat{F}) + \mathcal{M}_\delta(\mathcal{I}))C_2^{(3\sigma+2)n/\sigma}$$

$$= \frac{1}{2}(\mathcal{M}_\delta(\widehat{g}) + \mathcal{M}_\delta(\widehat{F}) + \mathcal{M}_\delta(\mathcal{I}))\left(\frac{4C_1 \exp\left(m/2\right)}{\varepsilon}\right)^{2+3\sigma}. \tag{C.52}$$

For the width, we make the estimate $\text{width}_\varepsilon(\widehat{U}_{n,\omega}^0) \leq \mathcal{M}_n$.

$\square$

## C.6   PINN approximation of nonlinear parabolic equations

**Setting C.12.** *Assume Setting C.7, let $\widehat{g} \in C(\mathbb{T}^d, \mathbb{R}) \cap L^2(\mu)$[3] and let $\omega : [0,T] \times \mathbb{N}^2 \to \Omega$ be defined as in (C.37) in the proof of Theorem C.11. Let $\widehat{U}_{n,\omega}^\theta : [0,T] \times \mathbb{T}^d \times \Omega \to \mathbb{R}$, $n \in \mathbb{Z}$, $\theta \in \Theta$, satisfy for all $n \in \mathbb{N}_0$, $\varepsilon > 0$, $\theta \in \Theta$, $t \in [0,T]$, $x \in \mathbb{T}^d$ that*

$$\widehat{U}_{n,\omega}^\theta(t,x) = \frac{\mathbb{1}_\mathbb{N}(n)}{m^n}\left[\sum_{k=1}^{m^n}(\mathcal{I}_\varepsilon^{n-1} \circ \widehat{g})(x + W_{T-t}^{(\theta,0,-k)}(\omega(t,n,m)))\right]$$

$$+ \sum_{i=0}^{n-1} \frac{(T-t)}{m^{n-i}}\left[\sum_{k=1}^{m^{n-i}}\left((\mathcal{I}_\varepsilon^{n-i-1} \circ \widehat{F}_\varepsilon)(\widehat{U}_{i,\omega}^{(\theta,i,k)})\right.\right.$$

$$\left.\left. - \mathbb{1}_\mathbb{N}(i)(\mathcal{I}_\varepsilon^{n-i} \circ \widehat{F}_\varepsilon)(\widehat{U}_{i-1,\omega}^{(\theta,-i,k)})\right)\left(\mathcal{U}_t^{(\theta,i,k)}(\omega(t,n,m)), x + W_{\mathcal{U}_t^{(\theta,i,k)}-t}^{(\theta,i,k)}(\omega(t,n,m)))\right)\right]. \tag{C.53}$$

---

[3]The function $\widehat{g}$ can but need not be the same as the function $\widehat{g}_\varepsilon$, for some $\varepsilon > 0$, of Setting C.7.

**Lemma C.13.** *Assume Setting C.12. Under the assumption that,*

$$\max_{1\le j\le k}\left\|\mathcal{I}^j\right\|_{C^k([-\mathcal{L}-1,\mathcal{L}+1])} \le 2, \tag{C.54}$$

*where $\mathcal{I}^j$ denotes $j$ compositions of $\mathcal{I}$, it holds for all $\ell, k \in \mathbb{N}_0$ that,*

$$\sup_{\theta\in\Theta}\left\|\widehat{U}^\theta_{k,\omega}\right\|_{C^{(0,\ell)}([0,T]\times\mathbb{T}^d)} \le C_{k,\ell} := \left[\|\widehat{g}\|_{C^\ell(\mathbb{T}^d)} + 2B_\ell(1+\sqrt{2})^k(1+2B_\ell T\left\|\widehat{F}\right\|_{C^\ell(\mathbb{R})}^\ell)^k\right]^{2(\ell+1)!}, \tag{C.55}$$

*and where $B_\ell$ denote the $\ell$-th Bell number i.e., the number of possible partitions of a set with $\ell$ elements.*

*Proof.* We prove the claim by induction on $\ell$.

*Base case.* From Definition C.8, we find that for $\ell = 0$ and all $k \in \mathbb{N}_0$ it holds that

$$\sup_{\theta\in\Theta}\left\|\widehat{U}^\theta_{k,\omega}\right\|_{C^0([0,T]\times\mathbb{T}^d)} \le \|\widehat{g}\|_{C^0(\mathbb{T}^d)} + 2T\left\|\widehat{F}\right\|_{C^0(\mathbb{R})}k. \tag{C.56}$$

Claim (C.55) follows immediately for $\ell = 0$.

*Induction step.* We assume that claim (C.55) holds true for all $0 \le \ell^* \le \ell - 1$ and $k \in \mathbb{N}_0$. From this assumption, we will deduce that (C.55) holds true for $\ell$ and all $k \in \mathbb{N}_0$. We first observe that it follows from Lemma C.4, the induction hypothesis and the fact that $(C_{k,\ell})_{\ell\ge 0}$ is non-decreasing for any $k$, that for all $\theta \in \Theta$ and $0 \le i, j \le k$ it holds that,

$$\left|(\mathcal{I}^j\circ\widehat{F})(\widehat{U}^\theta_{i,\omega})\right|_{C^\ell([0,T]\times\mathbb{T}^d)} \le \left\|\mathcal{I}^j\circ\widehat{F}\right\|_{C^\ell(\mathbb{R})}\left(B_\ell C^\ell_{k,\ell-1} + \left|\widehat{U}^\theta_{i,\omega}\right|_{C^{(0,\ell)}([0,T]\times\mathbb{T}^d)}\right), \tag{C.57}$$

and where (again using Lemma C.4) it holds that $\left\|\mathcal{I}^j\circ\widehat{F}\right\|_{C^\ell(\mathbb{R})} \le 2B_\ell\left\|\widehat{F}\right\|_{C^\ell}^\ell$.

Using this estimate and the fact that $(C_{k,\ell})_{k\ge 0}$ is non-decreasing for any $\ell$, we can make the following calculation for every $k \in \mathbb{N}_0$,

$$\sup_{\theta\in\Theta}\left|\widehat{U}^\theta_{k,\omega}\right|_{C^{(0,\ell)}([0,T]\times\mathbb{T}^d)}$$

$$\le \mathbb{1}_{\mathbb{N}}(k)|\widehat{g}|_{C^\ell(\mathbb{T}^d)} + T\sum_{i=0}^{k-1}\sup_{\theta\in\Theta}\left|(\mathcal{I}^{k-i-1}\circ\widehat{F})(\widehat{U}^\theta_{i,\omega})\right|_{C^{(0,\ell)}(([0,T]\times\mathbb{T}^d))}$$

$$+ T\sum_{i=0}^{k-1}\mathbb{1}_{\mathbb{N}}(i)\sup_{\theta\in\Theta}\left|(\mathcal{I}^{k-i}\circ\widehat{F})(\widehat{U}^\theta_{i-1,\omega})\right|_{C^{(0,\ell)}(([0,T]\times\mathbb{T}^d))}$$

$$\le \mathbb{1}_{\mathbb{N}}(k)|\widehat{g}|_{C^\ell(\mathbb{T}^d)} + 2B_\ell T\left\|\widehat{F}\right\|_{C^\ell(\mathbb{R})}^\ell\sum_{i=0}^{k-1}\left(B_\ell C^\ell_{k,\ell-1} + \sup_{\theta\in\Theta}\left|\widehat{U}^\theta_{i,\omega}\right|_{C^{(0,\ell)}(([0,T]\times\mathbb{T}^d))}\right)$$

$$+ 2B_\ell T\left\|\widehat{F}\right\|_{C^\ell(\mathbb{R})}^\ell\sum_{i=0}^{k-1}\mathbb{1}_{\mathbb{N}}(i)\left(B_\ell C^\ell_{k,\ell-1} + \sup_{\theta\in\Theta}\left|\widehat{U}^\theta_{i-1,\omega}\right|_{C^{(0,\ell)}(([0,T]\times\mathbb{T}^d))}\right)$$

$$\le \mathbb{1}_{\mathbb{N}}(k)(|\widehat{g}|_{C^\ell(\mathbb{T}^d)} + 2B_\ell T\left\|\widehat{F}\right\|_{C^\ell(\mathbb{R})}^\ell C^\ell_{k,\ell-1}k)$$

$$+ \sum_{i=0}^{k-1}2B_\ell T\left\|\widehat{F}\right\|_{C^\ell(\mathbb{R})}^\ell\left(\sup_{\theta\in\Theta}\left|\widehat{U}^\theta_{i,\omega}\right|_{C^{(0,\ell)}(([0,T]\times\mathbb{T}^d))} + \mathbb{1}_{\mathbb{N}}(i)\sup_{\theta\in\Theta}\left|\widehat{U}^\theta_{i-1,\omega}\right|_{C^{(0,\ell)}(([0,T]\times\mathbb{T}^d))}\right) \tag{C.58}$$

Application of Lemma C.2 with $\alpha_0 \leftarrow |\widehat{g}|_{C^\ell(\mathbb{T}^d)}, \alpha_1 \leftarrow 2B_\ell C^\ell_{k,\ell-1}, \beta \leftarrow (1 + 2B_\ell T \|\widehat{F}\|^\ell_{C^\ell(\mathbb{R})})$ and $\gamma \leftarrow 0$ gives us

$$
\begin{aligned}
\sup_{\theta \in \Theta} \left\| \widehat{U}^\theta_{k,\omega} \right\|_{C^0([0,T] \times \mathbb{T}^d)} &\leq \frac{|\widehat{g}|_{C^\ell(\mathbb{T}^d)} + 2B_\ell C^\ell_{k,\ell-1}}{2} (1 + \sqrt{2})^k (1 + 2B_\ell T \|\widehat{F}\|^\ell_{C^\ell(\mathbb{R})})^k \\
&\leq \left[ |\widehat{g}|_{C^\ell(\mathbb{T}^d)} + 2B_\ell(1 + \sqrt{2})^k (1 + 2B_\ell T \|\widehat{F}\|^\ell_{C^\ell(\mathbb{R})})^k \right] C^\ell_{k,\ell-1}.
\end{aligned} \tag{C.59}
$$

Filling in the definition of $C_{k,\ell-1}$ indeed gives us the formula as stated in (C.55), thereby concluding the proof of the claim. $\qquad\square$

**Lemma C.14.** *Let $F$ be a polynomial. For every $\sigma, \varepsilon > 0$ there is an operator $\mathcal{U}^\varepsilon$ as in Assumption 3.1 such that for every $t \in [0, T]$,*

$$
\left\| \mathcal{U}^\varepsilon(u_0, t) - \mathcal{G}(v)(u_0, t) \right\|_{L^2(\mathbb{T}^d)} \leq \varepsilon, \qquad C^B_{\varepsilon, \ell} \leq C(B\varepsilon^{-\sigma})^{2(l+1)!}, \qquad C^\varepsilon_{\text{stab}} \leq C\varepsilon^{-\sigma}. \tag{C.60}
$$

*Moreover it holds that* $\operatorname{depth}(\mathcal{U}^\varepsilon(u_0, t)) \leq \operatorname{depth}(\widehat{u}_0) + C \ln(\varepsilon^{-1})$, $\operatorname{width}(\mathcal{U}^\varepsilon(u_0, t)) \leq \operatorname{width}(\widehat{u}_0)\varepsilon^{-2-\sigma}$ *and* $\operatorname{size}(\mathcal{U}^\varepsilon(u_0, t)) \leq \operatorname{size}(\widehat{u}_0)\varepsilon^{-2-\sigma}$.

*Proof.* The three bounds are a consequence of, respectively, Theorem C.11 and Lemma C.13 and (C.45). The size estimates follow from Theorem C.11. Note that one might have to rescale the constant $\sigma > 0$. $\qquad\square$

# D    Additional material for Section 4.2

## D.1    Errors of DeepONets

In [43], numerous error estimates for DeepONets are proven, with a focus on DeepONets that use the ReLU activation function. In order to quantify this error, the authors fix a probability measure $\mu \in \mathcal{P}(\mathcal{X})$ and define the error as,

$$
\widehat{\mathscr{E}} = \left( \int_{\mathcal{X}} \int_U |\mathcal{G}(u)(y) - \mathcal{G}_\theta(u)(y)|^2 \, dy \, d\mu(u) \right)^{1/2}, \tag{D.1}
$$

assuming that there exist embeddings $\mathcal{X} \hookrightarrow L^2(D)$ and $\mathcal{Y} \hookrightarrow L^2(U)$. From [43, Lemma 3.4], it then follows that $\widehat{\mathscr{E}}$ (D.1) can be bounded as,

$$
\widehat{\mathscr{E}} \leq \operatorname{Lip}_\alpha(\mathcal{G})\operatorname{Lip}(\mathcal{R} \circ \mathcal{P}) (\widehat{\mathscr{E}}_\mathcal{E})^\alpha + \operatorname{Lip}(\mathcal{R})\widehat{\mathscr{E}}_\mathcal{A} + \widehat{\mathscr{E}}_\mathcal{R}, \tag{D.2}
$$

where $\operatorname{Lip}_\alpha(\cdot)$ denotes the $\alpha$-Hölder coefficient of an operator and where $\widehat{\mathscr{E}}_\mathcal{E}$ quantifies the encoding error, where $\widehat{\mathscr{E}}_\mathcal{A}$ is the error incurred in approximating the approximator $\mathcal{A}$ and where $\widehat{\mathscr{E}}_\mathcal{R}$ quantifies the reconstruction error. Assuming that all Hölder coefficients are finite, one can prove that $\widehat{\mathscr{E}}$ is small if $\widehat{\mathscr{E}}_\mathcal{E}, \widehat{\mathscr{E}}_\mathcal{A}$ and $\widehat{\mathscr{E}}_\mathcal{R}$ are all small. We summarize how each of these three errors can be bounded using the results from [43].

- The upper bound on the encoding error $\widehat{\mathscr{E}}_\mathcal{E}$ depends on the chosen sensors and the spectral decay rate for the covariance operator associated to the measure $\mu$. Use bespoke sensor points to obtain optimals bounds when possible, otherwise use random sensors to obtain almost optimal bounds. More information can be found in [43, Section 3.5].

- The upper bound on the reconstruction error $\widehat{\mathscr{E}}_\mathcal{R}$ depends on the smoothness of the operator and the chosen basis functions $\tau$ i.e., neural networks, for the reconstruction operator $\mathcal{R}$. Following [43, Section 3.4], one first chooses a standard basis $\widetilde{\tau}$ of which the properties are well-known. We denote the corresponding reconstruction by $\widetilde{\mathcal{R}}$ and the corresponding reconstruction error by $\widehat{\mathscr{E}}_{\widetilde{\mathcal{R}}}$. In this work, we focus on Fourier and Legendre basis function,

both of which are introduced in **SM** A. One then proceeds by constructing the neural network basis $\boldsymbol{\tau}$ i.e., the trunk nets, that satisfy for some $\varepsilon > 0$ and $p \geq 1$ the condition

$$\max_{k=1,\ldots,p} \|\tau_k - \widetilde{\tau}_k\|_{L^2} \leq \frac{\varepsilon}{p^{3/2}}, \tag{D.3}$$

which is shown to imply that,

$$\widehat{\mathscr{E}}_{\mathcal{R}} \leq \widehat{\mathscr{E}}_{\widetilde{\mathcal{R}}} + C\varepsilon, \tag{D.4}$$

where $C \geq 1$ depends only on $\int_{L^2} \|u\|^2 \, d\mathcal{G}_{\#}\mu(u)$. Using standard approximation theory, one can calculate an upper bound on $\widehat{\mathscr{E}}_{\widetilde{\mathcal{R}}}$ and using neural network theory one can quantify the network size of $\boldsymbol{\tau}$ needed such that (D.3) is satisfied. For the Fourier and Legendre bases such results are presented in Lemma D.1 and Lemma D.2, respectively.

- The upper bound on the approximation error $\widehat{\mathscr{E}}_{\mathcal{A}}$ depends on the regularity of the operator $\mathcal{G}$. We present the tanh counterparts of some results of [43, Section 3.6] in the following sections, with the main result being Theorem D.6.

For bounded linear operators, these calculations are rather straightforward and are presented in [43, **SM** D]. For nonlinear operators, one has to complete all the above steps for each specific case. In [43, Section 4], this has been done for four types of differential equations.

## D.2 Auxiliary results for linear operators

Following Section D.1, we need results on the required neural network size to approximate the reconstruction basis to a certain accuracy (D.3). The following lemma provides such a result for the Fourier basis introduced in **SM** A.5.

**Lemma D.1.** *Let $s, d, p \in \mathbb{N}$. For any $\varepsilon > 0$, there exists a trunk net $\boldsymbol{\tau} : \mathbb{R}^d \to \mathbb{R}^p$ with 2 hidden layers of width $\mathcal{O}(p^{\frac{d+1}{d}} + ps\ln(ps\varepsilon^{-1}))$ and such that*

$$p^{3/2} \max_{j=1,\ldots,p} \|\tau_j - \mathbf{e}_j\|_{C^s([0,2\pi]^d)} \leq \varepsilon, \tag{D.5}$$

*where $\mathbf{e}_1, \ldots, \mathbf{e}_p$ denote the first $p$ elements of the Fourier basis, as in **SM** A.5.*

*Proof.* We note that each element in the (real) trigonometric basis $\mathbf{e}_1, \ldots, \mathbf{e}_p$ can be expressed in the form

$$\mathbf{e}_j(x) = \cos(\kappa \cdot x), \quad \text{or} \quad \mathbf{e}_j(x) = \sin(\kappa \cdot x), \tag{D.6}$$

for $\kappa = \kappa(j) \in \mathbb{Z}^d$ with $|\kappa|_\infty \leq N$, where $N$ is chosen as the smallest natural number such that $p \leq (2N+1)^d$. We focus only focus on the first form, as the proof for the second form is entirely similar. Define $f : [0, 2\pi]^d \to \mathbb{R} : x \mapsto \kappa \cdot x$ and $g : [-2\pi dN, 2\pi dN] \to \mathbb{R} : x \mapsto \cos(x)$. As $f([0, 2\pi]^d) \subset [-2\pi dN, 2\pi dN]$, the composition $g \circ f$ is well-defined and one can see that it coincides with a trigonometric basis function $\mathbf{e}_j$. Moreover, the linear map $f$ is a trivial neural network without hidden layers. Approximating $\mathbf{e}_j$ by a neural network $\tau_j$ therefore boils down to approximating $g$ by a suitable neural network.

From [15, Theorem 5.1] it follows that the function $g$ there exists an independent constant $R > 0$ such that for large enough $t \in \mathbb{N}$ there is a tanh neural network $\widehat{g}_t$ with two hidden layers and $\mathcal{O}(t + N)$ neurons such that

$$\|g - \widehat{g}_t\|_{C^s([-2\pi dN, 2\pi dN])} \leq 4(8(s+1)^3 R)^s \exp(t-s). \tag{D.7}$$

This can be proven from [15, eq. (74)] by setting $\delta \leftarrow \frac{1}{3}$, $k \leftarrow s$, $s \leftarrow t$, $N \leftarrow 2$ and using $\|g\|_{C^s} = 1$ and Stirling's approximation to obtain

$$\frac{1}{(t-s)!} \left(\frac{3}{2 \cdot 2}\right)^{t-s} \leq \frac{1}{\sqrt{2\pi(t-s)}} \left(\frac{e}{t-s}\right)^{t-s} \leq \exp(s-t) \quad \text{for } t > s + e^2. \tag{D.8}$$

Setting $t = \mathcal{O}(\ln(\delta^{-1}) + s\ln(s))$ then gives a neural network $\widehat{g}_t$ with $\|g - \widehat{g}_t\|_{C^s} < \eta$. Next, it follows from [15, Lemma A.7] that

$$\begin{aligned}\|g \circ f - \widehat{g}_t \circ f\|_{C^s([0,2\pi]^d)} &\leq 16(e^2 s^4 d^2)^s \|g - \widehat{g}_t\|_{C^s([-2\pi dN, 2\pi dN])} \|f\|_{C^s([0,2\pi]^d)}^s \\ &\leq 16(e^2 s^4 d^2)^s \eta (2\pi dN)^s.\end{aligned} \tag{D.9}$$

From this follows that we can obtain the desired accuracy (D.5) if we set $\tau_j = \widehat{g}_{t(\eta)} \circ f$ with

$$\eta = \frac{\varepsilon p^{-3/2}}{16(2\pi N d^3 e^2 s^4)^s}, \tag{D.10}$$

which amounts to $t = \mathcal{O}(s\ln(sN\varepsilon^{-1}))$. As a consequence, the tanh neural network $\tau_j$ has two hidden layers with $\mathcal{O}(s\ln(sN\varepsilon^{-1}) + N)$ neurons and therefore, by recalling that $p \sim N^d$, the combined network $\boldsymbol{\tau}$ has two hidden layers with

$$\mathcal{O}(p(s\ln(sN\varepsilon^{-1}) + N)) = \mathcal{O}(ps\ln(ps\varepsilon^{-1}) + p^{\frac{d+1}{d}}) \tag{D.11}$$

neurons. □

### D.3 Proof of Theorem 4.6

*Proof.* Consider the setting of Theorem 4.6. Using [43, Theorem D.3], the reasoning as in [43, Example D.4] and Lemma D.1 we find that there exists a constant $C = C(d, \ell) > 0$, such that for any $m, p, s \in \mathbb{N}$ there exists a DeepONet with trunk net $\boldsymbol{\tau}$ and branch net $\boldsymbol{\beta}$, such that

$$\mathrm{size}(\boldsymbol{\tau}) \leq C(p^{\frac{d+1}{d}} + ps\ln(ps\varepsilon^{-1})), \quad \mathrm{depth}(\boldsymbol{\tau}) = 3, \tag{D.12}$$

and where

$$\mathrm{size}(\boldsymbol{\beta}) \leq p, \quad \mathrm{depth}(\boldsymbol{\beta}) \leq 1, \tag{D.13}$$

and such that the DeepONet approximation error (D.1) is bounded by

$$\left\|\mathcal{G}(v) - \mathcal{G}_\theta(v)\right\|_{L^2(\mu \times \lambda)} \leq \varepsilon + C\exp\left(-c\,p^{1/d}\right) + C\exp\left(-\frac{c\,m^{1/d}}{\log(m)^{1/d}}\right). \tag{D.14}$$

Moreover, it holds that

$$\left|\mathcal{N}(u)(\cdot)\right|_{C^s} \leq Cp^{s/d}, \tag{D.15}$$

since in this case $\boldsymbol{\tau}$ approximates the Fourier basis (**SM** A.5). From (A.15), one can then deduce the estimate on the $C^s$-norm of the DeepONet. This proves that (3.7) in Theorem 3.9 holds with $\sigma(s) = s/d$. This concludes the proof. □

### D.4 Auxiliary results for nonlinear operators

We provide a neural network approximation result for the Legendre basis from **SM** A.4.

**Lemma D.2.** *Let $n, p \in \mathbb{N}$. For any $\varepsilon > 0$, there exists a trunk net $\boldsymbol{\tau} : \mathbb{R}^d \to \mathbb{R}^p$ with two hidden layers of width $\mathcal{O}(p)$ such that*

$$p^{3/2} \max_{j=1,\ldots,p} \|\tau_j - L_j\|_{C^s([-1,1]^d)} \leq \varepsilon, \tag{D.16}$$

*where $L_1, \ldots, L_p$ denote the first $p$ elements of the Legendre basis, as in **SM** A.4.*

*Proof.* Let $j \in 1, \ldots, p$. It holds by definition of Legendre polynomials and the corresponding enumeration (**SM** A.4) that the degree in every variable is at most $p$. Therefore, $L_j$ is a product of $d$ univariate polynomials of degree at most $p$. From [15, Lemma 3.2] it follows that one needs a shallow tanh neural network with $O(p)$ neurons to approximate a univariate polynomial to any accuracy. The result from [15, Corollary 3.7] can be used to construct a shallow tanh network that approximates the product of the $d$ univariate polynomials. Note that its size only depends on the dimension $d$ and not on the polynomial degree $p$ or the accuracy. Finally, [15, Lemma A.7] ensures the accuracy of the composition of the two subnetworks. It then follows that there exist a tanh neural network of width $\mathcal{O}(p)$ and two hidden layers that achieves the wanted error estimate. □

In our proofs, we require tanh counterparts to the results for DeepONets with ReLU activation function from [43]. We present these adapted results below for completeness.

The first lemma considers the neural network approximation of the map $\boldsymbol{u} \mapsto \widehat{Y}(\boldsymbol{u})$, as defined in [43, Eq. (3.59)].

**Lemma D.3.** *Let $N, d \in \mathbb{N}$, and denote $m := (2N+1)^d$. There exists a constant $C > 0$, independent of $N$, such that for every $N$ there exists a tanh neural network $\Psi : \mathbb{R}^m \to \mathbb{R}^m$, with*

$$\text{size}(\Psi) \le C(1 + m\log(m)), \quad \text{depth}(\Psi) \le C(1 + \log(m)), \tag{D.17}$$

*and such that $\Psi(\boldsymbol{u}) = (\widehat{Y}_1(\boldsymbol{u}), \dots, \widehat{Y}_m(\boldsymbol{u}))$, for all $\boldsymbol{u} \in \mathbb{R}^m$.*

*Proof.* The proof is identical to that of [43, Lemma 3.28]. $\square$

We can now state the following result [70, Theorem 3.10] which is the counterpart of [43, Theorem 3.32] for tanh neural networks.

**Theorem D.4.** *Let $V$ be a Banach space and let $\mathcal{J}$ be a countable index set. Let $\mathcal{F} : [-1,1]^{\mathcal{J}} \to V$ be a $(\boldsymbol{b}, \varepsilon, \kappa)$-holomorphic map for some $\boldsymbol{b} \in \ell^q(\mathbb{N})$ and $q \in (0,1)$, and an enumeration $\kappa : \mathbb{N} \to \mathcal{J}$. Then there exists a constant $C > 0$, such that for every $N \in \mathbb{N}$, there exists an index set*

$$\Lambda_N \subset \left\{ \boldsymbol{\nu} = (\nu_1, \nu_2, \dots) \in \textstyle\prod_{j \in \mathcal{J}} \mathbb{N}_0 \mid \nu_j \ne 0 \text{ for finitely many } j \in \mathcal{J} \right\}, \tag{D.18}$$

*with $|\Lambda_N| = N$, a finite set of coefficients $\{c_{\boldsymbol{\nu}}\}_{\boldsymbol{\nu} \in \Lambda_N} \subset V$, and a tanh network $\Psi : \mathbb{R}^N \to \mathbb{R}^{\Lambda_N}$, $y \mapsto \{\Psi_{\boldsymbol{\nu}}(y)\}_{\boldsymbol{\nu} \in \Lambda_N}$ with*

$$\text{size}(\Psi) \le C(1 + N\log(N)), \quad \text{depth}(\Psi) \le C(1 + \log\log(N)), \tag{D.19}$$

*and such that*

$$\sup_{\boldsymbol{y} \in [-1,1]^{\mathcal{J}}} \left\| \mathcal{F}(\boldsymbol{y}) - \sum_{\boldsymbol{\nu} \in \Lambda_N} c_{\boldsymbol{\nu}} \Psi_{\boldsymbol{\nu}}(y_{\kappa(1)}, \dots, y_{\kappa(N)}) \right\|_V \le CN^{1-1/q}. \tag{D.20}$$

Using this theorem, we can state the tanh counterpart to [43, Corollary 3.33].

**Corollary D.5.** *Let $V$ be a Banach space. Let $\mathcal{F} : [-1,1]^{\mathcal{J}} \to V$ be a $(\boldsymbol{b}, \varepsilon, \kappa)$-holomorphic map for some $\boldsymbol{b} \in \ell^q(\mathbb{N})$ and $q \in (0,1)$, where $\kappa : \mathbb{N} \to \mathcal{J}$ is an enumeration of $\mathcal{J}$. In particular, it is assumed that $\{b_j\}_{j \in \mathbb{N}}$ is a monotonically decreasing sequence. If $\mathcal{P} : V \to \mathbb{R}^p$ is a continuous linear mapping, then there exists a constant $C > 0$, such that for every $m \in \mathbb{N}$, there exists a tanh network $\Psi : \mathbb{R}^m \to \mathbb{R}^p$, with*

$$\text{size}(\Psi) \le C(1 + pm\log(m)), \quad \text{depth}(\Psi) \le C(1 + \log\log(m)), \tag{D.21}$$

*and such that*

$$\sup_{\boldsymbol{y} \in [-1,1]^{\mathcal{J}}} \|\mathcal{P} \circ \mathcal{F}(\boldsymbol{y}) - \Psi(y_{\kappa(1)}, \dots, y_{\kappa(m)})\|_{\ell^2(\mathbb{R}^p)} \le C\|\mathcal{P}\| m^{-s}, \tag{D.22}$$

*where $s := q^{-1} - 1 > 0$ and $\|\mathcal{P}\| = \|\mathcal{P}\|_{V \to \ell^2}$ denotes the operator norm.*

*Proof.* The proof is identical to the one presented in [43, Appendix C.18]. $\square$

Finally, we use this result to state the counterpart to [43, Theorem 3.34], which considers the approximation of a parametrized version of the operator $\mathcal{G}$, defined as a mapping

$$\mathcal{F} : [-1,1]^{\mathcal{J}} \to L^2(U) : \boldsymbol{y} \mapsto \mathcal{G}(u(\cdot; \boldsymbol{y})). \tag{D.23}$$

A more detailled discussion can be found in [43, Section 3.6.2].

**Theorem D.6.** *Let $\mathcal{F} : [-1,1]^{\mathcal{J}} \to L^2(U)$ be $(\boldsymbol{b}, \varepsilon, \kappa)$-holomorphic with $\boldsymbol{b} \in \ell^q(\mathbb{N})$ and $\kappa : \mathbb{N} \to \mathcal{J}$ an enumeration, and assume that $\mathcal{F}$ is given by (D.23). Assume that the encoder/decoder pair is constructed as in [43, Section 3.5.3], so that [43, Eq. (3.69)] holds. Given an affine reconstruction $\mathcal{R} : \mathbb{R}^p \to L^2(U)$, let $\mathcal{P} : L^2(U) \to \mathbb{R}^p$ denote the corresponding optimal linear projection [43, Eq. (3.17)]. Then given $k \in \mathbb{N}$, there exists a constant $C_k > 0$, independent of $m$, $p$ and an approximator $\mathcal{A} : \mathbb{R}^m \to \mathbb{R}^p$ that can be represented by a neural network with*

$$\text{size}(\mathcal{A}) \le C_k(1 + pm\log(m)), \quad \text{depth}(\mathcal{A}) \le C_k(1 + \log(m)).$$

*and such that the approximation error $\widehat{\mathscr{E}}_{\mathcal{A}}$ can be estimated by*

$$\widehat{\mathscr{E}}_{\mathcal{A}} \le C_k\|\mathcal{P}\| m^{-k},$$

*where $\|\mathcal{P}\| = \|\mathcal{P}\|_{L^2(U) \to \mathbb{R}^p}$ is the operator norm of $\mathcal{P}$.*

*Proof.* The proof is as in [43, Appendix C.19.1]. $\square$

## D.5 Gravity pendulum with external force

Next, we consider the following nonlinear ODE system, already considered in the context of approximation by DeepONets in [49] and [43],

$$\begin{cases} \dfrac{dv_1}{dt} = v_2, \\[2mm] \dfrac{dv_2}{dt} = -\gamma \sin(v_1) + u(t). \end{cases} \tag{D.24}$$

with initial condition $v(0) = 0$ and where $\gamma > 0$ is a parameter. Let us denote $v = (v_1, v_2)$ and

$$g(v) := \begin{pmatrix} v_2 \\ -\gamma \sin(v_1) \end{pmatrix}, \quad U(t) := \begin{pmatrix} 0 \\ u(t) \end{pmatrix}, \tag{D.25}$$

so that equation (D.24) can be written in the form

$$\mathcal{L}_u(v) := \frac{dv}{dt} - g(v) + U = 0, \quad v(0) = 0. \tag{D.26}$$

In (D.26), $v_1, v_2$ are the angle and angular velocity of the pendulum and the constant $\gamma$ denotes a frequency parameter. The dynamics of the pendulum is driven by an external force $u$. With the external force $u$ as the input, the output of the system is the solution vector $v$ and the underlying nonlinear operator is given by $\mathcal{G} : L^2([0,T]) \to L^2([0,T]) : u \mapsto \mathcal{G}(u) = v$. Following the discussion in [43], we choose an underlying (parametrized) measure $\mu \in \mathcal{P}(L^2([0,T]))$ as a law of a random field $u$, that can be expanded in the form

$$u(t; Y) = \sum_{k \in \mathbb{Z}} Y_k \alpha_k \mathbf{e}_k \left( \frac{2\pi t}{T} \right), \quad t \in [0,T], \tag{D.27}$$

where $\mathbf{e}_k(x)$, $k \in \mathbb{Z}$, denotes the one-dimensional standard Fourier basis (A.5) and where the coefficients $\alpha_k \geq 0$ decay to zero as $\alpha_k \leq C_\alpha \exp(-|k|\ell)$ for some constants $C_\alpha, \ell > 0$. Furthermore, we assume that the $\{Y_k\}_{k \in \mathbb{Z}}$ are iid random variables on $[-1, 1]$.

Assuming the described setting, the following lemma gives an error bound of tanh DeepONets in terms of the sizes of the corresponding branch and trunk nets.

**Lemma D.7.** *Consider the DeepONet approximation problem for the gravity pendulum (D.24), where the forcing $u(t)$ is distributed according to a probability measure $\mu \in \mathcal{P}(L^2([0,T]))$ given as the law of the random field (D.27). For any $k, r \in \mathbb{N}$, there exists a constant $C = C(k, r) > 0$, and a constant $c > 0$, independent of $m$, $p$, such that for any $m$, $p \in \mathbb{N}$, there exists a DeepONet $\mathcal{G}_\theta$ with trunk net $\boldsymbol{\tau}$ and branch net $\boldsymbol{\beta}$, such that*

$$\mathrm{size}(\boldsymbol{\tau}) \leq Cp, \quad \mathrm{depth}(\boldsymbol{\tau}) = 2, \tag{D.28}$$

*and*

$$\mathrm{size}(\boldsymbol{\beta}) \leq C(1 + pm\log(m)), \quad \mathrm{depth}(\boldsymbol{\beta}) \leq C(1 + \log(m)), \tag{D.29}$$

*and such that the DeepONet approximation error (D.1) is bounded by*

$$\widehat{\mathscr{E}} \leq Ce^{-c\ell m} + Cm^{-k} + Cp^{-r}, \tag{D.30}$$

*and that for all $s \in N$,*

$$\left| \mathcal{G}_\theta(u)(\cdot) \right|_{C^s} \leq Cp^{d/2 + 2sd}. \tag{D.31}$$

*Proof.* The proof of the statement is identical to that of [43, Theorem 4.10], with the only difference that we consider tanh neural networks instead of ReLU neural networks. As a result, the proof comes down to determining the size of the trunk net $\boldsymbol{\tau}$ using Lemma D.2 instead of [64, Proposition 2.10], thereby proving the tanh counterpart of [43, Proposition 4.5], and replacing [43, Proposition 4.9] by Theorem D.6. The $C^s$-bound of the DeepONet follows from the $C^s$-bound of Legendre polynomials (A.12) and Lemma D.2. □

We can again follow Theorem 3.9 to obtain error bounds for physics-informed DeepONets. Assumption 3.3 is satisfied for $[0, T]$. As a result, we can apply Theorem 3.9 to obtain the following result.

**Theorem D.8.** *Consider the setting of Lemma D.7. For every $\beta > 0$, there exists a constant $C > 0$ such that for any $p \in N$, there exists a DeepONet $\mathcal{G}_\theta$ with a trunk net $\boldsymbol{\tau} = (0, \tau_1, \ldots, \tau_p)$ with $p$ outputs and branch net $\boldsymbol{\beta} = (0, \beta_1, \ldots, \beta_p)$, such that*

$$\text{size}(\boldsymbol{\tau}) \le Cp, \quad \text{depth}(\boldsymbol{\tau}) = 2, \tag{D.32}$$

*and*

$$\text{size}(\boldsymbol{\beta}) \le C(1 + p^2 \log(p)), \quad \text{depth}(\boldsymbol{\beta}) \le C(1 + \log(p)), \tag{D.33}$$

*and such that*

$$\left\| \frac{d\mathcal{G}_\theta(u)_1}{dt} - \mathcal{G}_\theta(u)_2 \right\|_{L^2(\mu)} + \left\| \frac{d\mathcal{G}_\theta(u)_2}{dt} + \gamma \sin(\mathcal{G}_\theta(u)_1) - u(t) \right\|_{L^2(\mu)} \le Cp^{-\beta}. \tag{D.34}$$

*Proof.* Lemma D.7 with $s \leftarrow 1$, $k \leftarrow r$ and $m \leftarrow p$ then provides a DeepONet that satisfies the conditions of Theorem 3.9 with $r^* = +\infty$ and equation (3.7) with $\sigma(s) = d/2 + 2sd$. The smoothness of $v$ is guaranteed by [43, Lemma 4.3]. Moreover, it holds that,

$$\left\| \frac{d\mathcal{G}_\theta(u)_1}{dt} - \mathcal{G}_\theta(u)_2 \right\|_{L^2(\mu)} \le \left\| \frac{d\mathcal{G}_\theta(u)_1}{dt} - \frac{d\mathcal{G}(u)_1}{dt} \right\|_{L^2(\mu)} + \left\| \mathcal{G}(u)_2 - \mathcal{G}_\theta(u)_2 \right\|_{L^2(\mu)}, \tag{D.35}$$

and also that,

$$\left\| \frac{d\mathcal{G}_\theta(u)_2}{dt} + \gamma \sin(\mathcal{G}_\theta(u)_1) - u(t) \right\|_{L^2(\mu)}$$

$$\le \left\| \frac{d\mathcal{G}_\theta(u)_2}{dt} - \frac{d\mathcal{G}(u)_2}{dt} \right\|_{L^2(\mu)} + \gamma \left\| \sin(\mathcal{G}(u)_1) - \sin(\mathcal{G}_\theta(u)_1) \right\|_{L^2(\mu)} \tag{D.36}$$

$$\le \left\| \frac{d\mathcal{G}_\theta(u)_2}{dt} - \frac{d\mathcal{G}(u)_2}{dt} \right\|_{L^2(\mu)} + \gamma \left\| \mathcal{G}(u)_1 - \mathcal{G}_\theta(u)_1 \right\|_{L^2(\mu)}.$$

Combining this estimate with Theorem 3.9 with $k = 2$ then gives the wanted result. $\square$

## D.6 An elliptic PDE: Multi-d diffusion with variable coefficients

Next, again following [43], we consider a popular model problem for elliptic PDEs with unknown diffusion coefficients [11] and references therein. For the sake of definiteness and simplicity, we shall assume a periodic domain $D = \mathbb{T}^d$ in the following. For $b \in \mathbb{N}_0$, we consider an elliptic PDE with variable coefficients $a$,

$$\mathcal{L}_a(u) := \nabla \cdot (a(x)\nabla u(x)) + f(x) = 0, \tag{D.37}$$

for $u \in C^{b+2}(D)$ with suitable boundary conditions, and for fixed $f \in C^b(D)$. Similar to the previous examples, we fix a probability measure $\mu$ on the coefficient $a$ by assuming that every $a$ can be written as

$$a(x, Y) = \overline{a}(x) + \sum_{k \in \mathbb{Z}^d} \alpha_k Y_k \mathbf{e}_k(x), \tag{D.38}$$

with notation from **SM** A.5, and where for simplicity $\overline{a}(x) \equiv 1$ is assumed to be constant. Furthermore, we will consider the case of smooth coefficients $x \mapsto a(x; Y)$, which is ensured by requiring that there exist constants $C_\alpha > 0$ and $\ell > 1$, such that $|\alpha_k| \le C_\alpha \exp(-\ell|k|_\infty)$ for all $k \in \mathbb{Z}^d$. Still following [43], we define $\boldsymbol{b} = (b_1, b_2, \ldots) \in \ell^1(\mathbb{N})$ by

$$b_j := C_\alpha \exp(-\ell|\kappa(j)|_\infty), \tag{D.39}$$

where $\kappa : \mathbb{N} \to \mathbb{Z}^d$ is the enumeration for the standard Fourier basis, (**SM** A.5). Note that by assumption on the enumeration $\kappa$, we have that $\boldsymbol{b}$ is a monotonically decreasing sequence. In the following, we will assume throughout that $\|\boldsymbol{b}\|_{\ell^1} < 1$, ensuring a uniform coercivity condition on all random coefficients $a = a(\cdot; Y)$ in (D.37). Finally, we assume that the $Y_j \in [-1, 1]$ are centered random variables and we let $\mu \in \mathcal{P}(L^2(\mathbb{T}^d))$ denote the law of the random coefficient (D.38).

The following lemma provides an error estimate for DeepONets approximating the operator $\mathcal{G}$ that maps the input coefficient $a$ into the solution field $u$ of the PDE (D.37).

**Lemma D.9.** *For any $k, r \in \mathbb{N}$, there exists a constant $C > 0$, such that for any $m, p \in \mathbb{N}$, there exists a DeepONet $\mathcal{G}_\theta = \mathcal{R} \circ \mathcal{A} \circ \mathcal{E}$ with $m$ sensors, a trunk net $\boldsymbol{\tau} = (0, \tau_1, \dots, \tau_p)$ with $p$ outputs and branch net $\boldsymbol{\beta} = (0, \beta_1, \dots, \beta_p)$, such that*

$$\mathrm{size}(\boldsymbol{\beta}) \leq C(1 + pm\log(m)), \quad \mathrm{depth}(\boldsymbol{\beta}) \leq C(1 + \log(m)), \tag{D.40}$$

*and*

$$\mathrm{size}(\boldsymbol{\tau}) \leq Cp^{\frac{d+1}{d}} \quad \mathrm{depth}(\boldsymbol{\tau}) \leq 2 \tag{D.41}$$

*such that the DeepONet approximation error (D.1) satisfies*

$$\widehat{\mathscr{E}} \leq Ce^{-c\ell m^{\frac{1}{d}}} + Cm^{-k} + Cp^{-r}, \tag{D.42}$$

*and that for all $s \in N$*

$$\left| \mathcal{G}_\theta(u)(\cdot) \right|_{C^s} \leq Cp^{s/d}. \tag{D.43}$$

*Proof.* This statement is the tanh counterpart of [43, Theorem 4.19], which addresses ReLU Deep-ONets. We only highlight the differences in the proof. First, one should use Lemma D.1 instead of [43, Lemma 3.13], which then results in different network sizes in [43, Lemma 3.14, Proposition 3.17, Corollary 3.18, Proposition 4.17]. Second, one needs to replace [43, Proposition 4.18] with Theorem D.6.

Moreover, in this case the trunk net $\boldsymbol{\tau}$ approximates the Fourier basis (**SM** A.5). From (A.15), one can then deduce the estimate on the $C^s$-norm of the DeepONet. □

It is straightforward to verify that the conditions of Theorem 3.9 are satisfied in the current setting. Applying Theorem 3.9 then results in the following theorem on the error of physics-informed DeepONets for (D.37).

**Theorem D.10.** *Consider the elliptic equation (D.37) with $b \geq 1$. For every $\beta > 0$, there exists a constant $C > 0$ such that for any $p \in N$, there exists a DeepONet $\mathcal{G}_\theta$ with a trunk net $\boldsymbol{\tau} = (0, \tau_1, \dots, \tau_p)$ with $p$ outputs and branch net $\boldsymbol{\beta} = (0, \beta_1, \dots, \beta_p)$, such that*

$$\mathrm{size}(\boldsymbol{\beta}) \leq C(1 + p^2\log(p)), \quad \mathrm{depth}(\boldsymbol{\beta}) \leq C(1 + \log(p)), \tag{D.44}$$

*and*

$$\mathrm{size}(\boldsymbol{\tau}) \leq Cp^2 \quad \mathrm{depth}(\boldsymbol{\tau}) \leq 2 \tag{D.45}$$

*such that*

$$\left\| \nabla \cdot (a(x)\nabla\mathcal{G}_\theta(a)(x)) - f(x) \right\|_{L^2(\mu)} \leq Cp^{-\beta}. \tag{D.46}$$

*Proof.* We first check the conditions of Theorem 3.9. Lemma D.9 with $s \leftarrow 1$, $k \leftarrow r$ and $m \leftarrow p$ then provides a DeepONet that satisfies the conditions of Theorem 3.9 with $r^* = +\infty$ and equation (3.7) with $\sigma(s) = s/d$. Moreover, the following estimate holds,

$$
\begin{aligned}
&\left\| \nabla \cdot (a(x)\nabla\mathcal{G}_\theta(a)(x)) - f(x) \right\|_{L^2(\mu)} \\
&\leq \left\| \nabla \cdot (a(x)\nabla\mathcal{G}_\theta(a)(x)) - \nabla \cdot (a(x)\nabla\mathcal{G}(a)(x)) \right\|_{L^2(\mu)} \\
&\leq \sum_{j=1}^d \|a\|_{C^0} \left\| \partial_j^2(\mathcal{G}_\theta - \mathcal{G})(a)(x) \right\|_{L^2(\mu)} + \sum_{j=1}^d \|a\|_{C^1} \left\| \partial_j(\mathcal{G}_\theta - \mathcal{G})(a)(x) \right\|_{L^2(\mu)}.
\end{aligned}
\tag{D.47}
$$

Combining this estimate with Theorem 3.9 with $k = 2$ then gives the wanted result. □