# OpenReview forum: "Generic bounds on the approximation error for physics-informed (and) operator learning"
_NeurIPS.cc/2022/Conference — NeurIPS 2022 Accept_

### Official Review · Reviewer_Bnku · 2022-07-10

**Rating:** 4
**Confidence:** 3
**Soundness:** 2 fair
**Presentation:** 2 fair
**Contribution:** 2 fair

**Summary:**

This paper proposes a framework to derive some approximation error bounds for neural networks to approximate the solution operators of initial value problems.

**Questions:**

Please refer to the previous section.

**Limitations:**

I suggest the authors to clarify the function classes of L_a, G, and U.

**Strengths And Weaknesses:**

Strength:
Literature overview on recent operator learning such as DeepONets and Fourier Neural Operators is well written.

On the expressive power of neural networks for operators, I would suggest the following paper as an important previous work:

[GS19] Guss and Salakhutdinov, On Universal Approximation by Neural Networks with Uniform Guarantees on Approximation of Infinite Dimensional Maps, arxiv preprint:1910.01545

Weakness:
I found that the theoretical part is sloppy. First, the function classes are often lacking. For example, what is the differential operator L_a in Equation 2.1? Is G a bounded operator? In what norm? Secondly, many assumptions are incompletely stated without mentioning sufficient conditions when those are strictly satisfied. For example, in Assumption 3.1, both U^\epsilon and G are not defined. Dependence on a is unclear. Consequently, the main theorems and arguments are hard to read for mathematicians.

=== After Discussion with Authors
Theoretical details are clarified, and now I recognize the contributions clearer. So I raised my rating.

---

> ### Author Response · Authors · 2022-07-29
> **Reply to Reviewer Bnku**
>
> We start by thanking the reviewer for reading our paper and for your comments. Below, we address the concerns raised by the reviewer and thank the reviewer in advance for their patience in reading our detailed reply.
>
> 1. We would like to thank the reviewer for bringing reference [GS19] to our attention. We have cited it in the revised version of our article.
> 2. Regarding the reviewer's comment on *First, the function classes are often lacking. For example, what is the differential operator $L_a$ in Equation 2.1? Is G a bounded operator? In what norm?*, We start by saying that our article is meant for a very general audience, comprising of both mathematicians as well as machine learning and scientific computing domain experts, who need not necessarily be mathematicians. Hence, we had to present our results as well as the context for them such that the paper could appeal to this very general audience. Moreover, our main aim was to present a very general framework for obtaining error bounds for PINNs, Operator learning and Physics informed operator learning as well as their interconnections (Figure 1) such that practitioners can very easily and quickly verify the assumptions in their particular problems and use these generic bounds. Given these twin aims, we had to make some choices in our presentation and chose to keep the setting (Section 2) as general as possible while being mathematically precise in the assumptions, theorems and examples. Consequently, the precise function classes and domains and ranges of operators in section 2.1, where the problem has been introduced, were very general.  On the other hand, the function classes and precise definitions of operators involved were specified precisely in the assumptions and theorems of sections 3 and 4. Nonetheless, following the reviewer's suggestion, we have made the notation more precise in section 2.1 by making the following changes in the revised version,
>     - Eqn (2.1) represents a generic (time-dependent) PDE and $\mathcal{L}_a$ the underlying differential operator. We would like to point out that it is quite common in mathematical literature to denote a generic PDE as (2.1), while keeping the notation very general, see for instance references [59,60] and references therein. We have now defined the domain and range of this operator in terms of an abstract separable Banach space such that a very large range of PDEs can be considered. As examples, the linear Kolmogorov PDE of section 4.1, the nonlinear Allen-Cahn type parabolic PDE of section 4.3, the gravity pendulum ODE of SM Section D.5 and the elliptic PDE of SM section D.6, all have very precise definitions of the underlying differential operators as well as the function classes, to which both the inputs as well as the solutions belong to.
>     - The operator $\mathcal{G}$ is the input to solution operator for the generic PDE (2.1), which maps either the initial data (for a time-dependent problem) or the coefficient to the function space where the solution of the generic PDE (2.1) lives. This is a *bounded* operator in the corresponding *operator norm* for all the examples we considered in this paper and we have now specified this fact in our description of this input-to-solution operator, already in section 2.1 in the revised version. Moreover, the domain and range of concrete operators was previously specified, see for instance Theorems 3.6 and 3.9.
> 3. Regarding your comment *many assumptions are incompletely stated without mentioning sufficient conditions when those are strictly satisfied. For example, in Assumption 3.1, both $U^\epsilon$ and G are not defined*, we would like to politely disagree with your assertion by pointing out that the operator $\mathcal{U}^\epsilon$ was already defined above Assumption 3.1 and it is stated that the operator depends on the input $v$ (which can be a parameter $a$ or an initial condition $u_0$, depending on the context). The underlying function classes are made more precise in the revised version, while still keeping the generality of the framework for deriving generic bounds. We would also like to point out that the precise function classes for the underlying operators are specified in all the theorems, for instance Theorems 3.6 and 3.9.
> 4. Finally, the reviewer has given us a rating which amounts to saying that the paper *has technical flaws, weak evaluation, inadequate reproducibility or incompletely addressed ethical considerations and that the soundness and contribution of the paper is poor*. However, other than pointing out possible lack of precision in the notation in section 2.1, the reviewer has not commented on the technical content of the paper or pointed out possible flaws in our arguments or proofs of our theorems. We would be grateful if such flaws are pointed out and we will try to address them. Otherwise, we would like to request the reviewer to read the revised version and reconsider their negative evaluation of our paper.

---

> > ### Comment · Reviewer_Bnku · 2022-08-07
> > **Reply to rebuttal comments.**
> >
> > Thank you for your reply.
> >
> > I have read the rebuttal comments and skimmed the revised version. I found the notations are still vague.
> >
> > At L.82, L_a is clarified as a differential operator on a separable Banach space H that depends on a.
> > Can a “differential operator” always be defined on an abstract Banach space?
> > Where do you use the separable assumption on H?
> >
> > At L.86, G is simply stated to be bounded. Is G assumed to be bounded? or is it always bounded as a consequence of the definition of L_a stated as above? In Theorems 3.8, the topology of X is not specified (is it L^2(D)?).
> >
> > The paper is written as if the theorems cover everything. I do not think this style benefits “practitioners”.
> >
> > *operator  was already defined above Assumption 3.1*
> >
> > Assumption 3.1 is an assumption and the existence of U^\epsilon is not verified to be satisfied, and I consider the existence is nontrivial in general because it depends on G. If the assumptions are false, all the theorems can be automatically true.

---

> > > ### Author Response · Authors · 2022-08-07
> > > **Reply to Reviewer Bnku (part 1)**
> > >
> > > We start by thanking the reviewer for reading our rebuttal and for your reply. Below, we hope to address your concerns:
> > >
> > > 1\. As pointed out in our rebuttal, we seek to provide results that apply to as large classes of PDEs as possible, by putting as little constraints on the differential operator and input and output space thereof (and of the solution operator) as possible. To the best of our knowledge, there is no 'one fits all' notation for PDEs that allows more details than the current generic notation we have now used for $\mathcal{L}_a$, which, in any case, is widely used in the numerical analysis literature. Please note that we specify all details rigorously wherever possible, e.g. in our examples (see Section 4.1, Setting 4.1 and Theorem 4.2 for Linear Parabolic Kolmogorov PDEs, Setting 4.3 and Theorem 4.4 for Nonlinear Parabolic Allen-Cahn PDEs, SM Section D.5 for a nonlinear PDE and SM Section D.6 for a prototypical elliptic PDE). For all these examples, the exact function classes for the differential operator as well as the solution operator are rigorously specified. These examples also illustrate to a reader how the general theory develop in Section 3 can be applied in concrete situations.
> > >
> > > We had added the clarifications about the boundedness of $\mathcal{G}$ and the assumption on $\mathcal{H}$ as they still encompass the classes of PDEs we target.
> > >
> > > 2\. In Theorem 3.8, the set $\mathcal{X}$ is a subset of $L^2(D)$, as defined in Section 2.1. Also in equation (3.7) the $L^2(D)$-norm is used. Therefore, the results apply to the topology induced by the $L^2$-norm.
> > >
> > > [answer continues below]

---

> > > > ### Author Response · Authors · 2022-08-07
> > > > **Reply to Reviewer Bnku (part 2)**
> > > >
> > > > 3\. Regarding the concern that *The paper is written as if the theorems cover everything*, we would to like to clarify that this was never our intention. Starting from the abstract, it has been explicitly mentioned multiple times in the introduction, main body and discussion of the paper that we deduce how error estimates for different types of neural network architectures can be obtained *from one another*. We always mention that Assumption 3.1 is indeed essential, see the following examples as evidence of how we have presented the context for our results:
> > > > - Abstract L6: *Our framework utilizes existing neural network approximation results to obtain bounds on more involved learning architectures for PDEs*
> > > > - L60: *Our framework
> > > > is based on the observation that error estimates for different types of neural network architectures
> > > > can all be obtained from one another. As error estimates for neural network approximations of PDE
> > > > solutions at a fixed time are the easiest to obtain, and hence constitute the largest proportion of
> > > > currently available estimates, we devote particular attention to demonstrating how these available
> > > > estimates can be used to obtain novel bounds on the approximation error for space-time networks,
> > > > PINNs and (physics-informed) operator learning.*
> > > > - L132: *Every arrow in the flowchart represents a proof technique that allows one to transfer an error estimate
> > > > from one type of method to another.*
> > > > - Figure 1: It is highlighted that the top left estimate is assumed to be known and all the other bounds follow from it using different proof techniques.
> > > > - L134: *We give particular attention to the case where it is known that a neural network can efficiently
> > > > approximate the solution to a time-dependent PDE at a fixed time.*
> > > > - L294: *We demonstrate how Theorem 3.8 can be used to generalize available error estimates.*
> > > > - L316: *This is the first paper to rigorously expose the connections between the different deep learning
> > > > frameworks*
> > > > - L348:  *It is evident that the generic bounds presented here can only be obtained under suitable assumptions.*
> > > > - L351: *Assuming the existence of a neural network that approximates the solution
> > > > of PDE at a fixed time (Assumption 3.1) is of course essential, but such a result can usually be
> > > > obtained by emulating an existing numerical method.*
> > > >
> > > > The previous points also address your concern that *Assumption 3.1 is an assumption and the existence of $U^\epsilon$ is not verified to be satisfied, and I consider the existence is nontrivial in general because it depends on G.* As demonstrated above, we have stated multiple times that this assumption needs to be verified for our results to hold. Although nontrivial, this assumption is usually the simplest to verify in concrete examples as one often needs to emulate classical numerical methods such as a finite differences, finite elements, Monte Carlo, Picard iterations etc, see for instance [36,63,10,57] and references therein as a small subset of the widespread literature where this assumption is verified for different types of PDEs emulating different classical numerical methods. On the other hand, it was unclear how one can obtain error bounds for more involved learning architectures such as PINNs, DeepONets, FNOs etc.
> > > >
> > > > Hence, the main point of our paper is that given bounds on plain vanilla neural network approximations of PDEs at a fixed time, these can be translated into bounds on more involved architectures such as PINNs, DeepONets, FNOs, Physics-informed Operator Learning etc following the results in our paper.
> > > >
> > > > Moreover, we have also checked the veracity of this assumption in our paper in the concrete examples for parabolic PDEs (both linear and nonlinear, Section 4.1), nonlinear ODEs (SM D.5) and elliptic PDEs (SM D.6).

---

> > > > > ### Comment · Reviewer_Bnku · 2022-08-08
> > > > > **Thank you for your clarifications.**
> > > > >
> > > > > Since I was less familiar with PINN and Operator Learning, I surveyed previous works. The Introduction of this paper is well organized and helpful. Compared to previous theoretical papers like [35,40], this paper is much shorter as well.
> > > > >
> > > > > 1 and 3. … *there is no 'one fits all' notation for PDEs*
> > > > >
> > > > > I understand the convention/writing-style to state something in an abstract manner and then give nontrivial examples. However, it is better if the examples are presented as soon as possible.
> > > > >
> > > > > Typically, the existence of a solution operator depends on the domains, boundaries, initial values, parameterizations, norms, etc …. But in this study, all such arguments are replaced with Assumptions and Examples. In the current form, for example, Theorem 3.4 sounds tautological and confusing because formally it assumes that a solution operator G and NNs U^\epsilon exist to conclude a NN u_\theta exists. (I understand examples satisfying assumptions are provided later.) It could be better if Assumption 3.1 can be rephrased without neural networks. (I think this is possible because the network structure of U^\epsilon is not specified).
> > > > >
> > > > > 2. I see. I asked this because in Theorems such as 3.4, 3.8 and 3.9, G is assumed to be a mapping toward C^r(T^d).
> > > > >
> > > > >
> > > > >
> > > > > Minor comments:
> > > > >
> > > > > In Assumption 3.1. Is max_t a typo of sup_t? Since U^\epsilon is not continuous in t in general.
> > > > >
> > > > > In Assumption 3.1. The second condition “For any \ell \max_t | U^\epsilon(v,t) | is bounded” seems to be strong. Can we relax this?
> > > > >
> > > > > In Assumption 3.3. It is clearer to state “For any k” than “let k be …”.
> > > > >
> > > > > In Assumption 3.3. Is G_\theta = u_\theta?
> > > > >
> > > > > In Theorem 3.4. The space C^{s,r} and norms || . ||_{C^{s,r}} and || . ||_{\partial([0,T] x D)} are undefined. I guess M>2 because letting M=1 yields the RHS of (3.4) equals 0. Is it OK?
> > > > >
> > > > > In Theorem 3.6. What is L_N^2 ?
> > > > >
> > > > > In Theorem 3.8. C(\lambda) is undefined.
> > > > >
> > > > > In Setting 4.1 and 4.3. “let u \in C^{(s,r)}” is it assumed or concluded? I recommend “Assume/Suppose …” than “Let …” for assumptions. What is \beta?
> > > > >
> > > > > In Theorem 4.2 and 4.4. What is \sigma?

---

> > > > > > ### Author Response · Authors · 2022-08-08
> > > > > > **Reply to Reviewer Bnku**
> > > > > >
> > > > > > We start by thanking the reviewer for your prompt reading of our reply and for your comments and constructive suggestions. We reply to each of the points raised by you below:
> > > > > >
> > > > > > We fully agree with you that the existence (and regularity) of a solution operator depends on domains, boundaries, norms etc. Given the vast multitude of PDEs and of different conditions under which the underlying solution operator exists, we believe that it would be difficult to encapsulate these conditions under a single framework. Given our interest in developing techniques for bounds that can be very widely applied, we chose to focus on the current template of formulating assumptions and presenting examples that verify them. We hope that the reviewer can appreciate the rationale for our choices while at the same time, we concur that there could be other viable modes for presenting this material.
> > > > > >
> > > > > > Regarding your point that *Theorem 3.4 sounds tautological …*,  we would like to clarify that there is no tautology here as we use the assumption that there is a NN that approximates the solution of a time-dependent PDE, but only at fixed times, and from there, we prove that there is a NN that approximates this solution over space-time. This passage from fixed time snapshots to uniform in time bounds is the non-trivial part of this theorem. We will add a comment clarifying this fact in a camera-ready version of this paper, if accepted.
> > > > > >
> > > > > > Moreover, we fully agree with you that Assumption 3.1 and other results are not exclusively valid for neural networks, but for other approximation procedures too. We will add a remark to this effect in the next revised version.
> > > > > >
> > > > > > Below, we address your minor comments,
> > > > > >
> > > > > > - (Ass 3.1) Indeed, it should read sup_t, the second condition could be slightly relaxed but this might lead to more additional assumptions in some of our theorems. We will consider the benefits and drawbacks of doing this for the camera-ready version.
> > > > > > - (Ass 3.3) G_theta is an operator whereas u_theta is a function (solution of the PDE).
> > > > > > - (Thm 3.4) Actually, the assumption is that $s>k$. This typo will be fixed and a clarification about $C^{(s,r)}$ will also be added.
> > > > > > - (Thm 3.6) $L^2_N$ was introduced in the corresponding section about operator learning (Section 2.2) and is again defined in the proof of Thm 3.6 and in SM A.1, where we provide a glossary for the notation in the paper.
> > > > > > - (Thm 3.8) $C(\lambda)$ is defined in line 221.
> > > > > > - (Setting 4.1, 4.3) It is indeed an assumption, we will certainly change the wording. The constant $\sigma$ was defined in Thm 4.4 but was indeed forgotten in Thm 4.2.
> > > > > >
> > > > > > We thank you again for pointing out these typos as well as your suggestions, which will increase the quality of our paper. We will fix them for the next revision. In the hope that your concerns about the notation and clarity of our results are better addressed, we would be grateful if you would consider rating our paper more positively.

---

### Official Review · Reviewer_jfmr · 2022-07-10

**Rating:** 8
**Confidence:** 4
**Soundness:** 3 good
**Presentation:** 3 good
**Contribution:** 3 good

**Summary:**

This paper proposes a framework for analyzing the approximation  error of PINNs and operator learning. The framework was applied to obtain  error bounds for physics-informed operator learning methods for several class of PDEs. The error bounds show that dimension-independent convergence rate can be obtained if the solution is smooth enough.

**Questions:**

- The current results only prove the existence of some neural works achieving certain desired error accuracy. Can the authors comment on how the neural networks can be obtained? Especially, if one consider the empirical risk minimization approach, how does the generalization error scales?

- Theorem 4.6 concerns the approximation error of physical informed operator networks. The result is stated for the loss function $\|\mathcal{L}(\mathcal{G}_\theta)\|_{L^2}$. Can the author elaborate on whether this result implies an error bound for the operator itself, e.g. a bound for $\|\mathcal{G} -\mathcal{G}_{\theta}\|$? In addition, the Gaussian measure $\mu$ of the input functions requires subexponential decay in the KL expansion coefficients. Can this be relaxed?

- Is $p^\ast$ used in the statement of Assumption 3.5? If not, can the authors define it outside the Assumption?

- In Assumption 3.1, `` ...  there exist ... " there exist should be `` ... there exists ... "

- In Theorem B.5 of Appendix, are the indices $i$ and $j$ the same in equation (B.15)?


**Ethics Review Area:**

["I don’t know"]

**Limitations:**

The authors adequately addressed the limitations in the discussion section. I am not ware of any potential negative societal impact of their work.

**Strengths And Weaknesses:**

 - Strengths. This is a nice work on the error analysis of PINNs and operator learning, and to the best of my knowledge the first theoretical work on physics-informed operator networks. The main results are interesting, nontrivial and important for understanding the performance of operator learning in PDEs. The paper is very well written and easy to read.

- Weaknesses. The paper is strong enough to be accepted. I do not have much to comment on the weakness, except minor comments on the few main results and typos. See the section of Questions for more details.

---

> ### Author Response · Authors · 2022-07-29
> **Reply to Reviewer jfmr**
>
> We start by thanking the reviewer for your appreciation of the merits of our paper and your welcome suggestions for improving it. Below, we address the valid concerns raised by the reviewer and thank the reviewer in advance for their patience in reading our detailed reply.
>
> 1. Regarding your question about *current results only prove the existence of some neural works achieving certain desired error accuracy. Can the authors comment on how the neural networks can be obtained? Especially, if one consider the empirical risk minimization approach, how does the generalization error scales?*, this is indeed a very pertinent question. To answer it, we have analyzed how the generalization error (with respect to the empirical risk minimization) scales with the number of training samples and added a new section (section B.5) to the SM. Therein, we state and prove Theorem B.7, where a precise upper bound on the generalization error is derived. The rates are along expected lines as we use tools such as covering number arguments and concentration inequalities. Unfortunately, given the page limit in the main text, we have to add this result in the SM, but we have added a sentence in the discussion (page 9) directing the reader to this section on bounds on the generalization error. We thank the reviewer again for asking us to explore this additional avenue in the paper.
> 2. Regarding your question about *Does the PINN loss provide a bound on the approximation error of the operator*, we can reply that this is indeed the case. For instance, as seen in the proof of Theorem 4.6 (SM D.3), proving an upper bound on the error itself is an intermediate step in bounding the physics-informed loss (see eq. (D.14)).
> 3. Regarding your question about *can the assumption on sub-exponential decay of the KL eigenvalues be relaxed*, we would reply affirmatively as this assumption is taken from [42, Example D.4], see also SM D.3, and can be relaxed or adapted in various ways. In fact, any data distribution that leads to a convergence rate for DeepONets in terms of the number of data points and sensor points can be considered in our theorem. We have now clarified this issue in the main text itself.
> 4. Regarding $p^{\ast}$ in inequality (3.5), we thank the reviewer for correctly pointing this out and have defined the $p^{\ast}$ where it is needed.
> 5. The language in Assumption 3.1 is now corrected.
> 6. The indices in Equation (B.15) are fixed now and we thank the reviewer for pointing out these typos.

---

> > ### Author Response · Authors · 2022-08-09
> > **Reply to Reviewer jfmr (Contd)**
> >
> > As the author-reviewer discussion phase will soon conclude, we take the opportunity to thank the reviewer again for your very positive appreciation of our paper and your constructive comments and suggestions which enables us to improve the quality of our paper. We hope that we have addressed the remaining concerns of the reviewer to your satisfaction.

---

### Official Review · Reviewer_hG4A · 2022-07-11

**Rating:** 5
**Confidence:** 2
**Soundness:** 3 good
**Presentation:** 3 good
**Contribution:** 2 fair

**Summary:**

In this paper, the authors provide each bound and its connetions on PINN, DeepONet, FNO, physics-informed DeepONet, and physics-informed FNO. These general connections are also applied to show that the neural network representation can overcome the curse of dimensionality.

**Questions:**

## Questions
* The discussion says that assumption 3.3 is a natural assumption for various PDEs, but I wonder if this is correct. For example, I wonder if it is an assumption that can be easily made for the examples used in Burgers' and other operator learning papers.
* There is confusion in the notation of $u_\theta$, $\mathcal{G}_\theta(v)$ and $\mathcal{u}^\varepsilon$. Assumption3.1 and Assumption3.3 are assumptions related to $\mathcal{u}^\varepsilon$ and $\mathcal{G}_\theta(v)$, but Theorem 3.4 uses these assumptions to create $u_\theta$ and $\mathcal{L}(u_\theta)$. In case of PINN, $u_\theta$ is learned while $v$ is fixed.
* For the term $\|u_\theta-u\|_{L^2(\partial([0,T]\times D)}$ on the left side from the trace inequality in (3.4),  I wonder if this term affects the fact that there exists a nn that can sufficiently reduce the PINN loss.

## Minor
* Assumption 3.5. there is no $p^*$ is not in inequality (3.5)
* p.5 194 What is $L^2_N$?


**Ethics Review Area:**

["I don’t know"]

**Limitations:**

The authors noted their limitations and future works well in the conclusion section.

**Strengths And Weaknesses:**

## Strengths
* While various studies measured the approximation error for a specific PDE for one method among PINN, FNO, and DeepONet, this paper measured the error by connecting all methods at once for the first time.

## Weaknesses
* Theorem 3.4, Theorem 3.8, and Theorem 3.9 talked about the existence of nn that reduces the PDE residual. I can't guarantee it's a good match. These theorems only talk about the possibility of sufficiently reducing the loss. Furthermore, I wonder if the boundary condition or initial condition is not considered. Without these conditions, I am not sure if reducing $\|\mathcal{L}(\mathcal{G}_\theta)\|$ enough will lead to the convergence to the exact solution.
* [9] as well as [*] is a study showing that the curse of dimension can be solved when the solution of PDE is approximated by nn. Regarding these papers, it is necessary to explain the differences between Theorem 4.2 and Theorem 4.4 described in Section 4.1.




[*] Marwah, Tanya, Zachary Lipton, and Andrej Risteski. "Parametric complexity bounds for approximating PDEs with neural networks." Advances in Neural Information Processing Systems 34 (2021): 15044-15055.

---

> ### Author Response · Authors · 2022-07-29
> **Reply to Reviewer hG4A**
>
> We start by thanking the reviewer for the appreciation of the merits of our paper and the suggestions to improve it. Below, we address the valid concerns raised by the reviewer and thank the reviewer in advance for their patience in reading our detailed reply.
>
> 1. Regarding your point that *Theorems 3.4, 3.8 and 3.9 only talk about the possibility of sufficiently reducing the loss and if the boundary and initial conditions are not considered*, we agree that just reducing the PINN loss $\mathcal{L}(\mathcal{G}_\theta)$ does not guarantee that $\mathcal{G}_\theta$ is close to $\mathcal{G}$ i.e., the total error of the neural network approximation is sufficiently reduced. In this context, we would like to point out that the above theorems not only talk about reducing the PINN loss but also provide bounds on the total error. For instance, in Theorem 3.4, the $L^q$-loss (eqn. (3.3)) is minimized simultaneously with the PINN loss (eq. (3.4)). In Theorem 3.8, we already start from the assumption that $\mathcal{G}\approx \mathcal{G}_\theta$ (eq. (3.7)) and focus only on how small the PINN residual can be made. However, in the follow-up Theorem 3.9, where we had only mentioned that the PINN loss can be made arbitrarily small, the proof actually reveals that the $L^2$-loss is also simultaneously minimized. This fact is now clearly clarified in Theorem 3.9. Moreover, the boundary and initial conditions are taken care of in Theorem 3.4 by the second term of eq. (3.4) and in Theorem 3.8 and Theorem 3.9 by using suitable trace inequalities. Thus, these results prove that both the PINN loss as well as the total approximation error can be simultaneously minimized and we thank the reviewer for pointing this avenue for further sharpening our results.
> 2. Regarding *comparing theorems 4.2 and 4.4 with [9] and [$\ast$]*, we would like to say that [9] and [$\ast$] discuss the approximation error for *linear elliptic PDEs* in $H^1$-norm, resp. $L^2$-norm. Indeed, these results prove that the curse of dimensionality is broken for elliptic PDEs. On the other hand, Theorems 4.2 and 4.4 focus on linear parabolic and nonlinear parabolic time-dependent PDEs. Thus, there is no overlap between the two sets of results. However, given the very general framework for our bounds, we believe that applying results from Section 3 and combining them with [9,$\ast$] would lead to a proof of PINN losses being made as small as desired. Given this connection, we have added a sentence to these references in the discussion now.
> 3. Regarding your question about *if Assumption 3.3 is satisfied in general or not, and particularly for Burgers' and other PDEs considered in operator learning papers*, we would like to recall that Assumption 3.3 bounds the differential operator $\mathcal{L}$ in terms of suitable norms involving derivatives of the solution. We claim that this bound holds as long as classical solutions of the underlying PDE exist and are sufficiently regular for the PDE to hold in a pointwise sense. As this assumption is satisfied for a very large class of PDEs including all the PDEs considered in our article, assumption 3.3 holds. For instance, for the Burgers equation we have that,
> $\Vert\mathcal{L}(\mathcal{G}_{\theta}){\Vert} := \Vert\partial_t \mathcal{G}_\theta + \mathcal{G}_\theta\partial_x \mathcal{G}_\theta - \nu \partial^2_x \mathcal{G}_\theta \Vert \leq \Vert \partial_t(\mathcal{G}- \mathcal{G}_\theta)\Vert + \Vert\mathcal{G}_\theta\Vert _\infty \Vert \partial_x(\mathcal{G}- \mathcal{G}_\theta)\Vert + \Vert\partial_x \mathcal{G}\Vert _\infty \Vert\mathcal{G}-\mathcal{G}_\theta\Vert + \nu \Vert\partial^2_x(\mathcal{G}-\mathcal{G}_\theta)\Vert$, where $\Vert\cdot\Vert$ is the $L^q$-norm,
> thus verifying assumption 3.3. For other equations that are considered in operator learning papers (e.g. gravity pendulum, elliptic PDEs) this assumption holds as well and is e.g. verified in the proofs of Theorem D.8 and Theorem D.10, resp.
> 4. Regarding the *confusion in notation of Assumptions 3.1 and 3.3*, we apologize for this confusion. Indeed, as pointed out by you, for PINNs $v$ is fixed. So in this case one takes $\mathcal{X} = \{v\} = \{u_0\}$ and $\mathcal{G}(v) = u$ in Assumption 3.1 and 3.3. We have now clarified this notation in the revised version.
> 5. Regarding your question about *the left side from the trace inequality in (3.4), I wonder if this term affects the fact that there exists a nn that can sufficiently reduce the PINN loss*, we would like to point out that the neural network that is constructed in the proof of this theorem automatically minimizes both the PINN loss as well as the mentioned boundary term, so the proof is not affected by this term. We hopes that this adequately clarifies your concern.
> 6. Regarding *$p^{\ast}$ in (3.5)*, we thank the reviewer for correctly pointing this out and have defined the $p^{\ast}$ where it is needed.
> 7. The notation for $L^2_N$ is now specified and is stated in the glossary in SM A. 1.

---

> > ### Author Response · Authors · 2022-08-09
> > **Reply to Reviewer hG4A (Contd.)**
> >
> > As the author-reviewer discussion phase will soon conclude, we take the opportunity to thank the reviewer again for your constructive comments and suggestions which enables us to improve the quality of our paper. In the hope that we have addressed the concerns of the reviewer to your satisfaction, we would be very grateful if you could consider rating our paper more positively.

---

### Author Response · Authors · 2022-07-29
**Reply to all reviewers**

At the outset, we would like to thank the reviewers for their thorough and patient reading of our article. Their fair criticism and constructive suggestions have enabled us to improve the quality of our article. A revised version of the article is uploaded. We proceed to answer the points raised by each of the reviewers individually, below.

We would also like to point out that all the references to page and line numbers, sections, figures, tables, equation numbers and references, refer to those in the revised version.

Yours sincerely,

Authors of *Generic bounds on the approximation error for physics-informed (and) operator learning*

---

### Meta-Review · Area_Chair_9itE · 2022-08-25

**Recommendation:** Accept
**Confidence:** Certain

**Metareview:**

The paper studies approximation error bound for physics informed neural networks and operator learning. The result is technical sound and  useful for practical applications. The authors also adequately addressed concerns by the referees. The meta-reviewer recommends acceptance of the paper.

**Award:**

No

---

### Decision · Program_Chairs · 2022-09-14

Accept